# Byzantine Robustness and Partial Participation Can Be Achieved at Once: Just Clip Gradient Differences

**Grigory Malinovsky**[*]
KAUST,[†] MBZUAI[‡]

**Peter Richtárik**
KAUST

**Samuel Horváth**
MBZUAI

**Eduard Gorbunov**[§]
MBZUAI

## Abstract

Distributed learning has emerged as a leading paradigm for training large machine learning models. However, in real-world scenarios, participants may be unreliable or malicious, posing a significant challenge to the integrity and accuracy of the trained models. Byzantine fault tolerance mechanisms have been proposed to address these issues, but they often assume full participation from all clients, which is not always practical due to the unavailability of some clients or communication constraints. In our work, we propose the first distributed method with client sampling and provable tolerance to Byzantine workers. The key idea behind the developed method is the use of gradient clipping to control stochastic gradient differences in recursive variance reduction. This allows us to bound the potential harm caused by Byzantine workers, even during iterations when all sampled clients are Byzantine. Furthermore, we incorporate communication compression into the method to enhance communication efficiency. Under general assumptions, we prove convergence rates for the proposed method that match the existing state-of-the-art (SOTA) theoretical results. We also propose a heuristic on adjusting any Byzantine-robust method to a partial participation scenario via clipping.

## 1 Introduction

Distributed optimization problems are a cornerstone of modern machine learning research. They naturally arise in scenarios where data is distributed across multiple clients; for instance, this is typical in Federated Learning (FL) (Konečný et al., 2016; Kairouz et al., 2021). Such problems require specialized algorithms adapted to the distributed setup. Additionally, the adoption of distributed optimization methods is motivated by the sheer computational complexity involved in training modern machine learning models. Many models deal with massive datasets and intricate architectures, rendering training infeasible on a single machine (Li, 2020). Distributed methods, by parallelizing computations across multiple machines, offer a pragmatic solution to accelerate training and address these computational challenges, thus pushing the boundaries of machine learning capabilities.

To make distributed training accessible to the broader community, collaborative learning approaches have been actively studied in recent years (Kijsipongse et al., 2018b; Ryabinin and Gusev, 2020; Atre et al., 2021; Diskin et al., 2021a). In such applications, there is a high risk of the occurrence of so-called *Byzantine workers* (Lamport et al., 1982; Su and Vaidya, 2016)—participants who can violate the prescribed distributed algorithm/protocol either intentionally or simply because they are faulty. In general, such workers may even have access to some private data of certain participants and may collude to increase their impact on the training. Since the ultimate goal is to achieve robustness

---

[*]Part of the work was done when G. Malinovsky was visiting MBZUAI.

[†]King Abdullah University of Science and Technology.

[‡]Mohamed bin Zayed University of Artificial Intelligence.

[§]Corresponding author: eduard.gorbunov@mbzuai.ac.ae

in the worst case, many papers in the field make no assumptions limiting the power of Byzantine workers. Clearly, in this scenario, standard distributed methods based on the averaging of received information (e.g., stochastic gradients) are not robust, even to a single Byzantine worker. Such a worker can send an arbitrarily large vector that can shift the method arbitrarily far from the solution. This aspect makes it non-trivial to design methods with provable robustness to Byzantines (Baruch et al., 2019; Xie et al., 2020a). Despite all the challenges, multiple methods are developed/analyzed in the literature (Alistarh et al., 2018; Allen-Zhu et al., 2021; Wu et al., 2020; Zhu and Ling, 2021; Karimireddy et al., 2021, 2022; Gorbunov et al., 2022, 2023; Allouah et al., 2023).

However, literally, all existing methods with provable Byzantine robustness require *the full (or close to full) participation of clients or rely on extra assumptions*. The requirement of full participation is impractical for modern distributed learning problems since they can have millions of clients (Bonawitz et al., 2017; Niu et al., 2020). In such scenarios, it is more natural to use sampling of clients to speed up the training. Moreover, some clients can be unavailable at certain moments, e.g., due to a poor connection, low battery, or simply because of the need to use the computing power for some other tasks. Although *partial participation of clients* is a natural attribute of large-scale collaborative training, it is not studied under the presence of Byzantine workers. Moreover, this question is highly non-trivial: the existing methods can fail to converge if combined naïvely with partial participation since Byzantine can form a majority during particular rounds and, thus, destroy the whole training with just one round of communications. *Therefore, the field requires the development of new distributed methods that are provably robust to Byzantine attacks and can work with partial participation even when Byzantine workers form a majority during some rounds.*

**Our Contributions** We develop Byzantine-tolerant Variance-Reduced MARINA with Partial Participation (Byz-VR-MARINA-PP, Algorithm 1) – the first distributed method having Byzantine robustness and allowing partial participation of clients without strong additional assumptions. Our method uses variance reduction to handle Byzantine workers and clipping of stochastic gradient differences to bound the potential harm of Byzantine workers even when they form a majority during particular rounds of communication. To make the method even more communication efficient, we add communication compression. We prove the convergence of Byz-VR-MARINA-PP for general smooth non-convex functions and Polyak-Łojasiewicz functions. In the special case of full participation, our complexity bounds recover the ones for Byz-VR-MARINA (Gorbunov et al., 2023) that are the current SOTA convergence results. Moreover, we prove that in some cases, partial participation is theoretically beneficial for Byz-VR-MARINA-PP. We also propose a simplified version of Byz-VR-MARINA-PP with better neighborhood term in the convergence bounds (Byz-VR-MARINA-PP+, Algorithm 3) and a heuristic on how to use clipping to adapt any Byzantine-robust method to the partial participation setup and illustrate its performance in experiments.

## 1.1 Related Work

Below, we overview closely related works. Additional discussion is deferred to Appendix A.

**Byzantine robustness.** The primary vulnerability of standard distributed methods to Byzantine attacks lies in the aggregation rule: even one worker can arbitrarily distort the average. Therefore, many papers on Byzantine robustness focus on the application of robust aggregation rules, such as the geometric median (Pillutla et al., 2022), coordinate-wise median, trimmed median (Yin et al., 2018), Krum (Blanchard et al., 2017), and Multi-Krum (Damaskinos et al., 2019). However, simply robustifying the aggregation rule is insufficient to achieve provable Byzantine robustness, as illustrated by Baruch et al. (2019) and Xie et al. (2020a), who design special Byzantine attacks that can bypass standard defenses. This implies that more significant algorithmic changes are required to achieve Byzantine robustness, a point also formally proven by Karimireddy et al. (2021), who demonstrate that permutation-invariant algorithms – i.e., algorithms independent of the order of stochastic gradients at each step – cannot provably converge to any predefined accuracy in the presence of Byzantines.

Wu et al. (2020) are the first who exploit variance reduction to tolerate Byzantine attacks. They propose and analyze the method called Byrd-SAGA, which uses SAGA-type (Defazio et al., 2014) gradient estimators on the good workers and geometric median for the aggregation. Gorbunov et al. (2023) develop another variance-reduced method called Byz-VR-MARINA, which is based on (conditionally biased) GeomSARAH/PAGE-type (Horváth et al., 2023; Li et al., 2021) gradient estimator and any robust aggregation in the sense of the definition from (Karimireddy et al., 2021,

2022), and derive the improved convergence guarantees that are the current SOTA in the literature. There are also many other approaches and we discuss some of them in Appendix A.

**Partial participation and client sampling.** In the context of Byzantine-robust learning, there exists several works that develop and analyze methods with partial participation (Data and Diggavi, 2021; El-Mhamdi et al., 2021; Boubouh et al., 2022; Allouah et al., 2024a). However, these works rely on the restrictive assumption that the number of participating clients at each round is larger than the number of Byzantine workers. In this case, Byzantines cannot form a majority, and standard methods can be applied without any changes. In contrast, our method converges in more challenging scenarios, e.g., Byz-VR-MARINA-PP provably converges even when the server samples one client, which can be Byzantine. If the number of participating clients is such that Byzantine clients can form majority, these methods have a certain probability of divergence and this probability grows with each communication round. We provide a more detailed discussion in Appendix A.

## 2 Preliminaries

In this section, we formally introduce the problem, main definition, and assumptions used in the analysis. That is, we consider finite-sum distributed optimization problem[5]

$$\min_{x \in \mathbb{R}^d} \left\{ f(x) := \frac{1}{G} \sum_{i \in \mathcal{G}} f_i(x) \right\}, \quad f_i(x) := \frac{1}{m} \sum_{j=1}^m f_{i,j}(x) \quad \forall i \in \mathcal{G}, \tag{1}$$

where $\mathcal{G}$ is a set of regular clients of size $G := |\mathcal{G}|$. In the context of distributed learning, $f_i : \mathbb{R}^d \to \mathbb{R}$ corresponds to the loss function on the data of client $i$, and $f_{i,j} : \mathbb{R}^d \to \mathbb{R}$ is the loss computed on the $j$-th sample from the dataset of client $i$. Next, we assume that the set of all clients taking part in the training is $[n] = \{1, 2, \ldots, n\}$ and $\mathcal{G} \subseteq [n]$. The remaining clients $\mathcal{B} := [n] \setminus \mathcal{G}$ are Byzantine ones. We assume that $B := |\mathcal{B}| := \delta_{\text{real}} n \leq \delta n$, where $\delta_{\text{real}}$ is an exact ratio of Byzantine workers and $\delta$ is a known upper bound for $\delta_{\text{real}}$. We also assume that $0 \leq \delta_{\text{real}} \leq \delta < 1/2$ since otherwise Byzantine workers form a majority and problem (1) becomes impossible to solve in general.

**Notation.** We use a standard notation for the literature on distributed stochastic optimization. Everywhere in the text $\|x\|$ denotes a standard $\ell_2$-norm of $x \in \mathbb{R}^d$, $\langle a, b \rangle$ refers to the standard inner product of vectors $a, b \in \mathbb{R}^d$. The clipping operator is defined as follows: $\text{clip}_\lambda(x) := \min\{1, \lambda/\|x\|\}x$ for $x \neq 0$ and $\text{clip}_\lambda(0) := 0$. Finally, $\text{Prob}\{A\}$ denotes the probability of event $A$, $\mathbb{E}[\xi]$ is the full expectation of random variable $\xi$, $\mathbb{E}[\xi \mid A]$ is the expectation of $\xi$ conditioned on the event $A$. We also sometimes use $\mathbb{E}_k[\xi]$ to denote an expectation of $\xi$ w.r.t. the randomness coming from step $k$.

**Robust aggregator.** We follow the definition from (Gorbunov et al., 2023) of $(\delta, c)$-robust aggregation, which is a generalization of the definitions proposed by Karimireddy et al. (2021, 2022).

**Definition 2.1** ($(\delta, c)$-Robust Aggregator). Assume that $\{x_1, x_2, \ldots, x_n\}$ is such that there exists a subset $\mathcal{G} \subseteq [n]$ of size $|\mathcal{G}| = G \geq (1 - \delta)n$ for $\delta \leq \delta_{\max} < 0.5$ and there exists $\sigma \geq 0$ such that $\frac{1}{G(G-1)} \sum_{i,l \in \mathcal{G}} \mathbb{E}\left[\|x_i - x_l\|^2\right] \leq \sigma^2$ where the expectation is taken w.r.t. the randomness of $\{x_i\}_{i \in \mathcal{G}}$. We say that the quantity $\widehat{x}$ is $(\delta, c)$-Robust Aggregator $(\delta, c)$-RAgg and write $\widehat{x} = \text{RAgg}(x_1, \ldots, x_n)$ for some $c > 0$, if the following inequality holds:

$$\mathbb{E}\left[\|\widehat{x} - \bar{x}\|^2\right] \leq c\delta\sigma^2, \tag{2}$$

where $\bar{x} := \frac{1}{|\mathcal{G}|} \sum_{i \in \mathcal{G}} x_i$. If additionally $\widehat{x}$ is computed without the knowledge of $\sigma^2$, we say that $\widehat{x}$ is $(\delta, c)$-Agnostic Robust Aggregator $(\delta, c)$-ARAgg and write $\widehat{x} = \text{ARAgg}(x_1, \ldots, x_n)$.

One can interpret the definition as follows. Ideally, we would like to filter out all Byzantine workers and compute just an average $\bar{x}$ over the set of good clients. However, this is impossible in general since we do not know apriori who are Byzantine workers. Instead of this, it is natural to expect that the aggregation rule approximates the ideal average up in a certain sense, e.g., in terms of the expected squared distance to $\bar{x}$. As Karimireddy et al. (2021) formally show, in terms of such criterion ($\mathbb{E}[\|\widehat{x} - \bar{x}\|^2]$), the definition of $(\delta, c)$-RAgg cannot be improved (up to the numerical constant). Moreover,

---

[5]For simplicity, we assume that all regular workers have the same size of local datasets. Our analysis can be easily generalized to the case of different sizes of local datasets: this will affect only the value of $\mathcal{L}_\pm$ from Assumption D.3 for some sampling strategies.

standard aggregators such as Krum (Blanchard et al., 2017), geometric median, and coordinate-wise median do not satisfy Definition 2.1 (Karimireddy et al., 2021), though another popular standard aggregation rule called coordinate-wise trimmed mean (Yin et al., 2018) satisfies Definition 2.1 as shown by Allouah et al. (2023) through the more general definition of robust aggregation. To address this issue, Karimireddy et al. (2021) develop the aggregator called CenteredClip and prove that it fits the definition of $(\delta, c)$-RAgg. Karimireddy et al. (2022) propose a procedure called Bucketing that fixes Krum, geometric median, and coordinate-wise median, i.e., with Bucketing Krum, geometric, and coordinate-wise median become $(\delta, c)$-ARAgg, which is important for our algorithm since the variance of the vectors received from regular workers changes over time in our method. We notice here that $\delta$ is a part of the input that should satisfy $\delta_{\text{real}} \leq \delta \leq \delta_{\text{max}}$.

**Compression operators.** In our work, we use standard unbiased compression operators with relatively bounded variance (Khirirat et al., 2018; Horváth et al., 2023).

**Definition 2.2** (Unbiased compression). Stochastic mapping $\mathcal{Q} : \mathbb{R}^d \to \mathbb{R}^d$ is called unbiased compressor/compression operator if there exists $\omega \geq 0$ such that for any $x \in \mathbb{R}^d$ $\mathbb{E}[\mathcal{Q}(x)] = x$, $\mathbb{E}\left[\|\mathcal{Q}(x) - x\|^2\right] \leq \omega\|x\|^2$. For the given unbiased compressor $\mathcal{Q}(x)$, one can define the expected density[6] as $\zeta_{\mathcal{Q}} := \sup_{x \in \mathbb{R}^d} \mathbb{E}\left[\|\mathcal{Q}(x)\|_0\right]$, where $\|y\|_0$ is the number of non-zero components of $y \in \mathbb{R}^d$.

In this definition, parameter $\omega$ reflects how lossy the compression operator is: the larger $\omega$ the more lossy the compression. For example, this class of compression operators includes random sparsification (RandK) (Stich et al., 2018) and quantization (Goodall, 1951; Roberts, 1962; Alistarh et al., 2017). For RandK compression $\omega = \frac{d}{K} - 1, \zeta_{\mathcal{Q}} = K$ and for $\ell_2$-quantization $\omega = \sqrt{d} - 1, \zeta_{\mathcal{Q}} = \sqrt{d}$, see the proofs in (Beznosikov et al., 2020).

**Assumptions.** Up to a couple of assumptions that are specific to our work, we use the same assumptions as in (Gorbunov et al., 2023). We start with two new assumptions.

**Assumption 2.3** (Bounded ARAgg). We assume that the server applies aggregation rule $\mathcal{A}$ such that $\mathcal{A}$ is $(\delta, c)$-ARAgg and there exists constant $F_{\mathcal{A}} > 0$ such that for any inputs $x_1, \ldots, x_n \in \mathbb{R}^d$ the norm of the aggregator is not greater than the maximal norm of the inputs: $\|\mathcal{A}(x_1, \ldots, x_n)\| \leq F_{\mathcal{A}} \max_{i \in [n]} \|x_i\|$.

The above assumption is satisfied for popular $(\delta, c)$-robust aggregation rules presented in the literature (Karimireddy et al., 2021, 2022). Therefore, this assumption is more a formality than a real limitation: it is needed to exclude some pathological examples of $(\delta, c)$-robust aggregation rules, e.g., for any $\mathcal{A}$ that is $(\delta, c)$-RAgg one can construct unbounded $(\delta, 2c)$-RAgg as $\overline{\mathcal{A}} = \mathcal{A} + X$, where $X$ is a random sample from the Gaussian distribution $\mathcal{N}(0, c\delta\sigma^2)$.

Next, for part of our results, we also make the following assumption.

**Assumption 2.4** (Bounded compressor (optional)). We assume that workers use compression operator $\mathcal{Q}$ satisfying Definition 2.2 and bounded as follows: $\|\mathcal{Q}(x)\| \leq D_Q\|x\| \quad \forall x \in \mathbb{R}^d$.

For example, RandK and $\ell_2$-quantization meet this assumption with $D_Q = \frac{d}{K}$ and $D_Q = \sqrt{d}$ respectively. In general, constant $D_Q$ can be large (proportional to $d$). However, in practice, one can use RandK with $K = \frac{d}{100}$ and, thus, have moderate $D_Q = 100$. We also have the results without Assumption 2.4, but with worse dependence on some other parameters, see Section 4.

Next, we assume that good workers have $\zeta^2$-heterogeneous local loss functions.

**Assumption 2.5** ($\zeta^2$-heterogeneity). We assume that good clients have $\zeta^2$-heterogeneous local loss functions for some $\zeta \geq 0$, i.e., $\frac{1}{G} \sum_{i \in \mathcal{G}} \|\nabla f_i(x) - \nabla f(x)\|^2 \leq \zeta^2 \quad \forall x \in \mathbb{R}^d$.

The above assumption is quite standard for the literature on Byzantine robustness (Wu et al., 2020; Karimireddy et al., 2022; Gorbunov et al., 2023; Allouah et al., 2023). Moreover, some kind of a bound on the heterogeneity of good clients is necessary since otherwise Byzantine robustness cannot be achieved in general. In the appendix, all proofs are given under a more general version of

---

[6]This quantity is well-suited for sparsification-type compression operators like random sparsification (Stich et al., 2018) and 1-level $\ell_2$-quantization (Alistarh et al., 2017). For other compressors, such as quantization with more than one level (Goodall, 1951; Roberts, 1962), $\zeta_{\mathcal{Q}}$ is not the main characteristic describing their properties.

---

**Algorithm 1** Byz-VR-MARINA-PP: Byzantine-tolerant VR-MARINA with Partial Participation

---

1: **Input:** vectors $x^0, g^0 \in \mathbb{R}^d$, stepsize $\gamma$, mini-batch size $b$, probability $p \in (0, 1]$, number of iterations $K$, $(\delta, c)$-`ARAgg`, clients' sample size $1 \leq C \leq \widehat{C} \leq n$, clipping coefficients $\{\alpha_k\}_{k \geq 1}$

2: **for** $k = 0, 1, \ldots, K - 1$ **do**

3:     Get a sample from Bernoulli distribution with parameter $p$: $c_k \sim \text{Be}(p)$

4:     Sample the set of clients $S_k \subseteq [n]$, $|S_k| = C$ if $c_k = 0$; otherwise $|S_k| = \widehat{C}$

5:     Broadcast $g^k, c_k$ to all workers

6:     **for** $i \in \mathcal{G} \cap S_k$ in parallel **do**

7:         $x^{k+1} = x^k - \gamma g^k$ and $\lambda_{k+1} = \alpha_{k+1} \|x^{k+1} - x^k\|$

8:         Set $g_i^{k+1} = \begin{cases} \nabla f_i(x^{k+1}), & \text{if } c_k = 1, \\ g^k + \texttt{clip}_{\lambda_{k+1}}\left(\mathcal{Q}\left(\widehat{\Delta}_i(x^{k+1}, x^k)\right)\right), & \text{otherwise}, \end{cases}$

        where $\widehat{\Delta}_i(x^{k+1}, x^k)$ is a mini-batched estimator of $\nabla f_i(x^{k+1}) - \nabla f_i(x^k)$, $\mathcal{Q}(\cdot)$ for $i \in \mathcal{G} \cap S_k$ are computed independently

9:     **end for**

10:    $g^{k+1} = \begin{cases} \texttt{ARAgg}\left(\{g_i^{k+1}\}_{i \in S_k}\right), & \text{if } c_k = 1, \\ g^k + \texttt{ARAgg}\left(\left\{\texttt{clip}_{\lambda_{k+1}}\left(\mathcal{Q}\left(\widehat{\Delta}_i(x^{k+1}, x^k)\right)\right)\right\}_{i \in S_k}\right), & \text{otherwise} \end{cases}$

11: **end for**

---

Assumption 2.5, see Assumption D.5. Finally, the case of homogeneous data ($\zeta = 0$) is also quite popular for collaborative learning (Diskin et al., 2021b; Kijsipongse et al., 2018a).

The following assumption is classical for the literature on non-convex optimization.

**Assumption 2.6** (Smoothness (simplified))**.** We assume that for all $i \in \mathcal{G}$ and $j \in [m]$ there exists $\mathcal{L} \geq 0$ such that $f_{i,j}$ is $\mathcal{L}$-smooth, i.e., for all $x, y \in \mathbb{R}^d$

$$\|\nabla f_{i,j}(x) - \nabla f_{i,j}(y)\| \leq \mathcal{L}\|x - y\|. \tag{3}$$

Moreover, we assume that $f$ is uniformly lower bounded by $f^* \in \mathbb{R}$, i.e., $f^* := \inf_{x \in \mathbb{R}^d} f(x)$.

For the sake of simplicity, we do not differentiate between various notions of smoothness in the main text. However, our analysis takes into account the differences between smoothness constants, similarity of local functions, and sampling strategy (see Appendix D.1).

Finally, we also consider functions satisfying Polyak-Łojasiewicz (PŁ) condition (Polyak, 1963; Łojasiewicz, 1963). This assumption belongs to the class of assumptions on the structured non-convexity that allows achieving linear convergence (Necoara et al., 2019).

**Assumption 2.7** (PŁ condition (optional))**.** We assume that function $f$ satisfies Polyak-Łojasiewicz (PŁ) condition with parameter $\mu > 0$, i.e., for all $x \in \mathbb{R}^d$ there exists $f^* := \inf_{x \in \mathbb{R}^d} f(x)$ such that $\|\nabla f(x)\|^2 \geq 2\mu \left(f(x) - f^*\right)$.

# 3   New Method: Byz-VR-MARINA-PP

We propose a new method called Byzantine-tolerant Variance-Reduced MARINA with Partial Participation (Byz-VR-MARINA-PP, Algorithm 1). Our method extends Byz-VR-MARINA (Gorbunov et al., 2023) to the partial participation case via the proper usage of the clipping operator. To illustrate how Byz-VR-MARINA-PP works, we first consider a special case of full participation.

**Special case: Byz-VR-MARINA.** If all clients participate at each round ($S_k \equiv [n]$) and clipping is turned off ($\lambda_k \equiv +\infty$), then Byz-VR-MARINA-PP reduces to Byz-VR-MARINA that works as follows. Consider the case when no compression is applied ($\mathcal{Q}(x) = x$) and $\widehat{\Delta}_i(x^{k+1}, x^k) = \nabla f_{i,j_k}(x^{k+1}) - \nabla f_{i,j_k}(x^k)$, where $j_k$ is sampled uniformly at random from $[m]$, $i \in \mathcal{G}$. Then, regular workers compute GeomSARAH/PAGE gradient estimator at each step: for $i \in \mathcal{G}$

$$g_i^{k+1} = \begin{cases} \nabla f_i(x^{k+1}), & \text{with probability } p, \\ g^k + \nabla f_{i,j_k}(x^{k+1}) - \nabla f_{i,j_k}(x^k), & \text{otherwise} \end{cases}$$

With small probability $p$, good workers compute full gradients, and with larger probability $1 - p$ they update their estimator via adding stochastic gradient difference. To balance the oracle cost of these two cases, one can choose $p \sim 1/m$ (for $b$-size mini-batched estimator $- p \sim b/m$). Such estimators are known to be optimal for finding stationary points in the stochastic first-order optimization (Fang et al., 2018; Arjevani et al., 2023). Next, good workers send $g_i^{k+1}$ or $\nabla f_{i,j_k}(x^{k+1}) - \nabla f_{i,j_k}(x^k)$ to the server who robustly aggregate the received vectors. Since estimators are conditionally biased, i.e., $\mathbb{E}[g_i^{k+1} \mid x^{k+1}, x^k] \neq \nabla f_i(x^{k+1})$, the additional bias coming from the aggregation does not cause significant issues in the analysis or practice. Moreover, the variance of $\{g_i^{k+1}\}_{i \in \mathcal{G}}$ w.r.t. the sampling of the stochastic gradients is proportional to $\|x^{k+1} - x^k\|^2 \to 0$ with probability $1 - p$ (due to Assumption D.3) that progressively limits the effect of Byzantine attacks. For a more detailed explanation of why recursive variance reduction works better than SAGA/SVRG-type variance reduction, we refer to (Gorbunov et al., 2023). Arbitrary sampling allows the improvement of the dependence on the smoothness constants. Unbiased communication compression also naturally fits the framework since it is applied to the stochastic gradient difference, meaning that the variance of $\{g_i^{k+1}\}_{i \in \mathcal{G}}$ w.r.t. the sampling of the stochastic gradients and compression remains proportional to $\|x^{k+1} - x^k\|^2$ with probability $1 - p$.

**New ingredients: client sampling and clipping.** The algorithmic novelty of Byz-VR-MARINA-PP in comparison to Byz-VR-MARINA is twofold: with (typically large) probability $1 - p$ only $C$ clients sampled uniformly at random from the set of all clients participate at each round, and clipping is applied to the compressed stochastic gradient differences. With a small probability $p$, a larger number[7] of clients $\widehat{C} \leq n$ takes part in the communication. The main role of clipping is to ensure that the method can withstand the attacks of Byzantines when they form a majority or, more precisely when there are more than $\delta C$ Byzantine workers among the sampled ones. *Indeed, without clipping (or some other algorithmic changes) such situations are critical for convergence: Byzantine workers can shift the method arbitrarily far from the solution, e.g., they can collectively send some vector with the arbitrarily large norm.* In contrast, Byz-VR-MARINA-PP tolerates any attacks even when all sampled clients are Byzantine workers since the update remains bounded due to the clipping. Via choosing $\lambda_{k+1} \sim \|x^{k+1} - x^k\|$ we ensure that the norm of transmitted vectors decreases with the same rate as it does in Byz-VR-MARINA with full client participation. Finally, with probability $1 - p$ regular workers can transmit just compressed vectors and leave the clipping operation to the server since Byzantines can ignore clipping operation.

## 4 Convergence Results

We define $\mathcal{G}_C^k = \mathcal{G} \cap S_k$ and $G_C^k = |\mathcal{G}_C^k|$ and $\binom{n}{k} = \frac{n!}{k!(n-k)!}$ represents the binomial coefficient. We also use the following probabilities:

$$p_G := \text{Prob}\left\{G_C^k \geq (1 - \delta)C\right\} = \sum_{\lceil (1-\delta)C \rceil \leq t \leq C} \frac{\binom{G}{t}\binom{n-G}{C-t}}{\binom{n}{C}},$$

$$\mathcal{P}_{\mathcal{G}_C^k} := \text{Prob}\left\{i \in \mathcal{G}_C^k \mid G_C^k \geq (1 - \delta)C\right\} = \frac{C}{np_G} \cdot \sum_{\lceil (1-\delta)C \rceil \leq t \leq C} \frac{\binom{G-1}{t-1}\binom{n-G}{C-t}}{\binom{n-1}{C-1}}.$$

These probabilities naturally appear in the analysis and statements of the theorems. When $c_k = 0$, then server samples $C$ clients, and two situations can appear: either $G_C^k$ is at least $(1 - \delta)C$ meaning that the aggregator can ensure robustness according to Definition 2.1 or $G_C^k < (1 - \delta)C$. Probability $p_G$ is the probability of the first event, and the second event implies that the aggregation can be spoiled by Byzantine workers (but clipping bounds the "harm"). Finally, we use $\mathcal{P}_{\mathcal{G}_C^k}$ in the computation of some conditional expectations when the first event occurs. The mentioned probabilities can be easily computed for some special cases. For example, if $C = 1$, then $p_G = G/n$ and $\mathcal{P}_{\mathcal{G}_C^k} = 1/G$; if $C = 2$, then $p_G = G(G-1)/n(n-1)$ and $\mathcal{P}_{\mathcal{G}_C^k} = 2/G$; finally, if $C = n$, then $p_G = 1$ and $\mathcal{P}_{\mathcal{G}_C^k} = 1$.

The next theorem is our main convergence result for general unbiased compression operators.

---

[7]As one can see from our analysis, it is sufficient to take $\widehat{C} \geq \max\{1, \delta_{\text{real}}n/\delta\}$ similarly to (Data and Diggavi, 2021). However, in contrast to the approach from Data and Diggavi (2021), Byz-VR-MARINA-PP requires such communications only with small probability $p$.

**Theorem 4.1.** *Let Assumptions 2.3, 2.5, 2.6 hold, $\lambda_{k+1} = 2\mathcal{L}\left\|x^{k+1} - x^k\right\|$, and $\widehat{C} \geq \max\{1, \delta_{real}n/\delta\}$. Assume that $0 < \gamma \leq 1/\mathcal{L}(1+\sqrt{A})$, where constant $A$ is defined as*

$$A = \frac{32 p_G G \mathcal{P}_{\mathcal{G}_C^k}}{p^2(1-\delta)C} \left(30\omega + 11\right)\left(1 + 2c\delta\right) + \frac{16(1-p_G)(1 + 4F_{\mathcal{A}}^2)}{p^2}. \tag{4}$$

*Then for all $K \geq 0$ the iterates produced by* Byz-VR-MARINA-PP *(Algorithm 1) satisfy*

$$\mathbb{E}\left[\left\|\nabla f\left(\widehat{x}^K\right)\right\|^2\right] \leq \frac{2\Phi^0}{\gamma(K+1)} + \frac{4\widehat{D}\zeta^2}{p}, \tag{5}$$

*where $\widehat{D} = \frac{2\delta\mathcal{P}_{\mathcal{G}_{\widehat{C}}^k}}{1-\delta}\left(\frac{6cG}{\widehat{C}} + p\right) + \widetilde{D}$, where $\widetilde{D} = 0$ when $\widehat{C} = n$, $\widetilde{D} = \frac{\mathcal{P}_{\mathcal{G}_{\widehat{C}}^k}G}{(1-\delta)\widehat{C}}$ when $\widehat{C} = n$, and $\widehat{x}^K$ is chosen uniformly at random from $x^0, x^1, \ldots, x^K$, and $\Phi^0 = f\left(x^0\right) - f^* + \frac{2\gamma}{p}\left\|g^0 - \nabla f\left(x^0\right)\right\|^2$. If, in addition, Assumption 2.7 holds and $0 < \gamma \leq 1/\mathcal{L}(1+\sqrt{2A})$, then for all $K \geq 0$ the iterates produced by* Byz-VR-MARINA-PP *(Algorithm 1) with $\rho = \min\left\{\gamma\mu, \frac{p}{8}\right\}$ satisfy*

$$\mathbb{E}\left[f\left(x^K\right) - f\left(x^*\right)\right] \leq (1-\rho)^K \Phi^0 + \frac{4\widehat{D}\zeta^2\gamma}{p\rho}, \tag{6}$$

*where $\Phi^0 = f\left(x^0\right) - f^* + \frac{4\gamma}{p}\left\|g^0 - \nabla f\left(x^0\right)\right\|^2$.*

The above theorem establishes similar guarantees to the current SOTA ones obtained for Byz-VR-MARINA. That is, in the general non-convex case, we prove $\mathcal{O}(1/K)$ rate, which is optimal (Arjevani et al., 2023), and for PŁ-functions we derive linear convergence result to the neighborhood depending on the heterogeneity. The size of this neighborhood matches the one derived for Byz-VR-MARINA by Gorbunov et al. (2023). However, since our result is obtained considering the challenging scenario of partial participation of clients, the maximal theoretically allowed stepsize in our analysis of Byz-VR-MARINA-PP is smaller than the one from (Gorbunov et al., 2023).

In particular, the second term in the constant $A$ appears due to the partial participation, and the whole expression for $A$ is proportional to $1/p^2$. In contrast, a similar constant $A$ from the result for Byz-VR-MARINA is proportional to $1/p$, which can be noticeably smaller than $1/p^2$. Indeed, to make the expected number of clients participating in the communication round equal to $\mathcal{O}(C)$, to make the expected number of stochastic oracle calls equal to $\mathcal{O}(b)$, and to make the expected number of transmitted components for each worker taking part in the communication round equal $\mathcal{O}(\zeta_{\mathcal{Q}})$, parameter $p$ should be chosen as $p = \min\{C/n, b/m, \zeta_{\mathcal{Q}}/d\}$, where the latter term in the minimum often equals to $\Theta(1/(\omega+1))$ (Gorbunov et al., 2021). Therefore, in some scenarios, $p$ can be small.

Next, in the special case of full participation, we have $C = \widehat{C} = n$, $p_G = \mathcal{P}_{\mathcal{G}_C^k} = 1$, meaning that $A = \Theta((1+\omega)(1+c\delta)/p^2)$ for Byz-VR-MARINA-PP. In contrast, the corresponding constant for Byz-VR-MARINA is of the order $\Theta((1+\omega)/pn + (1+\omega)c\delta/p^2)$, which is strictly better than our bound. In this special case, we do not recover the result for Byz-VR-MARINA.

Such a complexity deterioration can be explained as follows: the presence of clipping introduces additional technical difficulties in the analysis, resulting in a reduced step size compared to Byz-VR-MARINA, even when $C = \widehat{C} = n$. To achieve a more favorable convergence rate, particularly in scenarios of complete participation, we also establish the results under Assumption 2.4.

**Theorem 4.2.** *Let Assumptions 2.3, 2.4, 2.5, 2.6 hold, $\lambda_{k+1} = D_Q\mathcal{L}\left\|x^{k+1} - x^k\right\|$, and $\widehat{C} \geq \max\{1, \delta_{real}n/\delta\}$. Assume that $0 < \gamma \leq 1/\mathcal{L}(1+\sqrt{A})$, where constant $A$ equals*

$$A = \frac{4p_G G \mathcal{P}_{\mathcal{G}_C^k}}{p(1-\delta)C}\left(\frac{3\omega + 2}{(1-\delta)C} + \frac{8(5\omega + 4)c\delta}{p}\right) + \frac{8(1-p_G)(2 + F_{\mathcal{A}}^2 D_Q^2)}{p^2}. \tag{7}$$

*Then for all $K \geq 0$ the iterates produced by* Byz-VR-MARINA-PP *(Algorithm 1) satisfy*

$$\mathbb{E}\left[\left\|\nabla f\left(\widehat{x}^K\right)\right\|^2\right] \leq \frac{2\Phi^0}{\gamma(K+1)} + \frac{2\widehat{D}\zeta^2}{p}, \tag{8}$$

*where $\widehat{D} = \frac{2\delta\mathcal{P}_{\mathcal{G}_{\widehat{C}}^k}}{1-\delta}\left(\frac{6cG}{\widehat{C}} + p\right) + \widetilde{D}$, where $\widetilde{D} = 0$ when $\widehat{C} = n$, $\widetilde{D} = \frac{\mathcal{P}_{\mathcal{G}_{\widehat{C}}^k}G}{(1-\delta)\widehat{C}}$ when $\widehat{C} = n$, and $\widehat{x}^K$ is chosen uniformly at random from $x^0, x^1, \ldots, x^K$, and $\Phi^0 = f\left(x^0\right) - f^* + \frac{\gamma}{p}\left\|g^0 - \nabla f\left(x^0\right)\right\|^2$.*

*If, in addition, Assumption 2.7 holds and $0 < \gamma \leq 1/\mathcal{L}(1+\sqrt{2A})$, then for all $K \geq 0$ the iterates produced by* Byz-VR-MARINA-PP *(Algorithm 1) satisfy with* $\rho = \min\left\{\gamma\mu, \frac{p}{4}\right\}$

$$\mathbb{E}\left[f\left(x^K\right) - f\left(x^*\right)\right] \leq (1-\rho)^K \Phi^0 + \frac{2\widehat{D}\zeta^2\gamma}{p\rho}, \tag{9}$$

*where* $\Phi^0 = f\left(x^0\right) - f^* + \frac{2\gamma}{p}\left\|g^0 - \nabla f\left(x^0\right)\right\|^2$.

With Assumption 2.4, vectors $\{\mathcal{Q}(\widehat{\Delta}_i(x^{k+1}, x^k))\}_{i \in \mathcal{G}_C^k}$ can be upper bounded by $D_Q\mathcal{L}\left\|x^{k+1} - x^k\right\|$. Using this fact, one can take the clipping level sufficiently large such that it is turned off for the regular workers. This allows us to simplify the proof and remove $1/p$ factor in front of the terms not proportional to $\delta$ or to $1 - p_G$ in the expression for $A$ that can make the stepsize larger. However, the second term in (7) can be larger than (4), since it depends on potentially large constant $D_Q$. Therefore, the rates of convergence from Theorems 4.1 and 4.2 cannot be compared directly. We also highlight that the clipping level from Theorem 4.2 is in general larger than the clipping level from Theorem 4.1 and, thus, it is expected that with full participation Theorem 4.2 gives better results than Theorem 4.1: the bias introduced due to the clipping becomes smaller with the increase of the clipping level. However, in the partial participation regime, the price for this is a decrease of the stepsize to compensate for the increased harm from Byzantine clients in situations when they form a majority. Further discussion of the technical challenges we overcame is deferred to Appendix E.3.

Nevertheless, in the case of full participation, we have $C = \widehat{C} = n$, $p_G = \mathcal{P}_{\mathcal{G}_C^k} = \mathcal{P}_{\mathcal{G}_{\widehat{C}}^k} = 1$, meaning that $A = \Theta((1+\omega)/pn + (1+\omega)c\delta/p^2)$ in Theorem 4.2. That is, in this case, we recover the result of Byz-VR-MARINA. More generally, if $p_G = 1$, which is equivalent to $C \geq \max\{1, \delta_{\text{real}}n/\delta\}$, then $\mathcal{P}_{\mathcal{G}_C^k} = \text{Prob}\{i \in \mathcal{G}_C^k\} = \min\{1, C/G\}$, $\mathcal{P}_{\mathcal{G}_{\widehat{C}}^k} = \text{Prob}\{i \in \mathcal{G}_{\widehat{C}}^k\} = \min\{1, \widehat{C}/G\}$ and we have $A = \Theta((1+\omega)/pC + (1+\omega)c\delta/p^2)$. Here, the first term in $A$ is $n/C$ worse than the corresponding term for Byz-VR-MARINA. However, the second term in $A$ matches the corresponding term for Byz-VR-MARINA. Moreover, this term is the main one if $c\delta \geq p/C$, which is typically the case since parameter $p$ is often small ($p = \min\{C/\widehat{C}, b/m, \zeta_{\mathcal{Q}}/d\}$). In such cases, Byz-VR-MARINA-PP has the same rate of convergence as Byz-VR-MARINA while utilizing, on average, just $\mathcal{O}(C)$ workers at each step in contrast to Byz-VR-MARINA that uses $n$ workers at each step. *That is, in some cases, partial participation is provably beneficial for* Byz-VR-MARINA-PP.

Byz-VR-MARINA+**: simplified version of** Byz-VR-MARINA. In Appendix F, we propose a simplified version of Byz-VR-MARINA called Byz-VR-MARINA+ (Algorithm 3). The only difference is related to Line 10 of the method: when $c_k = 0$, Byz-VR-MARINA+ computes just the average of $\left\{\texttt{clip}_{\lambda_{k+1}}\left(\mathcal{Q}\left(\widehat{\Delta}_i(x^{k+1}, x^k)\right)\right)\right\}_{i \in S_k}$ instead of robust aggregation, while keeps using ARAgg when $c_k = 1$. Of course, when $c_k = 0$ and at least one Byzantine worker is sampled, then the step can be useless, but the "harm" of this step is bounded due to the clipping. However, in certain regimes (e.g., when $C$ is small enough and the number of Byzantine workers is much smaller than the number of regular workers), the probability of sampling only regular workers is larger than sampling at least one Byzantine worker when $c_k = 0$, meaning that with high enough probability the resulting estimator has no additional bias coming from the robust aggregation. We formally analyze Byz-VR-MARINA+ and show that such a modification of the method leads to better theoretical results (especially when $C$ is small). In particular, in the settings of Theorem 4.2, we prove that Byz-VR-MARINA+ exhibits the same $\mathcal{O}(1/K)$ rate but converges to $\mathcal{O}(1/p)$ smaller neighborhood when $\widehat{C} = n$, i.e., the neighborhood term for Byz-VR-MARINA+ is optimal (Karimireddy et al., 2022; Allouah et al., 2024b). Moreover, our results for Byz-VR-MARINA+ allow larger stepsizes when $C$ is small enough. For further details and complete proofs, we refer to Appendix F.

**Extensions without full-batch gradient computations.** The proposed methods – Byz-VR-MARINA and Byz-VR-MARINA+ – have a common limitation related to the full-batch gradient computation with probability $p$. Although this probability is typically small, even one full-gradient computation can be very expensive for certain problems. To address this issue, we propose the modifications of Byz-VR-MARINA and Byz-VR-MARINA+ without full-batch gradient computations at all (see Algorithms 4 and 5 in Appendix G). That is, these modifications differ from Byz-VR-MARINA and Byz-VR-MARINA+ in Line 8 only: when $c_k = 1$, every good worker $i$ from $S_k$ computes and sends to the server $b'$-size mini-batched stochastic gradient estimator $\widetilde{\nabla}f_i(x^{k+1})$ of $\nabla f_i(x^{k+1})$.

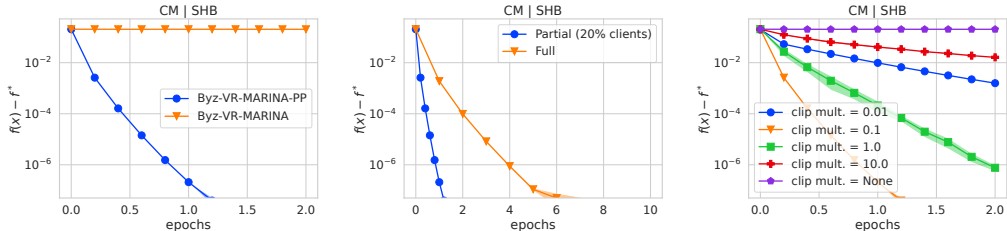

Figure 1: The optimality gap $f(x^k) - f(x^*)$ for 3 different scenarios. We use coordinate-wise mean with bucketing equal to 2 as an aggregation and shift-back as an attack. We use the a9a dataset, where each worker accesses the full dataset with 15 good and 5 Byzantine workers. We do not use any compression. In each step, we sample 20% of clients uniformly at random to participate in the given round unless we specifically mention that we use full participation. Left: Linear convergence of Byz-VR-MARINA-PP with clipping versus non-convergence without clipping. Middle: Full versus partial participation, showing faster convergence with clipping. Right: Clipping multiplier $\lambda$ sensitivity, demonstrating consistent linear convergence across varying $\lambda$ values.

Under the additional assumption that the variance of $\widetilde{\nabla} f_i(x^{k+1})$ is uniformly bounded by $\sigma^2/b'$, which is a standard assumption for variance-reduced methods without full-batch gradient computations (Fang et al., 2018; Cutkosky and Orabona, 2019; Li et al., 2021; Gorbunov et al., 2021), we prove that both methods converge similarly as in the case of the (periodical) full-batch gradient computations but to the neighborhood having an additional term proportional to $\left(c\delta + \mathcal{P}_{\mathcal{G}_{\widehat{C}}^k} G/\widehat{C}^2\right)\sigma^2/b'$. For further details and complete proofs, we refer to Appendix G.

**Heuristic extension of** Byz-VR-MARINA-PP. In this short remark, we illustrate how the proposed clipping technique can be applied to a general class of Byzantine-robust methods to adapt them to the case of partial participation. Consider the methods having the following update rule: $x^{k+1} = x^k - \gamma \cdot \text{Agg}(\{g_i^k\}_{i \in [n]})$, where $\{g_i^k\}_{i \in [n]}$ are the vectors received from workers at iteration $k$ and Agg is some aggregation rule. A vast majority of existing Byzantine-robust methods fit this scheme. In the case of partial participation of clients, we propose to modify the scheme as follows:

$$x^{k+1} = x^k - \gamma g^k, \quad \text{where } g^k := g^{k-1} + \text{Agg}\left(\left\{\text{clip}_{\lambda_k}(g_i^k - g^{k-1})\right\}_{i \in S_k}\right), \qquad (10)$$

where $S_k \subseteq [n]$ is a subset of clients participating in round $k$ and $\{\lambda_k\}_{k \geq 0}$ is sequence of clipping parameters specified by the server. In particular, Byz-VR-MARINA-PP can be seen as an application of scheme (10) to Byz-VR-MARINA (up to a minor modification when $c_k = 1$ in Byz-VR-MARINA) with $\lambda_{k+1} = \lambda \|x^{k+1} - x^k\|$. We suggest to use $\lambda_{k+1} = \lambda \|x^{k+1} - x^k\|$ with tunable parameter $\lambda > 0$ for other methods as well.

## 5 Numerical Experiments

Firstly, we showcase the benefits of employing clipping to remedy the presence of Byzantine workers and partial participation. For this task, we consider the standard logistic regression model with $\ell_2$-regularization, i.e., $f_{i,j}(x) = -y_{i,j} \log(h(x, a_{i,j})) - (1 - y_{i,j}) \log(1 - h(x, a_{i,j})) + \eta \|x\|^2$, where $y_{i,j} \in \{0, 1\}$ is the label, $a_{i,j} \in \mathbb{R}^d$ represents the feature vector, $\eta$ is the regularization parameter, and $h(x, a) = 1/(1+e^{-a^\top x})$. This objective is smooth, and for $\eta > 0$, it is also strongly convex, satisfying the PŁ-condition. We consider the *a9a* LIBSVM dataset (Chang and Lin, 2011) and set $\eta = 0.01$. In the experiments, we focus on an important feature of Byz-VR-MARINA-PP: it has linear convergence for homogeneous datasets across clients even in the presence of Byzantine workers and partial participation, as shown in Theorems 4.1 and 4.2.

To demonstrate this experimentally, we consider the setup with 15 good workers and 5 Byzantines, *each worker can access the entire dataset*, and the server uses coordinate-wise median with bucketing as the aggregator (see also Appendix C). For the attack, we propose a new attack that we refer to as the *shift-back* attack, which acts in the following way. If Byzantine workers are in the majority in the current round $k$, then each Byzantine worker sends $x^0 - x^k$. Otherwise, they follow protocol and act as benign workers. Further experimental details are deferred to Appendix H.

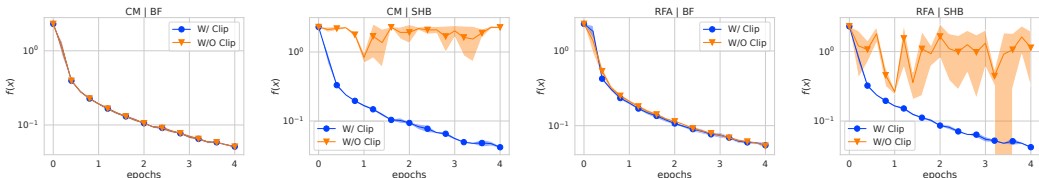

Figure 2: Training loss of 2 aggregation rules (CM, RFA) under 2 attacks (BF, SHB) on the MNIST dataset under heterogeneous data split with 20 clients, 5 of which are malicious. Additional experiments on CIFAR10 are provided in Appendix H.

We compare our Byz-VR-MARINA-PP with its version without clipping. We note that the setup that we consider is the most favorable in terms of minimized variance in terms of data and gradient heterogeneity. We show that even in this simplest setup, the method without clipping does not converge since there is no method that can withstand the Byzantine majority. Therefore, any more complex scenario would also fall short using our simple attack. On the other hand, we show that once clipping is applied, Byz-VR-MARINA-PP is able to converge linearly to the exact solution, complementing our theoretical results.

Figure 1 showcases these observations. On the left, we can see Byz-VR-MARINA-PP converges linearly to the optimal solution, while the version without clipping remains stuck at the starting point since Byzantines are always able to push the solution back to the origin since they can create the majority in some rounds. In the middle plot, we compare the full participation scenario in which all the clients participate in each round, which does not require clipping since, in each step, we are guaranteed that Byzantines are not in the majority, to partial participation with clipping. We can see, when we compare the total number of computations (measured in epochs), Byz-VR-MARINA-PP leads to faster convergence even though we need to employ clipping. Finally, in the right plot, we measure the sensitivity of clipping multiplier $\lambda$. We can see that Byz-VR-MARINA-PP is not very sensitive to $\lambda$ in terms of convergence, i.e., for all the values of $\lambda$, we still converge linearly. However, the suboptimal choice of $\lambda$ leads to slower convergence.

Furthermore, we also realize that other attacks and more complicated experiments could potentially damage clipping more than methods not using clipping. Therefore, we provide additional experiments with neural networks and different attacks in heterogeneous settings. For our experimental setup, we follow (Karimireddy et al., 2021). However, when working with neural networks, the choice of standard variance reduction is not effective (Defazio and Bottou, 2019). Therefore, we use Byzantine Robust Momentum SGD (Karimireddy et al., 2021) as an underlying optimization method; see (10).

We consider the MNIST dataset (LeCun and Cortes, 1998) with heterogeneous splits with 20 clients, 5 of which are malicious. For the attacks, we consider A Little is Enough (ALIE) (Baruch et al., 2019), Bit Flipping (BF), and aforementioned Shift-Back (SHB). For the aggregations, we consider coordinate median (CM) (Chen et al., 2017) and robust federated averaging (RFA) (Pillutla et al., 2022) with bucketing.

From Figure 2, we can see that clipping does not lead to performance degradation. On the contrary, clipping performs on par or better than its variant without clipping. Furthermore, we can see that no robust aggregator is able to withstand the shift-back attack without clipping.

## 6 Conclusion and Future Work

This work makes an important step in the direction of achieving Byzantine robustness under the partial participation of clients. However, some important questions remain open. First of all, it will be interesting to understand whether the derived bounds can be further improved in terms of the dependence on $\omega, m$, and $C$. Next, it would be interesting to rigorously prove that our heuristic works for SGD with client momentum (Karimireddy et al., 2021, 2022) and other Byzantine-robust methods. Finally, studying other participation patterns (non-uniform sampling/arbitrary client participation) is also a very prominent direction for future research.

## Acknowledgements

The work of G. Malinovsky and P. Richtárik was supported by funding from King Abdullah University of Science and Technology (KAUST): i) KAUST Baseline Research Scheme, ii) Center of Excellence for Generative AI, under award number 5940, iii) SDAIA-KAUST Center of Excellence in Artificial Intelligence and Data Science.

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

# Contents

# A  Extra Related Work

**Further Comparison with Data and Diggavi (2021).**   As we mention in the main text, Data and Diggavi (2021) assume that $3B$ is smaller than $C$. More precisely, Data and Diggavi (2021) assume that $B \leq \epsilon C$, where $\epsilon \leq \frac{1}{3} - \epsilon'$ for some parameter $\epsilon' > 0$ that will be explained later. That is, the results from Data and Diggavi (2021) do not hold when $C$ is smaller than $3B$, and, in particular, their algorithm cannot tolerate the situation when the server samples only Byzantine workers at some particular communication round. We also notice that when $C \geq 4B$, then existing methods such as Byz-VR-MARINA (Gorbunov et al., 2023) or Client Momentum (Karimireddy et al., 2021, 2022) can be applied without any changes to get a provable convergence.

Next, Data and Diggavi (2021) derive the upper bounds for the expected squared distance to the solution (in the strongly convex case) and the averaged expected squared norm of the gradient (in the non-convex case), where the expectation is taken w.r.t. the sampling of stochastic gradients only and the bounds itself hold with probability at least $1 - \frac{K}{H} \exp\left(-\frac{\epsilon'^2(1-\epsilon)C}{16}\right)$, where $H$ is the number of local steps. For simplicity consider the best-case scenario: $H = 1$ (local steps deteriorate the results from Data and Diggavi (2021)). Then, the lower bound for this probability becomes negative when either $C$ is not large enough or when $K$ is large or when $\epsilon$ is close to $\frac{1}{3}$, e.g., for $K = 10^6, \epsilon = \epsilon' = \frac{1}{6}, C = 5000$ this lower bound is smaller than $-720$, meaning that in this case, the result does not guarantee convergence. In contrast, our results have classical convergence criteria, where the expectations are taken w.r.t. the all randomness.

Finally, the bounds from Data and Diggavi (2021) have non-reduceable terms even for homogeneous data case: these terms are proportional to $\frac{\sigma^2}{b}$, where $\sigma^2$ is the upper bound for the variance of the stochastic estimator on regular clients and $b$ is the batchsize. In contrast, our results have only decreasing terms in the upper bounds when the data is homogeneous.

**Byzantine robustness.**   There exist various approaches to achieving Byzantine robustness (Lyu et al., 2020). Alistarh et al. (2018); Allen-Zhu et al. (2021) rely on the concentration inequalities for the stochastic gradients with bounded noise to iteratively remove them from the training. Karimireddy et al. (2021) formalize the definition of robust aggregation and propose the first provably robust aggregation rule called CenteredClip and the first provably Byzantine robust method under bounded variance assumption for homogeneous problems, i.e., when all good workers share one dataset. In particular, the method from (Karimireddy et al., 2021) uses client momentum on the clients that helps to memorize previous steps for good workers and withstand time-coupled attacks. This approach is extended by He et al. (2022) to the setup of decentralized learning. Allouah et al. (2023) develop an alternative definition for robust aggregation and propose a new aggregation rule satisfying their definition. Karimireddy et al. (2022) generalize these results to the heterogeneous data case and derive lower bounds for the optimization error that one can achieve in the heterogeneous case. Based on the formalism from Karimireddy et al. (2021), Gorbunov et al. (2022) propose a server-free approach that uses random checks of computations and bans of peers. This trick allows the elimination of all Byzantine workers after a finite number of steps on average. There are also many other approaches, e.g., one can use redundant computations of the stochastic gradients (Chen et al., 2018; Rajput et al., 2019) or introduce reputation metrics (Rodríguez-Barroso et al., 2020; Regatti et al., 2020; Xu and Lyu, 2020) to achieve some robustness, see also a recent survey by Lyu et al. (2020).

**Variance reduction.**   The literature on variance-reduced methods is very rich (Gower et al., 2020). The first variance-reduced methods are designed to fix the convergence of standard Stochastic Gradient Descent (SGD) and make it convergent to any predefined accuracy even with constant stepsizes. Such methods as SAG (Schmidt et al., 2017), SVRG (Johnson and Zhang, 2013), SAGA (Defazio et al., 2014) are developed mainly for (strongly) convex smooth optimization problems, while methods like SARAH (Nguyen et al., 2017), STORM (Cutkosky and Orabona, 2019), GeomSARAH (Horváth et al., 2023), PAGE (Li et al., 2021) are designed for general smooth non-convex problems. In this paper, we use GeomSARAH/PAGE-type variance reduction as the main building block of the method that makes the method robust to Byzantine attacks.

**Partial participation and client sampling.**   In the context of Byzantine-robust learning, there exists one work that develops and analyzes the method with partial participation (Data and Diggavi, 2021). However, this work relies on the restrictive assumption that the number of participating clients at each

round is at least three times larger than the number of Byzantine workers. In this case, Byzantines cannot form a majority, and standard methods can be applied without any changes. In contrast, our method converges in more challenging scenarios, e.g., Byz-VR-MARINA-PP provably converges even when the server samples one client, which can be Byzantine. The results from Data and Diggavi (2021) have some other noticeable limitations that we discuss in Appendix A.

**Communication compression.**   The literature on communication compression can be roughly divided into two huge groups. The first group studies the methods with unbiased communication compression. Different compression operators in the application to Distributed SGD/GD are studied in (Alistarh et al., 2017; Wen et al., 2017; Khirirat et al., 2018). To improve the convergence rate by fixing the error coming from the compression Mishchenko et al. (2019) propose to apply compression to the special gradient differences. Multiple extensions and generalizations of mentioned techniques are proposed and analyzed in the literature, e.g., see (Horváth et al., 2023; Gorbunov et al., 2021; Li et al., 2020; Qian et al., 2021; Basu et al., 2019; Haddadpour et al., 2021; Sadiev et al., 2022; Islamov et al., 2021; Safaryan et al., 2022).

Another large part of the literature on compressed communication is devoted to biased compression operators (Ajalloeian and Stich, 2020; Demidovich et al., 2023). Typically, such compression operators require more algorithmic changes than unbiased compressors since naïve combinations of biased compression with standard methods (e.g., Distributed GD) can diverge (Beznosikov et al., 2020). Error feedback is one of the most popular ways of utilizing biased compression operators in practice (Seide et al., 2014; Stich et al., 2018; Vogels et al., 2019), see also (Richtárik et al., 2021; Fatkhullin et al., 2021) for the modern version of error feedback with better theoretical guarantees for non-convex problems.

In the context of Byzantine robustness, methods with communication compression are also studied. The existing approaches are based on aggregation rules based on the norms of the updates (Ghosh et al., 2020, 2021), SignSGD and majority vote (Bernstein et al., 2019), SAGA-type variance reduction coupled with unbiased compression (Zhu and Ling, 2021), and GeomSARAH/PAGE-type variance reduction combined with unbiased compression (Gorbunov et al., 2023).

**Gradient clipping.**   Gradient clipping has multiple useful properties and applications. Originally it was used by Pascanu et al. (2013) to reduce the effect of exploding gradients during the training of RNNs. Gradient clipping is also a popular tool for achieving provable differential privacy (Abadi et al., 2016; Chen et al., 2020), convergence under generalized notions of smoothness (Zhang et al., 2020a; Mai and Johansson, 2021) and better (high-probability) convergence under heavy-tailed noise assumption (Zhang et al., 2020b; Nazin et al., 2019; Gorbunov et al., 2020; Sadiev et al., 2023; Nguyen et al., 2023). In the context of Byzantine-robust learning, gradient clipping is also utilized to design provably robust aggregation (Karimireddy et al., 2021). Our work proposes a novel useful application of clipping, i.e., we utilize clipping to achieve Byzantine robustness with partial participation of clients.

**Byzantine-robust asynchronous methods.**   Byzantine-robust asynchronous methods are also very relevant to the problem of partial participation in the Byzantine-robust learning. Indeed, the asynchronous methods like Asynchronous SGD (Agarwal and Duchi, 2011; Nedić et al., 2001) naturally have partial participation since whenever some worker finishes the computation (of the stochastic gradients), this worker immediately sends the update to the server and the server applies this update without waiting all other clients. However, without extra assumptions asynchronous methods cannot be tolerate Byzantine attacks: Byzantine clients could immediately send any vector to the server to guarantee that their update is received earlier than the updates from regular clients. Clearly, such a behavior of Byzantine workers leads to the divergence of the method unless the server has additional information that can be used for acceptance/rejection of the update or some other alternation of the communication protocol preventing the situations when some client updates the model too many times in a row is applied.

Therefore, the existing approaches addressing this important problem rely on extra assumptions. Damaskinos et al. (2018) propose to use Lipschitz filter and frequency filters in order to filter out Byzantine workers. Next, Xie et al. (2020b); Fang et al. (2022) use additional validation data on the server to decide whether to accept the update from workers. This assumption is restrictive for many FL applications when the data on clients is private and is not available on the server. Yang and Li

(2023) propose so-called BASGD (and its momentum version) where the key idea is to split workers into the buffers and wait until each buffer gets at least one gradient update. In the case when the number of buffers is sufficiently large (at least $2B$, where $B$ is the number of Byzantine workers), the authors show that BASGD converges. However, this means that to make the step BASGD requires to collect sufficiently large number of gradients such that the good buffers form majority, which is closer to full participation than to the partial participation in the worst case.

We emphasize that in our work we consider a different setup of synchronous communications with partial participation. The approaches discussed in the above paragraph cannot be directly applied to the problem considered in this paper without extra assumptions.

# B Useful Facts

For all $a, b \in \mathbb{R}^d$ and $\alpha > 0, p \in (0, 1]$ the following relations hold:

$$2\langle a, b \rangle = \|a\|^2 + \|b\|^2 - \|a - b\|^2 \tag{11}$$

$$\|a + b\|^2 \leq (1 + \alpha)\|a\|^2 + \left(1 + \alpha^{-1}\right)\|b\|^2 \tag{12}$$

$$-\|a - b\|^2 \leq -\frac{1}{1 + \alpha}\|a\|^2 + \frac{1}{\alpha}\|b\|^2, \tag{13}$$

$$(1 - p)\left(1 + \frac{p}{2}\right) \leq 1 - \frac{p}{2}, \quad p \geq 0 \tag{14}$$

$$(1 - p)\left(1 + \frac{p}{2}\right)\left(1 + \frac{p}{4}\right) \leq 1 - \frac{p}{4} \quad p \geq 0. \tag{15}$$

**Lemma B.1.** *(Lemma 5 from (Richtárik et al., 2021)). Let $a, b > 0$. If $0 \leq \gamma \leq \frac{1}{\sqrt{a}+b}$, then $a\gamma^2 + b\gamma \leq 1$. The bound is tight up to the factor of 2 since $\frac{1}{\sqrt{a}+b} \leq \min\left\{\frac{1}{\sqrt{a}}, \frac{1}{b}\right\} \leq \frac{2}{\sqrt{a}+b}$.*

# C   Justification of Assumption 2.3

---

**Algorithm 2** Bucketing Algorithm (Karimireddy et al., 2022)

---
1: **Input:** $\{x_1, \ldots, x_n\}$, $s \in \mathbb{N}$ – bucket size, $\texttt{Aggr}$ – aggregation rule
2: Sample random permutation $\pi = (\pi(1), \ldots, \pi(n))$ of $[n]$
3: Compute $y_i = \frac{1}{s} \sum_{k=s(i-1)+1}^{\min\{si, n\}} x_{\pi(k)}$ for $i = 1, \ldots, \lceil n/s \rceil$
4: **Return:** $\widehat{x} = \texttt{Aggr}(y_1, \ldots, y_{\lceil n/s \rceil})$

---

**Krum and Krum ∘ Bucketing.**   Krum aggregation rule is defined as

$$\mathrm{Krum}(x_1, \ldots, x_n) = \underset{x_i \in \{x_1, \ldots, x_n\}}{\mathrm{argmin}} \sum_{j \in S_i} \|x_j - x_i\|^2,$$

where $S_i \subset \{x_1, \ldots, x_n\}$ is the subset of $n - B - 2$ closest vectors to $x_i$. By definition, $\mathrm{Krum}(x_1, \ldots, x_n) \in \{x_1, \ldots, x_n\}$ and, thus $\|\mathrm{Krum}(x_1, \ldots, x_n)\| \leq \max_{i \in [n]} \|x_i\|$, i.e., Assumption 2.3 holds with $F_{\mathcal{A}} = 1$. Since Krum ∘ Bucketing applies Krum aggregation to averages $y_i$ over the buckets and $\|y_i\| \leq \frac{1}{s} \sum_{k=s(i-1)+1}^{\min\{si, n\}} \|x_{\pi(k)}\| \leq \max_{i \in [n]} \|x_i\|$, we have that $\|\mathrm{Krum} \circ \mathrm{Bucketing}(x_1, \ldots, x_n)\| \leq \max_{i \in [n]} \|x_i\|$.

**Geometric median (GM) and GM ∘ Bucketing.**   Geometric median is defined as follows:

$$\mathrm{GM}(x_1, \ldots, x_n) = \underset{x \in \mathbb{R}^d}{\mathrm{argmin}} \sum_{i=1}^{n} \|x - x_i\|. \tag{16}$$

One can show that $\mathrm{GM}(x_1, \ldots, x_n) \in \mathrm{Conv}(x_1, \ldots, x_n) := \{x \in \mathbb{R}^d \mid x = \sum_{i=1}^{n} \alpha_i x_i$ for some $\alpha_1, \ldots, \alpha_n \geq 1$ such that $\sum_{i=1}^{n} \alpha_i = 1\}$, i.e., geometric median belongs to the convex hull of the inputs. Indeed, let $\mathrm{GM}(x_1, \ldots, x_n) = x = \hat{x} + \tilde{x}$, where $\hat{x}$ is the projection of $x$ on $\mathrm{Conv}(x_1, \ldots, x_n)$ and $\tilde{x} = x - \hat{x}$. Then, the optimality condition implies that $\langle \hat{x} - x, y - \hat{x} \rangle \geq 0$ for all $y \in \mathrm{Conv}(x_1, \ldots, x_n)$. In particular, for all $i \in [n]$ we have $\langle \hat{x} - x, x_i - \hat{x} \rangle \geq 0$. Since

$$
\begin{aligned}
\langle \hat{x} - x, x_i - \hat{x} \rangle &= \langle \tilde{x}, \hat{x} - x_i \rangle = \frac{1}{2}\|\tilde{x} + \hat{x} - x_i\|^2 - \frac{1}{2}\|\tilde{x}\|^2 - \frac{1}{2}\|\hat{x} - x_i\|^2 \\
&= \frac{1}{2}\|x - x_i\|^2 - \frac{1}{2}\|\tilde{x}\|^2 - \frac{1}{2}\|\hat{x} - x_i\|^2 \\
&\leq \frac{1}{2}\|x - x_i\|^2 - \frac{1}{2}\|\hat{x} - x_i\|^2,
\end{aligned}
$$

we get that $\|x - x_i\| \geq \|\hat{x} - x_i\|$ for all $i \in [n]$ and the equality holds if and only if $\tilde{x} = 0$. Therefore, $\mathrm{argmin}$ from (16) is achieved for $x$ such that $x = \hat{x}$, meaning that $\mathrm{GM}(x_1, \ldots, x_n) \in \mathrm{Conv}(x_1, \ldots, x_n)$. Therefore, there exist some coefficients $\alpha_1, \ldots, \alpha_n \geq 0$ such that $\sum_{i=1}^{n} \alpha_i = 1$ and $\mathrm{GM}(x_1, \ldots, x_n) = \sum_{i=1}^{n} \alpha_i x_i$, implying that

$$\|\mathrm{GM}(x_1, \ldots, x_n)\| \leq \sum_{i=1}^{n} \alpha_i \|x_i\| \leq \max_{i \in [n]} \|x_i\|.$$

That is, GM satisfies Assumption 2.3 with $F_{\mathcal{A}} = 1$. Similarly to the case of Krum ∘ Bucketing, we also have $\|\mathrm{GM} \circ \mathrm{Bucketing}(x_1, \ldots, x_n)\| \leq \max_{i \in [n]} \|x_i\|$.

**Coordinate-wise median (CM) and CM ∘ Bucketing.**   Coordinate-wise median (CM) is formally defined as

$$\mathrm{CM}(x_1, \ldots, x_n) = \underset{x \in \mathbb{R}^d}{\mathrm{argmin}} \sum_{i=1}^{n} \|x - x_i\|_1, \tag{17}$$

where $\|\cdot\|_1$ denotes $\ell_1$-norm. This is equivalent to geometric median/median applied to vectors $x_1, \ldots, x_n$ component-wise. Therefore, from the above derivations for GM we have

$$
\begin{aligned}
\|\mathrm{CM}(x_1, \ldots, x_n)\|_\infty &\leq \max_{i \in [n]} \|x_i\|_\infty, \\
\|\mathrm{CM} \circ \mathrm{Bucketing}(x_1, \ldots, x_n)\|_\infty &\leq \max_{i \in [n]} \|x_i\|_\infty,
\end{aligned}
$$

where $\|\cdot\|_\infty$ denotes $\ell_\infty$-norm. Therefore, due to the standard relations between $\ell_2$- and $\ell_\infty$-norms, i.e., $\|a\|_\infty \leq \|a\| \leq \sqrt{d}\|a\|_\infty$ for any $a \in \mathbb{R}^d$, we have

$$
\begin{aligned}
\|\mathrm{CM}(x_1,\ldots,x_n)\| &\leq \sqrt{d}\max_{i\in[n]}\|x_i\|, \\
\|\mathrm{CM}\circ\mathrm{Bucketing}(x_1,\ldots,x_n)\| &\leq \sqrt{d}\max_{i\in[n]}\|x_i\|,
\end{aligned}
$$

i.e., Assumption 2.3 is satisfied with $F_{\mathcal{A}} = \sqrt{d}$.

# D  General Analysis

## D.1  Refined Assumptions

For simplicity, in the main part of our paper, we present simplified versions of our main results. However, our analysis works under more general assumptions presented in this section.

**Assumption on $\widehat{C}$.**   In all the results of this paper, we assume that $n \geq \widehat{C} \geq \max\{1, \delta_{\text{real}} n/\delta\}$. This condition ensures that the robust aggregation makes sense when $c_k = 1$, i.e., at least $1 - \delta$ proportion of sampled workers are not Byzantine ones when $c_k = 1$.

**Refined smoothness.**   The following assumption is classical for the literature on non-convex optimization.

**Assumption D.1** (*L*-smoothness).  We assume that function $f : \mathbb{R}^d \to \mathbb{R}$ is L-smooth, i.e., for all $x, y \in \mathbb{R}^d$ we have
$$\|\nabla f(x) - \nabla f(y)\| \leq L\|x - y\|. \tag{18}$$
Moreover, we assume that $f$ is uniformly lower bounded by $f^* \in \mathbb{R}$, i.e., $f^* := \inf_{x \in \mathbb{R}^d} f(x)$. In addition, we assume that $f_i$ is $L_i$-smooth for all $i \in \mathcal{G}$, i.e., for all $x, y \in \mathbb{R}^d$
$$\|\nabla f_i(x) - \nabla f_i(y)\| \leq L_i\|x - y\|. \tag{19}$$

We notice here that (19) implies *L*-smoothness of $f$ with $L \leq \frac{1}{G}\sum_{i \in \mathcal{G}} L_i$, i.e., smoothness constant of $f$ can be better than the averaged smoothness constant of the local loss functions on the regular clients.

Following Gorbunov et al. (2023), we consider refined assumptions on the smoothness.

**Assumption D.2** (Global Hessian variance assumption (Szlendak et al., 2022)).  We assume that there exists $L_\pm \geq 0$ such that for all $x, y \in \mathbb{R}^d$
$$\frac{1}{G}\sum_{i \in \mathcal{G}} \|\nabla f_i(x) - \nabla f_i(y)\|^2 - \|\nabla f(x) - \nabla f(y)\|^2 \leq L_\pm^2\|x - y\|^2. \tag{20}$$

We notice that (19) implies (20) with $L_\pm \leq \max_{i \in \mathcal{G}} L_i$. Szlendak et al. (2022) prove that $L_\pm$ satisfies the following relation: $L_{\text{avg}}^2 - L^2 \leq L_\pm^2 \leq L_{\text{avg}}^2$, where $L_{\text{avg}}^2 := \frac{1}{G}\sum_{i \in \mathcal{G}} L_i^2$. In particular, it is possible that $L_\pm = 0$ even if the data on the good workers is heterogeneous.

**Assumption D.3** (Local Hessian variance assumption (Gorbunov et al., 2023)).  We assume that there exists $\mathcal{L}_\pm \geq 0$ such that for all $x, y \in \mathbb{R}^d$
$$\frac{1}{G}\sum_{i \in \mathcal{G}} \mathbb{E}\left\|\widehat{\Delta}_i(x, y) - \Delta_i(x, y)\right\|^2 \leq \frac{\mathcal{L}_\pm^2}{b}\|x - y\|^2, \tag{21}$$

where $\Delta_i(x, y) := \nabla f_i(x) - \nabla f_i(y)$ and $\widehat{\Delta}_i(x, y)$ is an unbiased mini-batched estimator of $\Delta_i(x, y)$ with batch size $b$.

This assumption incorporates considerations for the smoothness characteristics inherent in all functions $\{f_{i,j}\}_{i \in \mathcal{G}, j \in [m]}$, the sampling policy, and the similarity among the functions $\{f_{i,j}\}_{i \in \mathcal{G}, j \in [m]}$. Gorbunov et al. (2023) have demonstrated that, assuming smoothness of $\{f_{i,j}\}_{i \in \mathcal{G}, j \in [m]}$, Assumption D.3 holds for various standard sampling strategies, including uniform and importance samplings.

For part of our results, we also need to assume smoothness of all $\{f_{i,j}\}_{i \in \mathcal{G}, j \in [m]}$ explicitly.

**Assumption D.4** (Smoothness of $f_{i,j}$ (optional)).  We assume that for all $i \in \mathcal{G}$ and $j \in [m]$ there exists $L_{i,j} \geq 0$ such that $f_{i,j}$ is $L_{i,j}$-smooth, i.e., for all $x, y \in \mathbb{R}^d$
$$\|\nabla f_{i,j}(x) - \nabla f_{i,j}(y)\| \leq L_{i,j}\|x - y\|. \tag{22}$$

**Refined heterogeneity.**   Instead of Assumption 2.5, we consider a more generalized one.

**Assumption D.5** $((B, \zeta^2)$-heterogeneity).  We assume that good clients have $(B, \zeta^2)$-heterogeneous local loss functions for some $B \geq 0, \zeta \geq 0$, i.e.,
$$\frac{1}{G}\sum_{i \in \mathcal{G}} \|\nabla f_i(x) - \nabla f(x)\|^2 \leq B\|\nabla f(x)\|^2 + \zeta^2 \quad \forall x \in \mathbb{R}^d$$

When $B = 0$, the above assumption recovers Assumption 2.5. However, it also covers some situations when the model is over-parameterized (Vaswani et al., 2019) and can hold with smaller values of $\zeta^2$. This assumption is also used in (Karimireddy et al., 2022; Gorbunov et al., 2023).

## D.2 Technical Lemmas

**Lemma D.6.** *Let $X$ be a random vector in $\mathbb{R}^d$ and $\widetilde{X} = \texttt{clip}_\lambda(X)$. Assume that $\mathbb{E}[X] = x \in \mathbb{R}^d$ and $\|x\| \leq \lambda/2$, then*

$$\mathbb{E}\left[\|\widetilde{X} - x\|^2\right] \leq 10\mathbb{E}\|X - x\|^2.$$

*Proof.* The proof follows a similar procedure to that presented in Lemma F.5 from (Gorbunov et al., 2020). To commence the proof, we introduce two indicator random variables:

$$\chi = \mathbb{I}_{\{X:\|X\|>\lambda\}} = \begin{cases} 1, & \text{if } \|X\| > \lambda, \\ 0, & \text{otherwise} \end{cases}, \eta = \mathbb{I}_{\left\{X:\|X-x\|>\frac{\lambda}{2}\right\}} = \begin{cases} 1, & \text{if } \|X - x\| > \frac{\lambda}{2} \\ 0, & \text{otherwise} \end{cases}.$$

Moreover, since $\|X\| \leq \|x\| + \|X - x\| \overset{\|x\|\leq\lambda/2}{\leq} \frac{\lambda}{2} + \|X - x\|$, we have $\chi \leq \eta$. Using that we get

$$\widetilde{X} = \min\left\{1, \frac{\lambda}{\|X\|}\right\} X = \chi \frac{\lambda}{\|X\|} X + (1 - \chi)X.$$

By Markov's inequality,

$$\mathbb{E}[\eta] = \mathbb{P}\left\{\|X - x\| > \frac{\lambda}{2}\right\} = \mathbb{P}\left\{\|X - x\|^2 > \frac{\lambda^2}{4}\right\} \leq \frac{4}{\lambda^2}\mathbb{E}\left[\|X - x\|^2\right]. \tag{23}$$

Using $\|\widetilde{X} - x\| \leq \|\widetilde{X}\| + \|x\| \leq \lambda + \frac{\lambda}{2} = \frac{3\lambda}{2}$, we obtain

$$\mathbb{E}\left[\|\widetilde{X} - x\|^2\right] = \mathbb{E}\left[\|\widetilde{X} - x\|^2\chi + \|\widetilde{X} - x\|^2(1 - \chi)\right]$$

$$= \mathbb{E}\left[\chi\left\|\frac{\lambda}{\|X\|}X - x\right\|^2 + \|X - x\|^2(1 - \chi)\right]$$

$$\leq \mathbb{E}\left[\chi\left(\left\|\frac{\lambda}{\|X\|}X\right\| + \|x\|\right)^2 + \|X - x\|^2(1 - \chi)\right]$$

$$\overset{\|x\|\leq\frac{\lambda}{2}}{\leq} \left(\mathbb{E}\left[\chi\left(\frac{3\lambda}{2}\right)^2 + \|X - x\|^2\right]\right),$$

where in the last inequality we applied $1 - \chi \leq 1$. Using (23) and $\chi \leq \eta$ we get

$$\mathbb{E}\left[\|\widetilde{X} - x\|^2\right] \leq \frac{9\lambda^2}{4}\left(\frac{2}{\lambda}\right)^2 \mathbb{E}\left[\|X - x\|^2\right] + \mathbb{E}\left[\|X - x\|^2\right]$$

$$\leq 10\mathbb{E}\left[\|X - x\|^2\right].$$

$\square$

**Lemma D.7** (Lemma 2 from Li et al. (2021))**.** *Assume that function $f$ is $L$-smooth (Assumption D.1) and $x^{k+1} = x^k - \gamma g^k$. Then*

$$f\left(x^{k+1}\right) \leq f\left(x^k\right) - \frac{\gamma}{2}\left\|\nabla f\left(x^k\right)\right\|^2 - \left(\frac{1}{2\gamma} - \frac{L}{2}\right)\left\|x^{k+1} - x^k\right\|^2 + \frac{\gamma}{2}\left\|g^k - \nabla f\left(x^k\right)\right\|^2.$$

**Lemma D.8.** *Let Assumptions D.1, D.2, D.3 hold and the Compression Operator satisfy Definition 2.2. Let us define "ideal" estimator:*

$$\overline{g}^{k+1} = \begin{cases} \frac{1}{G_C^k} \sum_{i\in\mathcal{G}_C^k} \nabla f_i(x^{k+1}), & c_n = 1, & [1] \\ g^k + \nabla f\left(x^{k+1}\right) - \nabla f\left(x^k\right), & c_n = 0 \text{ and } G_C^k < (1 - \delta)C, & [2] \\ g^k + \frac{1}{G_C^k} \sum_{i\in\mathcal{G}_C^k} \texttt{clip}_\lambda\left(\mathcal{Q}\left(\widehat{\Delta}_i\left(x^{k+1}, x^k\right)\right)\right), & c_n = 0 \text{ and } G_C^k \geq (1 - \delta)C. & [3] \end{cases}$$

*Then for all $k \geq 0$ the iterates produced by* Byz-VR-MARINA-PP *(Algorithm 1) satisfy*

$$A_1 = \mathbb{E}\left[\left\|\bar{g}^{k+1} - \nabla f\left(x^{k+1}\right)\right\|^2\right]$$

$$\leq (1-p)\left(1+\frac{p}{4}\right)\mathbb{E}\left[\left\|g^k - \nabla f(x^k)\right\|^2\right] + p\frac{\delta \cdot \mathcal{P}_{\mathcal{G}_{\widehat{C}}^k}}{(1-\delta)}\mathbb{E}\left[B\|\nabla f(x)\|^2 + \zeta^2\right]$$

$$+ (1-p)p_G\left(1+\frac{4}{p}\right)\frac{2 \cdot \mathcal{P}_{\mathcal{G}_C^k}\, n}{C}\left(10\omega L^2 + (10\omega+1)L_\pm^2 + \frac{10(\omega+1)\mathcal{L}_\pm^2}{b}\right)\mathbb{E}\left[\|x^{k+1}-x^k\|^2\right],$$

*where* $p_G = \mathrm{Prob}\left\{G_C^k \geq (1-\delta)C\right\}$ *and* $\mathcal{P}_{\mathcal{G}_C^k} = \mathrm{Prob}\left\{i \in \mathcal{G}_C^k \mid G_C^k \geq (1-\delta)\,C\right\}$.

*Proof.* Let us examine the expected value of the squared difference between ideal estimator and full gradient:

$$A_1 = \mathbb{E}\left[\left\|\bar{g}^{k+1} - \nabla f\left(x^{k+1}\right)\right\|^2\right]$$

$$= \mathbb{E}\left[\mathbb{E}_k\left[\left\|\bar{g}^{k+1} - \nabla f\left(x^{k+1}\right)\right\|^2\right]\right]$$

$$= (1-p)\,p_G\,\mathbb{E}\left[\mathbb{E}_k\left[\left\|g^k + \frac{1}{G_C^k}\sum_{i\in\mathcal{G}_C^k}\mathtt{clip}_\lambda\left(\mathcal{Q}\left(\widehat{\Delta}_i\left(x^{k+1},x^k\right)\right)\right) - \nabla f\left(x^{k+1}\right)\right\|^2\right]\mid [3]\right]$$

$$+ (1-p)(1-p_G)\mathbb{E}\left[\mathbb{E}_k\left[\left\|g^k - \nabla f(x^k)\right\|^2\right]\mid [2]\right] + p\mathbb{E}\left[\left\|\frac{1}{G_{\widehat{C}}^k}\sum_{i\in\mathcal{G}_{\widehat{C}}^k}\nabla f_i(x^{k+1}) - \nabla f(x^{k+1})\right\|^2\right].$$

Using (12) and $\nabla f\left(x^k\right) - \nabla f\left(x^k\right) = 0$ we obtain

$$B_1 = \mathbb{E}\left[\mathbb{E}_k\left[\left\|g^k + \frac{1}{G_C^k}\sum_{i\in\mathcal{G}_C^k}\mathtt{clip}_\lambda\left(\mathcal{Q}\left(\widehat{\Delta}_i\left(x^{k+1},x^k\right)\right)\right) - \nabla f\left(x^{k+1}\right)\right\|^2\right]\mid [3]\right]$$

$$= \mathbb{E}\left[\mathbb{E}_k\left[\left\|g^k + \frac{1}{G_C^k}\sum_{i\in\mathcal{G}_C^k}\mathtt{clip}_\lambda\left(\mathcal{Q}\left(\widehat{\Delta}_i\left(x^{k+1},x^k\right)\right)\right) - \nabla f\left(x^{k+1}\right) + \nabla f\left(x^k\right) - \nabla f\left(x^k\right)\right\|^2\right]\mid [3]\right]$$

$$\overset{(12)}{\leq} \left(1+\frac{p}{4}\right)\mathbb{E}\left[\left\|g^k - \nabla f\left(x^k\right)\right\|^2\right]$$

$$+ \left(1+\frac{4}{p}\right)\mathbb{E}\left[\mathbb{E}_k\left[\left\|\frac{1}{G_C^k}\sum_{i\in\mathcal{G}_C^k}\mathtt{clip}_\lambda\left(\mathcal{Q}\left(\widehat{\Delta}_i\left(x^{k+1},x^k\right)\right)\right) - \left(\nabla f(x^{k+1}) - \nabla f(x^k)\right)\right\|^2\right]\mid [3]\right]$$

$$= \left(1+\frac{p}{4}\right)\mathbb{E}\left[\left\|g^k - \nabla f(x^k)\right\|^2\right]$$

$$+ \left(1+\frac{4}{p}\right)\mathbb{E}\left[\mathbb{E}_k\left[\left\|\frac{1}{G_C^k}\sum_{i\in\mathcal{G}_C^k}\mathtt{clip}_\lambda\left(\mathcal{Q}\left(\widehat{\Delta}_i\left(x^{k+1},x^k\right)\right)\right) - \Delta\left(x^{k+1},x^k\right)\right\|^2\right]\mid [3]\right].$$

Let us consider the last part of the inequality:

$$B_1' = \mathbb{E}\left[\mathbb{E}_k\left[\left\|\frac{1}{G_C^k}\sum_{i\in\mathcal{G}_C^k}\mathtt{clip}_\lambda\left(\mathcal{Q}\left(\widehat{\Delta}_i\left(x^{k+1},x^k\right)\right)\right) - \Delta\left(x^{k+1},x^k\right)\right\|^2\right]\mid [3]\right]$$

$$= \mathbb{E}\left[\mathbb{E}_{S_k}\left[\mathbb{E}_k\left[\left\|\frac{1}{G_C^k}\sum_{i\in\mathcal{G}_C^k}\mathtt{clip}_\lambda\left(\mathcal{Q}\left(\widehat{\Delta}_i\left(x^{k+1},x^k\right)\right)\right) - \Delta\left(x^{k+1},x^k\right)\right\|^2\right]\mid [3]\right]\right].$$

Note that $G_C^k \geq (1-\delta)C$ in this case:

$$B_1' \leq \frac{1}{C(1-\delta)} \mathbb{E}\left[\mathbb{E}_{S_k}\left[\sum_{i \in \mathcal{G}_C^k} \mathbb{E}_k\left[\left\|\text{clip}_\lambda\left(\mathcal{Q}\left(\widehat{\Delta}_i\left(x^{k+1}, x^k\right)\right)\right) - \Delta\left(x^{k+1}, x^k\right)\right\|^2\right] \mid [3]\right]\right]$$

$$\leq \frac{1}{C(1-\delta)} \mathbb{E}\left[\sum_{i \in \mathcal{G}} \mathbb{E}_{S_k}\left[\mathcal{I}_{\mathcal{G}_C^k}\right] \mathbb{E}_k\left[\left\|\text{clip}_\lambda\left(\mathcal{Q}\left(\widehat{\Delta}_i\left(x^{k+1}, x^k\right)\right)\right) - \Delta\left(x^{k+1}, x^k\right)\right\|^2\right] \mid [3]\right]$$

$$= \frac{1}{C(1-\delta)} \mathbb{E}\left[\sum_{i \in \mathcal{G}} \mathcal{P}_{\mathcal{G}_C^k} \cdot \mathbb{E}_k\left[\left\|\text{clip}_\lambda\left(\mathcal{Q}\left(\widehat{\Delta}_i\left(x^{k+1}, x^k\right)\right)\right) - \Delta\left(x^{k+1}, x^k\right)\right\|^2\right] \mid [3]\right], \quad (24)$$

where $\mathcal{I}_{\mathcal{G}_C^k}$ is an indicator function for the event $\left\{i \in \mathcal{G}_C^k \mid G_C^k \geq (1-\delta)C\right\}$ and $\mathcal{P}_{\mathcal{G}_C^k} = \text{Prob}\left\{i \in \mathcal{G}_C^k \mid G_C^k \geq (1-\delta)C\right\}$ is probability of such event. Note that $\mathbb{E}_{S_k}\left[\mathcal{I}_{\mathcal{G}_C^k}\right] = \mathcal{P}_{\mathcal{G}_C^k}$. In case of uniform sampling of clients we have

$$\forall i \in \mathcal{G} \quad \mathcal{P}_{\mathcal{G}_C^k} = \text{Prob}\left\{i \in \mathcal{G}_C^k \mid G_C^k \geq (1-\delta)C\right\}$$

$$= \frac{C}{n p_G} \cdot \sum_{(1-\delta)C \leq t \leq C} \left(\binom{G}{t}\binom{n-G}{C-t}\left(\binom{n}{C}\right)^{-1}\right),$$

$$p_G = \sum_{(1-\delta)C \leq t \leq C} \left(\binom{G-1}{t-1}\binom{n-G}{C-t}\left(\binom{n-1}{C-1}\right)^{-1}\right)$$

Now we can continue with inequalities:

$$B_1' \leq \frac{\mathcal{P}_{\mathcal{G}_C^k}}{C(1-\delta)} \mathbb{E}\left[\sum_{i \in \mathcal{G}} \mathbb{E}_k\left[\left\|\text{clip}_\lambda\left(\mathcal{Q}\left(\widehat{\Delta}_i\left(x^{k+1}, x^k\right)\right)\right) - \Delta\left(x^{k+1}, x^k\right)\right\|^2\right] \mid [3]\right]$$

$$\leq \frac{\mathcal{P}_{\mathcal{G}_C^k}}{C(1-\delta)} \mathbb{E}\left[\sum_{i \in \mathcal{G}} \mathbb{E}_k\left[\mathbb{E}_\mathcal{Q}\left[\left\|\text{clip}_\lambda\left(\mathcal{Q}\left(\widehat{\Delta}_i\left(x^{k+1}, x^k\right)\right)\right) - \Delta\left(x^{k+1}, x^k\right)\right\|^2\right]\right] \mid [3]\right]$$

$$\overset{(12)}{\leq} \frac{\mathcal{P}_{\mathcal{G}_C^k}}{C(1-\delta)} \mathbb{E}\left[\sum_{i \in \mathcal{G}} 2\mathbb{E}_k\left[\mathbb{E}_\mathcal{Q}\left[\left\|\text{clip}_\lambda\left(\mathcal{Q}\left(\widehat{\Delta}_i\left(x^{k+1}, x^k\right)\right)\right) - \Delta_i\left(x^{k+1}, x^k\right)\right\|^2\right]\right] \mid [3]\right]$$

$$+ \frac{\mathcal{P}_{\mathcal{G}_C^k}}{C(1-\delta)} \mathbb{E}\left[\sum_{i \in \mathcal{G}} 2\mathbb{E}_k\left[\left\|\Delta_i\left(x^{k+1}, x^k\right) - \Delta\left(x^{k+1}, x^k\right)\right\|^2\right] \mid [3]\right].$$

Using Lemma D.6 we have

$$B_1' \overset{\text{Lemma D.6}}{\leq} \frac{\mathcal{P}_{\mathcal{G}_C^k}}{C(1-\delta)} \mathbb{E}\left[\sum_{i \in \mathcal{G}} 20\mathbb{E}_k\left[\mathbb{E}_\mathcal{Q}\left[\left\|\mathcal{Q}\left(\widehat{\Delta}_i\left(x^{k+1}, x^k\right)\right) - \Delta_i\left(x^{k+1}, x^k\right)\right\|^2\right]\right] \mid [3]\right]$$

$$+ \frac{\mathcal{P}_{\mathcal{G}_C^k}}{C(1-\delta)} \mathbb{E}\left[\sum_{i \in \mathcal{G}} 2\mathbb{E}_k\left[\left\|\Delta_i\left(x^{k+1}, x^k\right) - \Delta\left(x^{k+1}, x^k\right)\right\|^2\right] \mid [3]\right]$$

$$\leq \frac{20 \cdot \mathcal{P}_{\mathcal{G}_C^k}}{C(1-\delta)} \mathbb{E}\left[\sum_{i \in \mathcal{G}} \mathbb{E}_k\left[\mathbb{E}_\mathcal{Q}\left[\left\|\mathcal{Q}\left(\widehat{\Delta}_i\left(x^{k+1}, x^k\right)\right) - \Delta_i\left(x^{k+1}, x^k\right)\right\|^2\right]\right] \mid [3]\right]$$

$$+ \frac{2 \cdot \mathcal{P}_{\mathcal{G}_C^k}}{C(1-\delta)} \mathbb{E}\left[\sum_{i \in \mathcal{G}} \mathbb{E}_k\left[\left\|\Delta_i\left(x^{k+1}, x^k\right) - \Delta\left(x^{k+1}, x^k\right)\right\|^2\right] \mid [3]\right]$$

$$\leq \frac{20 \cdot \mathcal{P}_{\mathcal{G}_C^k}}{C(1-\delta)} \mathbb{E}\left[\sum_{i \in \mathcal{G}} \mathbb{E}_k\left[\mathbb{E}_\mathcal{Q}\left[\left\|\mathcal{Q}\left(\widehat{\Delta}_i\left(x^{k+1}, x^k\right)\right)\right\|^2\right]\right] - \sum_{i \in \mathcal{G}}\left\|\Delta_i\left(x^{k+1}, x^k\right)\right\|^2 \mid [3]\right]$$

$$+ \frac{2 \cdot \mathcal{P}_{\mathcal{G}_C^k}}{C(1-\delta)} \mathbb{E}\left[\sum_{i \in \mathcal{G}} \mathbb{E}_k\left[\left\|\Delta_i\left(x^{k+1}, x^k\right) - \Delta\left(x^{k+1}, x^k\right)\right\|^2\right] \mid [3]\right].$$

Applying Definition [2.2] of Unbiased Compressor we have

$$
B_1' \le \frac{20 \cdot \mathcal{P}_{\mathcal{G}_C^k}}{C(1-\delta)} \mathbb{E}\left[\sum_{i \in \mathcal{G}}(1+\omega)\mathbb{E}_k \left\|\widehat{\Delta}_i\left(x^{k+1}, x^k\right)\right\|^2 - \sum_{i \in \mathcal{G}}\left\|\Delta_i\left(x^{k+1}, x^k\right)\right\|^2 \mid [3]\right]
$$
$$
+ \frac{2 \cdot \mathcal{P}_{\mathcal{G}_C^k}}{C(1-\delta)} \mathbb{E}\left[\sum_{i \in \mathcal{G}}\left\|\Delta_i\left(x^{k+1}, x^k\right) - \Delta\left(x^{k+1}, x^k\right)\right\|^2 \mid [3]\right]
$$
$$
\le \frac{20 \cdot \mathcal{P}_{\mathcal{G}_C^k}}{C(1-\delta)} \mathbb{E}\left[\sum_{i \in \mathcal{G}}(1+\omega)\mathbb{E}_k \left\|\widehat{\Delta}_i\left(x^{k+1}, x^k\right) - \Delta_i\left(x^{k+1}, x^k\right)\right\|^2\right]
$$
$$
+ \frac{20 \cdot \mathcal{P}_{\mathcal{G}_C^k}}{C(1-\delta)} \mathbb{E}\left[\sum_{i \in \mathcal{G}}(1+\omega)\mathbb{E}_k \left\|\Delta_i\left(x^{k+1}, x^k\right)\right\|^2 - \sum_{i \in \mathcal{G}}\mathbb{E}_k \left\|\Delta_i\left(x^{k+1}, x^k\right)\right\|^2 \mid [3]\right]
$$
$$
+ \frac{2 \cdot \mathcal{P}_{\mathcal{G}_C^k}}{C(1-\delta)} \mathbb{E}\left[\sum_{i \in \mathcal{G}}\left\|\Delta_i\left(x^{k+1}, x^k\right) - \Delta\left(x^{k+1}, x^k\right)\right\|^2 \mid [3]\right].
$$

Now we combine terms and have

$$
B_1' \le \frac{20 \cdot \mathcal{P}_{\mathcal{G}_C^k}}{C(1-\delta)}(1+\omega)\mathbb{E}\left[\sum_{i \in \mathcal{G}}\mathbb{E}_k \left[\left\|\widehat{\Delta}_i\left(x^{k+1}, x^k\right) - \Delta_i\left(x^{k+1}, x^k\right)\right\|^2\right] \mid [3]\right]
$$
$$
+ \frac{20 \cdot \mathcal{P}_{\mathcal{G}_C^k}}{C(1-\delta)}\omega\mathbb{E}\left[\sum_{i \in \mathcal{G}}\left\|\Delta_i\left(x^{k+1}, x^k\right)\right\|^2 \mid [3]\right]
$$
$$
+ \frac{2 \cdot \mathcal{P}_{\mathcal{G}_C^k}}{C(1-\delta)}\mathbb{E}\left[\sum_{i \in \mathcal{G}}\left\|\Delta_i\left(x^{k+1}, x^k\right) - \Delta\left(x^{k+1}, x^k\right)\right\|^2 \mid [3]\right]
$$
$$
= \frac{20 \cdot \mathcal{P}_{\mathcal{G}_C^k}}{C(1-\delta)}(1+\omega)\mathbb{E}\left[\sum_{i \in \mathcal{G}}\mathbb{E}_k \left[\left\|\widehat{\Delta}_i\left(x^{k+1}, x^k\right) - \Delta_i\left(x^{k+1}, x^k\right)\right\|^2\right] \mid [3]\right]
$$
$$
+ \frac{20 \cdot \mathcal{P}_{\mathcal{G}_C^k}}{C(1-\delta)}\omega\mathbb{E}\left[\sum_{i \in \mathcal{G}}\left\|\Delta_i\left(x^{k+1}, x^k\right) - \Delta\left(x^{k+1}, x^k\right)\right\|^2 + \left\|\Delta\left(x^{k+1}, x^k\right)\right\|^2 \mid [3]\right]
$$
$$
+ \frac{2 \cdot \mathcal{P}_{\mathcal{G}_C^k}}{C(1-\delta)}\mathbb{E}\left[\sum_{i \in \mathcal{G}}\left\|\Delta_i\left(x^{k+1}, x^k\right) - \Delta\left(x^{k+1}, x^k\right)\right\|^2 \mid [3]\right].
$$

Rearranging terms leads to

$$
B_1' \le \frac{20 \cdot \mathcal{P}_{\mathcal{G}_C^k}}{C(1-\delta)}(1+\omega)\mathbb{E}\left[\sum_{i \in \mathcal{G}}\mathbb{E}_k \left[\left\|\widehat{\Delta}_i\left(x^{k+1}, x^k\right) - \Delta_i\left(x^{k+1}, x^k\right)\right\|^2\right] \mid [3]\right]
$$
$$
+ \frac{2 \cdot \mathcal{P}_{\mathcal{G}_C^k}}{C(1-\delta)}(10\omega+1)\mathbb{E}\left[\sum_{i \in \mathcal{G}}\left\|\Delta_i\left(x^{k+1}, x^k\right) - \Delta\left(x^{k+1}, x^k\right)\right\|^2 \mid [3]\right]
$$
$$
+ \frac{20 \cdot \mathcal{P}_{\mathcal{G}_C^k}}{C(1-\delta)}\omega\mathbb{E}\left[\sum_{i \in \mathcal{G}}\left\|\Delta\left(x^{k+1}, x^k\right)\right\|^2 \mid [3]\right].
$$

Now we apply Assumptions [D.1], [D.2], [D.3]:

$$
B_1' \le \frac{20 \cdot \mathcal{P}_{\mathcal{G}_C^k}}{C(1-\delta)}(1+\omega)\mathbb{E}\left[G\frac{\mathcal{L}_{\pm}^2}{b}\|x^{k+1} - x^k\|^2\right]
$$
$$
+ \frac{2 \cdot \mathcal{P}_{\mathcal{G}_C^k}}{C(1-\delta)}(10\omega+1)\mathbb{E}\left[GL_{\pm}^2\|x^{k+1} - x^k\|^2\right]
$$
$$
+ \frac{20 \cdot \mathcal{P}_{\mathcal{G}_C^k}}{C(1-\delta)}\omega\mathbb{E}\left[GL^2\left\|x^{k+1} - x^k\right\|^2\right].
$$

Finally, we have

$$B_1' \leq \frac{2 \cdot \mathcal{P}_{\mathcal{G}_C^k} \cdot G}{C(1-\delta)} \left( 10\omega L^2 + (10\omega + 1)L_\pm^2 + \frac{10(\omega + 1)\mathcal{L}_\pm^2}{b} \right) \mathbb{E}\left[ \|x^{k+1} - x^k\|^2 \right].$$

Let us plug obtained results:

$$
\begin{aligned}
B_1 &\leq \left( 1 + \frac{p}{4} \right) \mathbb{E}\left[ \|g^k - \nabla f(x^k)\|^2 \right] \\
&+ \left( 1 + \frac{4}{p} \right) \frac{2 \cdot \mathcal{P}_{\mathcal{G}_C^k} \cdot G}{C(1-\delta)} \left( 10\omega L^2 + (10\omega + 1)L_\pm^2 + \frac{10(\omega + 1)\mathcal{L}_\pm^2}{b} \right) \mathbb{E}\left[ \|x^{k+1} - x^k\|^2 \right].
\end{aligned}
$$

Let us consider the term $\mathbb{E}\left[ \left\| \frac{1}{G_{\widehat{C}}^k} \sum_{i \in \mathcal{G}_{\widehat{C}}^k} \nabla f_i(x^{k+1}) - \nabla f(x^{k+1}) \right\|^2 \right]$:

$$
\begin{aligned}
\mathbb{E}\left[ \left\| \frac{1}{G_{\widehat{C}}^k} \sum_{i \in \mathcal{G}_{\widehat{C}}^k} \nabla f_i(x^{k+1}) - \nabla f(x^{k+1}) \right\|^2 \right] &\leq \mathbb{E}\left[ \frac{1}{G_{\widehat{C}}^k} \sum_{i \in \mathcal{G}_{\widehat{C}}^k} \left\| \nabla f_i(x^{k+1}) - \nabla f(x^{k+1}) \right\|^2 \right] \\
&\leq \frac{1}{(1-\delta)\widehat{C}} \mathbb{E}\left[ \sum_{i \in \mathcal{G}_{\widehat{C}}^k} \left\| \nabla f_i(x^{k+1}) - \nabla f(x^{k+1}) \right\|^2 \right] \\
&= \frac{1}{(1-\delta)\widehat{C}} \mathbb{E}\left[ \sum_{i \in \mathcal{G}} \mathcal{I}_{\mathcal{G}_{\widehat{C}}^k} \left\| \nabla f_i(x^{k+1}) - \nabla f(x^{k+1}) \right\|^2 \right]
\end{aligned}
$$

Using definition of $\mathcal{P}_{\mathcal{G}_C^k}$ we get

$$
\begin{aligned}
\mathbb{E}\left[ \left\| \frac{1}{G_{\widehat{C}}^k} \sum_{i \in \mathcal{G}_{\widehat{C}}^k} \nabla f_i(x^{k+1}) - \nabla f(x^{k+1}) \right\|^2 \right] &\leq \frac{\mathcal{P}_{\mathcal{G}_{\widehat{C}}^k}}{(1-\delta)\widehat{C}} \mathbb{E}\left[ \sum_{i \in \mathcal{G}} \left\| \nabla f_i(x^{k+1}) - \nabla f(x^{k+1}) \right\|^2 \right] \\
&\leq \frac{G \cdot \mathcal{P}_{\mathcal{G}_{\widehat{C}}^k}}{(1-\delta)\widehat{C}G} \mathbb{E}\left[ \sum_{i \in \mathcal{G}} \left\| \nabla f_i(x^{k+1}) - \nabla f(x^{k+1}) \right\|^2 \right]
\end{aligned}
$$

Using Assumption D.5 we get

$$
\begin{aligned}
\mathbb{E}\left[ \left\| \frac{1}{G_{\widehat{C}}^k} \sum_{i \in \mathcal{G}_{\widehat{C}}^k} \nabla f_i(x^{k+1}) - \nabla f(x^{k+1}) \right\|^2 \right] &\leq \frac{G \cdot \mathcal{P}_{\mathcal{G}_{\widehat{C}}^k}}{(1-\delta)\widehat{C}} \mathbb{E}\left[ B\|\nabla f(x)\|^2 + \zeta^2 \right] \\
&\leq \frac{\delta_{\text{real}} n \cdot \mathcal{P}_{\mathcal{G}_{\widehat{C}}^k}}{(1-\delta)\frac{\delta_{\text{real}} n}{\delta}} \mathbb{E}\left[ B\|\nabla f(x)\|^2 + \zeta^2 \right] \\
&= \frac{\delta \cdot \mathcal{P}_{\mathcal{G}_{\widehat{C}}^k}}{(1-\delta)} \mathbb{E}\left[ B\|\nabla f(x)\|^2 + \zeta^2 \right] \qquad (25)
\end{aligned}
$$

Also, we have

$$A_1 = \mathbb{E}\left[\left\|\overline{g}^{k+1} - \nabla f(x^{k+1})\right\|^2\right]$$

$$\leq (1-p)p_G B_1 + (1-p)(1-p_G)\mathbb{E}\left[\left\|g^k - \nabla f(x^k)\right\|^2\right] + p\frac{\delta \cdot \mathcal{P}_{\mathcal{G}^k_{\widehat{C}}}}{(1-\delta)}\mathbb{E}\left[B\|\nabla f(x)\|^2 + \zeta^2\right]$$

$$\leq (1-p)p_G\left(1+\frac{p}{4}\right)\mathbb{E}\left[\left\|g^k - \nabla f(x^k)\right\|^2\right]$$

$$+ (1-p)p_G\left(1+\frac{4}{p}\right)\frac{2\cdot\mathcal{P}_{\mathcal{G}^k_C}\cdot G}{C(1-\delta)}\left(10\omega L^2 + (10\omega+1)L^2_{\pm} + \frac{10(\omega+1)\mathcal{L}^2_{\pm}}{b}\right)\mathbb{E}\left[\|x^{k+1}-x^k\|^2\right]$$

$$+ (1-p)(1-p_G)\mathbb{E}\left[\left\|g^k - \nabla f(x^k)\right\|^2\right] + p\frac{\delta \cdot \mathcal{P}_{\mathcal{G}^k_{\widehat{C}}}}{(1-\delta)}\mathbb{E}\left[B\|\nabla f(x)\|^2 + \zeta^2\right].$$

To simplify the bound we use $\left(1+\frac{p}{4} > 1\right)$ and obtain

$$A_1 \leq (1-p)p_G\left(1+\frac{p}{4}\right)\mathbb{E}\left[\left\|g^k - \nabla f(x^k)\right\|^2\right] + p\frac{\delta \cdot \mathcal{P}_{\mathcal{G}^k_{\widehat{C}}}}{(1-\delta)}\mathbb{E}\left[B\|\nabla f(x)\|^2 + \zeta^2\right]$$

$$+ (1-p)p_G\left(1+\frac{4}{p}\right)\frac{2\cdot\mathcal{P}_{\mathcal{G}^k_C}\cdot G}{C(1-\delta)}\left(10\omega L^2 + (10\omega+1)L^2_{\pm} + \frac{10(\omega+1)\mathcal{L}^2_{\pm}}{b}\right)\mathbb{E}\left[\|x^{k+1}-x^k\|^2\right]$$

$$+ (1-p)(1-p_G)\mathbb{E}\left[\left\|g^k - \nabla f(x^k)\right\|^2\right]$$

$$\leq (1-p)p_G\left(1+\frac{p}{4}\right)\mathbb{E}\left[\left\|g^k - \nabla f(x^k)\right\|^2\right]$$

$$+ (1-p)p_G\left(1+\frac{4}{p}\right)\frac{2\cdot\mathcal{P}_{\mathcal{G}^k_C}\cdot G}{C(1-\delta)}\left(10\omega L^2 + (10\omega+1)L^2_{\pm} + \frac{10(\omega+1)\mathcal{L}^2_{\pm}}{b}\right)\mathbb{E}\left[\|x^{k+1}-x^k\|^2\right]$$

$$+ (1-p)(1-p_G)\left(1+\frac{p}{4}\right)\mathbb{E}\left[\left\|g^k - \nabla f(x^k)\right\|^2\right] + p\frac{\delta \cdot \mathcal{P}_{\mathcal{G}^k_{\widehat{C}}}}{(1-\delta)}\mathbb{E}\left[B\|\nabla f(x)\|^2 + \zeta^2\right]$$

$$\leq (1-p)\left(1+\frac{p}{4}\right)\mathbb{E}\left[\left\|g^k - \nabla f(x^k)\right\|^2\right] + p\frac{\delta \cdot \mathcal{P}_{\mathcal{G}^k_{\widehat{C}}}}{(1-\delta)}\mathbb{E}\left[B\|\nabla f(x)\|^2 + \zeta^2\right]$$

$$+ (1-p)p_G\left(1+\frac{4}{p}\right)\frac{2\cdot\mathcal{P}_{\mathcal{G}^k_C}n}{C}\left(10\omega L^2 + (10\omega+1)L^2_{\pm} + \frac{10(\omega+1)\mathcal{L}^2_{\pm}}{b}\right)\mathbb{E}\left[\|x^{k+1}-x^k\|^2\right].$$

$\square$

**Lemma D.9.** *Let us define "ideal" estimator:*

$$\overline{g}^{k+1} = \begin{cases} \frac{1}{G^k_C}\sum\limits_{i\in\mathcal{G}^k_C}\nabla f_i(x^{k+1}), & c_n = 1, & [1] \\ g^k + \nabla f\left(x^{k+1}\right) - \nabla f\left(x^k\right), & c_n = 0 \text{ and } G^k_C < (1-\delta)C, & [2] \\ g^k + \frac{1}{G^k_C}\sum\limits_{i\in\mathcal{G}^k_C}clip_\lambda\left(\mathcal{Q}\left(\widehat{\Delta}_i\left(x^{k+1},x^k\right)\right)\right), & c_n = 0 \text{ and } G^k_C \geq (1-\delta)C. & [3] \end{cases}$$

*Also let us introduce the notation*

$$ARAgg^{k+1}_Q = ARAgg\left(clip_{\lambda_{k+1}}\left(\mathcal{Q}\left(\widehat{\Delta}_1(x^{k+1},x^k)\right)\right),\ldots,clip_{\lambda_{k+1}}\left(\mathcal{Q}\left(\widehat{\Delta}_C(x^{k+1},x^k)\right)\right)\right).$$

*Then for all $k \geq 0$ the iterates produced by* Byz-VR-MARINA-PP *(Algorithm 1) satisfy*

$$A_2 = \mathbb{E}\left[\left\|g^{k+1} - \overline{g}^{k+1}\right\|^2\right]$$

$$\leq p\mathbb{E}\left[\mathbb{E}_k\left[\left\|ARAgg\left(\{g^{k+1}_i\}_{i\in S_k}\right) - \nabla f(x^{k+1})\right\|^2\right] \mid [1]\right]$$

$$+ (1-p)p_G\mathbb{E}\left[\mathbb{E}_k\left[\left\|\frac{1}{G^k_C}\sum\limits_{i\in\mathcal{G}^k_C}clip_\lambda\left(\mathcal{Q}\left(\widehat{\Delta}_i\left(x^{k+1},x^k\right)\right)\right) - ARAgg^{k+1}_Q\right\|^2 \mid [3]\right]\right]$$

$$+ (1-p)(1-p_G)\mathbb{E}\left[\mathbb{E}_k\left[\left\|\nabla f(x^{k+1}) - \nabla f(x^k) - ARAgg^{k+1}_Q\right\|^2 \mid [2]\right]\right],$$

*where* $p_G = \text{Prob}\{G_C^k \geq (1-\delta)C\}$.

*Proof.* Using conditional expectations we have

$$A_2 = \mathbb{E}\left[\mathbb{E}_k\left[\left\|g^{k+1} - \overline{g}^{k+1}\right\|^2\right]\right]$$

$$= p\mathbb{E}\left[\mathbb{E}_k\left[\left\|\mathtt{ARAgg}\left(\{g_i^{k+1}\}_{i \in S_k}\right) - \nabla f(x^{k+1})\right\|^2\right] \mid [1]\right]$$

$$+ (1-p)p_G\mathbb{E}\left[\mathbb{E}_k\left[\left\|g^k + \frac{1}{G_C^k}\sum_{i \in \mathcal{G}_C^k}\mathtt{clip}_\lambda\left(\mathcal{Q}\left(\widehat{\Delta}_i\left(x^{k+1}, x^k\right)\right)\right) - \left(g^k + \mathtt{ARAgg}_Q^{k+1}\right)\right\|^2\right] \mid [3]\right]$$

$$+ (1-p)(1-p_G)\mathbb{E}\left[\mathbb{E}_k\left[\left\|g^k + \nabla f(x^{k+1}) - \nabla f(x^k) - \left(g^k + \mathtt{ARAgg}_Q^{k+1}\right)\right\|^2\right] \mid [2]\right].$$

After simplification, we get the following bound:

$$A_2 \leq p\mathbb{E}\left[\mathbb{E}_k\left[\left\|\mathtt{ARAgg}\left(\{g_i^{k+1}\}_{i \in S_k}\right) - \nabla f(x^{k+1})\right\|^2\right] \mid [1]\right]$$

$$+ (1-p)p_G\mathbb{E}\left[\mathbb{E}_k\left[\left\|\frac{1}{G_C^k}\sum_{i \in \mathcal{G}_C^k}\mathtt{clip}_\lambda\left(\mathcal{Q}\left(\widehat{\Delta}_i\left(x^{k+1}, x^k\right)\right)\right) - \mathtt{ARAgg}_Q^{k+1}\right\|^2 \mid [3]\right]\right]$$

$$+ (1-p)(1-p_G)\mathbb{E}\left[\mathbb{E}_k\left[\left\|\nabla f(x^{k+1}) - \nabla f(x^k) - \mathtt{ARAgg}_Q^{k+1}\right\|^2 \mid [2]\right]\right].$$

$\square$

**Lemma D.10.** *Let Assumptions D.1 and D.5 hold and Aggregation Operator (ARAgg) satisfy Definition 2.1. Then for all $k \geq 0$ the iterates produced by* Byz-VR-MARINA-PP *(Algorithm 1) satisfy*

$$T_1 = \mathbb{E}\left[\mathbb{E}_k\left[\left\|\mathit{ARAgg}\left(\{g_i^{k+1}\}_{i \in S_k}\right) - \nabla f(x^{k+1})\right\|^2\right] \mid [1]\right]$$

$$\leq \left(\frac{8G\mathcal{P}_{\mathcal{G}_{\widehat{C}}^k}c\delta B}{(1-\delta)\widehat{C}} + 2\widetilde{B}\right)\mathbb{E}\left[\left\|\nabla f\left(x^k\right)\right\|^2 + L^2\left\|x^{k+1} - x^k\right\|^2\right] + \frac{4G\mathcal{P}_{\mathcal{G}_{\widehat{C}}^k}c\delta\zeta^2}{(1-\delta)\widehat{C}} + \widetilde{\zeta}^2,$$

*where* $\widetilde{B} := 0$ *and* $\widetilde{\zeta}^2 := 0$ *when* $\widehat{C} = n$, *and* $\widetilde{B} := \frac{\mathcal{P}_{\mathcal{G}_{\widehat{C}}^k}GB}{(1-\delta)\widehat{C}}$ *and* $\widetilde{\zeta}^2 := \frac{\mathcal{P}_{\mathcal{G}_{\widehat{C}}^k}G\zeta^2}{(1-\delta)\widehat{C}}$ *when* $\widehat{C} < n$.

*Proof.* Using the definition of aggregation operator, we have

$$T_1 = \mathbb{E}\left[\mathbb{E}_k\left[\left\|\mathtt{ARAgg}\left(\{g_i^{k+1}\}_{i \in S_k}\right) - \nabla f(x^{k+1})\right\|^2\right] \mid [1]\right]$$

$$\overset{(12)}{\leq} \mathbb{E}\left[\mathbb{E}_k\left[\left\|\mathtt{ARAgg}\left(\{g_i^{k+1}\}_{i \in S_k}\right) - \frac{1}{G_{\widehat{C}}^k}\sum_{i \in \mathcal{G}_{\widehat{C}}^k}\nabla f_i(x^{k+1})\right\|^2\right] \mid [1]\right]$$

$$+ \mathbb{E}\left[\mathbb{E}_k\left[\left\|\frac{1}{G_{\widehat{C}}^k}\sum_{i \in \mathcal{G}_{\widehat{C}}^k}\nabla f_i(x^{k+1}) - \nabla f(x^{k+1})\right\|^2\right] \mid [1]\right].$$

Since $\frac{1}{G_{\widehat{C}}^k} \sum_{i\in\mathcal{G}_{\widehat{C}}^k} \nabla f_i(x^{k+1}) = \nabla f(x^{k+1})$ with probability 1 when $\widehat{C} = n$, we can estimate the last term as

$$\mathbb{E}\left[\mathbb{E}_k\left[\left\|\frac{1}{G_{\widehat{C}}^k}\sum_{i\in\mathcal{G}_{\widehat{C}}^k}\nabla f_i(x^{k+1}) - \nabla f(x^{k+1})\right\|^2\right]\,\Big|\,[1]\right]$$

$$\leq \begin{cases} 0, & \text{if } \widehat{C} = n \\ \mathbb{E}\left[\frac{1}{G_{\widehat{C}}^k}\sum_{i\in\mathcal{G}_{\widehat{C}}^k}\mathbb{E}_k\left[\left\|\nabla f_i(x^{k+1}) - \nabla f(x^{k+1})\right\|^2\right]\,\Big|\,[1]\right], & \text{if } \widehat{C} < n \end{cases}$$

$$\leq \begin{cases} 0, & \text{if } \widehat{C} = n \\ \frac{\mathcal{P}_{\mathcal{G}_{\widehat{C}}^k}}{(1-\delta)\widehat{C}}\sum_{i\in\mathcal{G}}\mathbb{E}\left[\left\|\nabla f_i(x^{k+1}) - \nabla f(x^{k+1})\right\|^2\right], & \text{if } \widehat{C} < n \end{cases}$$

$$\overset{(\text{As. D.5})}{\leq} \begin{cases} 0, & \text{if } \widehat{C} = n \\ \frac{\mathcal{P}_{\mathcal{G}_{\widehat{C}}^k}G}{(1-\delta)\widehat{C}}\left(B\mathbb{E}\left[\|\nabla f(x^{k+1})\|^2\right] + \zeta^2\right), & \text{if } \widehat{C} < n \end{cases} = \widetilde{B}\mathbb{E}\left[\|\nabla f(x^{k+1})\|^2\right] + \widetilde{\zeta}^2,$$

where

$$\widetilde{B} := \begin{cases} 0, & \text{if } \widehat{C} = n, \\ \frac{\mathcal{P}_{\mathcal{G}_{\widehat{C}}^k}GB}{(1-\delta)\widehat{C}}, & \text{if } \widehat{C} < n, \end{cases} \quad \text{and} \quad \widetilde{\zeta}^2 := \begin{cases} 0, & \text{if } \widehat{C} = n, \\ \frac{\mathcal{P}_{\mathcal{G}_{\widehat{C}}^k}G\zeta^2}{(1-\delta)\widehat{C}}, & \text{if } \widehat{C} < n. \end{cases}$$

Using the above bound, we continue the estimation of $T_1$ as follows:

$$
T_1 \overset{\text{(Def. 2.1)}}{\leq} \mathbb{E}\left[\frac{c\delta}{G_{\widehat{C}}^k(G_{\widehat{C}}^k-1)} \sum_{\substack{i,l\in\mathcal{G}_{\widehat{C}}^k \\ i\neq l}} \mathbb{E}_k\left[\left\|\nabla f_i\left(x^{k+1}\right)-\nabla f_l\left(x^{k+1}\right)\right\|^2 \mid [1]\right]\right]
$$
$$
+ \widetilde{B}\mathbb{E}\left[\|\nabla f(x^{k+1})\|^2\right] + \widetilde{\zeta}^2
$$

$$
\overset{(12)}{\leq} \mathbb{E}\left[\frac{c\delta}{G_{\widehat{C}}^k(G_{\widehat{C}}^k-1)} \sum_{\substack{i,l\in\mathcal{G}_{\widehat{C}}^k \\ i\neq l}} \mathbb{E}\left[2\left\|\nabla f_i\left(x^{k+1}\right)-\nabla f\left(x^{k+1}\right)\right\|^2 \mid [1]\right]\right]
$$

$$
+ \mathbb{E}\left[\frac{c\delta}{G_{\widehat{C}}^k(G_{\widehat{C}}^k-1)} \sum_{\substack{i,l\in\mathcal{G}_{\widehat{C}}^k \\ i\neq l}} \mathbb{E}\left[2\left\|\nabla f_l\left(x^{k+1}\right)-\nabla f\left(x^{k+1}\right)\right\|^2 \mid [1]\right]\right]
$$

$$
+ \widetilde{B}\mathbb{E}\left[\|\nabla f(x^{k+1})\|^2\right] + \widetilde{\zeta}^2
$$

$$
= \mathbb{E}\left[\frac{c\delta}{G_{\widehat{C}}^k} \sum_{i\in\mathcal{G}_{\widehat{C}}^k} 4\mathbb{E}_k\left[\left\|\nabla f_i\left(x^{k+1}\right)-\nabla f\left(x^{k+1}\right)\right\|^2 \mid [1]\right]\right] + \widetilde{B}\mathbb{E}\left[\|\nabla f(x^{k+1})\|^2\right] + \widetilde{\zeta}^2
$$

$$
\leq \frac{\mathcal{P}_{\mathcal{G}_{\widehat{C}}^k} c\delta}{(1-\delta)\widehat{C}} \sum_{i\in\mathcal{G}} 4\mathbb{E}_k\left[\left\|\nabla f_i\left(x^{k+1}\right)-\nabla f\left(x^{k+1}\right)\right\|^2\right] + \widetilde{B}\mathbb{E}\left[\|\nabla f(x^{k+1})\|^2\right] + \widetilde{\zeta}^2
$$

$$
\overset{\text{(As. D.5)}}{\leq} \left(\frac{4G\mathcal{P}_{\mathcal{G}_{\widehat{C}}^k} c\delta B}{(1-\delta)\widehat{C}} + \widetilde{B}\right) \mathbb{E}\left[\left\|\nabla f\left(x^{k+1}\right)\right\|^2\right] + \frac{4G\mathcal{P}_{\mathcal{G}_{\widehat{C}}^k} c\delta\zeta^2}{(1-\delta)\widehat{C}} + \widetilde{\zeta}^2
$$

$$
\overset{(12)}{\leq} \left(\frac{8G\mathcal{P}_{\mathcal{G}_{\widehat{C}}^k} c\delta B}{(1-\delta)\widehat{C}} + 2\widetilde{B}\right) \mathbb{E}\left[\left\|\nabla f\left(x^k\right)\right\|^2 + \left\|\nabla f\left(x^{k+1}\right)-\nabla f\left(x^k\right)\right\|^2\right]
$$

$$
+ \frac{4G\mathcal{P}_{\mathcal{G}_{\widehat{C}}^k} c\delta\zeta^2}{(1-\delta)\widehat{C}} + \widetilde{\zeta}^2
$$

$$
\leq \left(\frac{8G\mathcal{P}_{\mathcal{G}_{\widehat{C}}^k} c\delta B}{(1-\delta)\widehat{C}} + 2\widetilde{B}\right) \mathbb{E}\left[\left\|\nabla f\left(x^k\right)\right\|^2 + L^2\left\|x^{k+1}-x^k\right\|^2\right] + \frac{4G\mathcal{P}_{\mathcal{G}_{\widehat{C}}^k} c\delta\zeta^2}{(1-\delta)\widehat{C}} + \widetilde{\zeta}^2,
$$

which concludes the proof. $\qquad\square$

**Lemma D.11.** *Let Assumptions D.1, D.2, D.3 hold and the Compression Operator satisfy Definition 2.2. Also let us introduce the notation*

$$
\mathtt{ARAgg}_Q^{k+1} = \mathtt{ARAgg}\left(\mathtt{clip}_{\lambda_{k+1}}\left(\mathcal{Q}\left(\widehat{\Delta}_1(x^{k+1},x^k)\right)\right),\ldots,\mathtt{clip}_{\lambda_{k+1}}\left(\mathcal{Q}\left(\widehat{\Delta}_C(x^{k+1},x^k)\right)\right)\right).
$$

*Then for all $k \geq 0$ the iterates produced by* Byz-VR-MARINA-PP *(Algorithm 1) satisfy*

$$
T_2 = \mathbb{E}\left[\mathbb{E}_k\left[\left\|\frac{1}{G_C^k}\sum_{i\in\mathcal{G}_C^k}\mathtt{clip}_\lambda\left(\mathcal{Q}\left(\widehat{\Delta}_i\left(x^{k+1},x^k\right)\right)\right)-\mathtt{ARAgg}_Q^{k+1}\right\|^2 \mid [3]\right]\right]
$$

$$
\leq \frac{8G\mathcal{P}_{\mathcal{G}_C^k}}{(1-\delta)C}\left(10(1+\omega)\frac{\mathcal{L}_\pm^2}{b} + (10\omega+1)L_\pm^2 + 10\omega L^2\right)c\delta\mathbb{E}\left[\|x^{k+1}-x^k\|^2\right],
$$

*where $\mathcal{P}_{\mathcal{G}_C^k} = \text{Prob}\left\{i\in\mathcal{G}_C^k \mid G_C^k \geq (1-\delta)C\right\}$.*

*Proof.* By the definition of robust aggregation, we have

$$T_2 = \mathbb{E}\left[\mathbb{E}_k\left[\left\|\frac{1}{G_C^k}\sum_{i\in\mathcal{G}_C^k}\texttt{clip}_\lambda\left(\mathcal{Q}\left(\widehat{\Delta}_i\left(x^{k+1},x^k\right)\right)\right)-\texttt{ARAgg}_\mathcal{Q}^{k+1}\right\|^2\mid [3]\right]\right]$$

$$\leq \mathbb{E}\left[\frac{c\delta}{D_2}\sum_{\substack{i,l\in\mathcal{G}_C^k\\i\neq l}}\mathbb{E}_k\left[\left\|\texttt{clip}_\lambda\left(\mathcal{Q}\left(\widehat{\Delta}_i\left(x^{k+1},x^k\right)\right)\right)-\texttt{clip}_\lambda\left(\mathcal{Q}\left(\widehat{\Delta}_l\left(x^{k+1},x^k\right)\right)\right)\right\|^2\mid [3]\right]\right],$$

where $D_2 = G_C^k(G_C^k-1)$. Next, we consider pair-wise differences:

$$T_2'(i,l) = \mathbb{E}_k\left[\left\|\texttt{clip}_\lambda\left(\mathcal{Q}\left(\widehat{\Delta}_i\left(x^{k+1},x^k\right)\right)\right)-\texttt{clip}_\lambda\left(\mathcal{Q}\left(\widehat{\Delta}_l\left(x^{k+1},x^k\right)\right)\right)\right\|^2\mid [3]\right]$$

$$\overset{(12)}{\leq} 2\mathbb{E}_k\left[\left\|\texttt{clip}_\lambda\left(\mathcal{Q}\left(\widehat{\Delta}_i\left(x^{k+1},x^k\right)\right)\right)-\Delta_i\left(x^{k+1},x^k\right)+\Delta_l\left(x^{k+1},x^k\right)-\texttt{clip}_\lambda\left(\mathcal{Q}\left(\widehat{\Delta}_l\left(x^{k+1},x^k\right)\right)\right)\right\|^2\mid [3]\right]$$

$$+2\mathbb{E}_k\left[\left\|\Delta_i\left(x^{k+1},x^k\right)-\Delta_l\left(x^{k+1},x^k\right)\right\|^2\mid [3]\right]$$

$$\overset{(12)}{\leq} 4\mathbb{E}_k\left[\left\|\texttt{clip}_\lambda\left(\mathcal{Q}\left(\widehat{\Delta}_i\left(x^{k+1},x^k\right)\right)\right)-\Delta_i\left(x^{k+1},x^k\right)\right\|^2\mid [3]\right]$$

$$+4\mathbb{E}_k\left[\left\|\Delta_l\left(x^{k+1},x^k\right)-\texttt{clip}_\lambda\left(\mathcal{Q}\left(\widehat{\Delta}_l\left(x^{k+1},x^k\right)\right)\right)\right\|^2\mid [3]\right]$$

$$+2\mathbb{E}_k\left[\left\|\Delta_l\left(x^{k+1},x^k\right)-\Delta_i\left(x^{k+1},x^k\right)\right\|^2\mid [3]\right]]$$

$$\overset{(12)}{\leq} 4\mathbb{E}_k\left[\left\|\texttt{clip}_\lambda\left(\mathcal{Q}\left(\widehat{\Delta}_i\left(x^{k+1},x^k\right)\right)\right)-\Delta_i\left(x^{k+1},x^k\right)\right\|^2\mid [3]\right]$$

$$+4\mathbb{E}_k\left[\left\|\Delta_l\left(x^{k+1},x^k\right)-\texttt{clip}_\lambda\left(\mathcal{Q}\left(\widehat{\Delta}_l\left(x^{k+1},x^k\right)\right)\right)\right\|^2\mid [3]\right]$$

$$+4\mathbb{E}_k\left[\left\|\Delta_l\left(x^{k+1},x^k\right)-\Delta\left(x^{k+1},x^k\right)\right\|^2\mid [3]\right]$$

$$+4\mathbb{E}_k\left[\left\|\Delta_i\left(x^{k+1},x^k\right)-\Delta\left(x^{k+1},x^k\right)\right\|^2\mid [3]\right].$$

Now we can combine all parts together:

$$
\widehat{T}_2 = \mathbb{E}\left[\frac{1}{G_C^k(G_C^k-1)}\sum_{\substack{i,l\in\mathcal{G}_C^k\\i\neq l}}T_2'(i,l)\right]
$$

$$
\leq \mathbb{E}\left[\frac{1}{D_2}\sum_{\substack{i,l\in\mathcal{G}_C^k\\i\neq l}}4\mathbb{E}_k\left[\left\|\mathtt{clip}_\lambda\left(\mathcal{Q}\left(\widehat{\Delta}_i\left(x^{k+1},x^k\right)\right)\right)-\Delta_i\left(x^{k+1},x^k\right)\right\|^2\mid[3]\right]\right]
$$

$$
+\mathbb{E}\left[\frac{1}{D_2}\sum_{\substack{i,l\in\mathcal{G}_C^k\\i\neq l}}4\mathbb{E}_k\left[\left\|\Delta_l\left(x^{k+1},x^k\right)-\mathtt{clip}_\lambda\left(\mathcal{Q}\left(\widehat{\Delta}_l\left(x^{k+1},x^k\right)\right)\right)\right\|^2\mid[3]\right]\right]
$$

$$
+\mathbb{E}\left[\frac{1}{D_2}\sum_{\substack{i,l\in\mathcal{G}_C^k\\i\neq l}}4\mathbb{E}_k\left[\left\|\Delta_l\left(x^{k+1},x^k\right)-\Delta\left(x^{k+1},x^k\right)\right\|^2\mid[3]\right]\right]
$$

$$
+\mathbb{E}\left[\frac{1}{D_2}\sum_{\substack{i,l\in\mathcal{G}_C^k\\i\neq l}}4\mathbb{E}_k\left[\left\|\Delta_i\left(x^{k+1},x^k\right)-\Delta\left(x^{k+1},x^k\right)\right\|^2\mid[3]\right]\right].
$$

Rearranging the terms, we obtain

$$
\widehat{T}_2 \leq \mathbb{E}\left[\frac{1}{D_2}\sum_{\substack{i,l\in\mathcal{G}_C^k\\i\neq l}}8\mathbb{E}_k\left[\left\|\mathtt{clip}_\lambda\left(\mathcal{Q}\left(\widehat{\Delta}_i\left(x^{k+1},x^k\right)\right)\right)-\Delta_i\left(x^{k+1},x^k\right)\right\|^2\mid[3]\right]\right]
$$

$$
+\mathbb{E}\left[\frac{1}{D_2}\sum_{\substack{i,l\in\mathcal{G}_C^k\\i\neq l}}8\mathbb{E}_k\left[\left\|\Delta_i\left(x^{k+1},x^k\right)-\Delta\left(x^{k+1},x^k\right)\right\|^2\mid[3]\right]\right].
$$

It leads to

$$
\widehat{T}_2 \leq \mathbb{E}\left[\frac{1}{G_C^k}\sum_{i\in\mathcal{G}_C^k}8\mathbb{E}_k\left[\left\|\mathtt{clip}_\lambda\left(\mathcal{Q}\left(\widehat{\Delta}_i\left(x^{k+1},x^k\right)\right)\right)-\Delta_i\left(x^{k+1},x^k\right)\right\|^2\mid[3]\right]\right]
$$

$$
+\mathbb{E}\left[\frac{1}{G_C^k}\sum_{i\in\mathcal{G}_C^k}8\mathbb{E}_k\left[\left\|\Delta_i\left(x^{k+1},x^k\right)-\Delta\left(x^{k+1},x^k\right)\right\|^2\mid[3]\right]\right]
$$

$$
\overset{\text{Lemma D.6}}{\leq}\mathbb{E}\left[\frac{1}{G_C^k}\sum_{i\in\mathcal{G}_C^k}80\mathbb{E}_k\left[\left\|\mathcal{Q}\left(\widehat{\Delta}_i\left(x^{k+1},x^k\right)\right)-\Delta_i\left(x^{k+1},x^k\right)\right\|^2\mid[3]\right]\right]
$$

$$
+\mathbb{E}\left[\frac{1}{G_C^k}\sum_{i\in\mathcal{G}_C^k}8\mathbb{E}_k\left[\left\|\Delta_i\left(x^{k+1},x^k\right)-\Delta\left(x^{k+1},x^k\right)\right\|^2\mid[3]\right]\right].
$$

Using variance decomposition we get

$$\widehat{T}_2 \leq \mathbb{E}\left[\frac{1}{G_C^k}\sum_{i\in\mathcal{G}_C^k}80\mathbb{E}_k\left[\left\|\mathcal{Q}\left(\widehat{\Delta}_i\left(x^{k+1},x^k\right)\right)\right\|^2 \mid [3]\right]\right]$$

$$-\mathbb{E}\left[\frac{1}{G_C^k}\sum_{i\in\mathcal{G}_C^k}80\mathbb{E}_k\left[\left\|\Delta_i\left(x^{k+1},x^k\right)\right\|^2 \mid [3]\right]\right]$$

$$+\mathbb{E}\left[\frac{1}{G_C^k}\sum_{i\in\mathcal{G}_C^k}8\mathbb{E}_k\left[\left\|\Delta_i\left(x^{k+1},x^k\right)-\Delta\left(x^{k+1},x^k\right)\right\|^2 \mid [3]\right]\right].$$

Using properties of unbiased compressors (Definition 2.2) we have

$$\widehat{T}_2 \leq \mathbb{E}\left[\frac{1}{G_C^k}\sum_{i\in\mathcal{G}_C^k}80(1+\omega)\mathbb{E}_k\left[\left\|\widehat{\Delta}_i\left(x^{k+1},x^k\right)\right\|^2 \mid [3]\right]\right]$$

$$-\mathbb{E}\left[\frac{1}{G_C^k}\sum_{i\in\mathcal{G}_C^k}80\mathbb{E}_k\left[\left\|\Delta_i\left(x^{k+1},x^k\right)\right\|^2 \mid [3]\right]\right]$$

$$+\mathbb{E}\left[\frac{1}{G_C^k}\sum_{i\in\mathcal{G}_C^k}8\mathbb{E}_k\left[\left\|\Delta_i\left(x^{k+1},x^k\right)-\Delta\left(x^{k+1},x^k\right)\right\|^2 \mid [3]\right]\right].$$

Also we have

$$\widehat{T}_2 \leq \mathbb{E}\left[\frac{1}{G_C^k}\sum_{i\in\mathcal{G}_C^k}80(1+\omega)\mathbb{E}_k\left[\left\|\widehat{\Delta}_i\left(x^{k+1},x^k\right)-\Delta_i\left(x^{k+1},x^k\right)\right\|^2 \mid [3]\right]\right]$$

$$+\mathbb{E}\left[\frac{1}{G_C^k}\sum_{i\in\mathcal{G}_C^k}80(1+\omega)\mathbb{E}_k\left[\left\|\Delta_i\left(x^{k+1},x^k\right)\right\|^2 \mid [3]\right]\right]$$

$$-\mathbb{E}\left[\frac{1}{G_C^k}\sum_{i\in\mathcal{G}_C^k}80\mathbb{E}_k\left[\left\|\Delta_i\left(x^{k+1},x^k\right)\right\|^2 \mid [3]\right]\right]$$

$$+\mathbb{E}\left[\frac{1}{G_C^k}\sum_{i\in\mathcal{G}_C^k}8\mathbb{E}_k\left[\left\|\Delta_i\left(x^{k+1},x^k\right)-\Delta\left(x^{k+1},x^k\right)\right\|^2 \mid [3]\right]\right].$$

Let us simplify the inequality:

$$\widehat{T}_2 \leq \mathbb{E}\left[\frac{1}{G_C^k}\sum_{i\in\mathcal{G}_C^k}80(1+\omega)\mathbb{E}_k\left[\left\|\widehat{\Delta}_i\left(x^{k+1},x^k\right)-\Delta_i\left(x^{k+1},x^k\right)\right\|^2 \mid [3]\right]\right]$$

$$+\mathbb{E}\left[\frac{1}{G_C^k}\sum_{i\in\mathcal{G}_C^k}80\omega\mathbb{E}_k\left[\left\|\Delta_i\left(x^{k+1},x^k\right)\right\|^2 \mid [3]\right]\right]$$

$$+\mathbb{E}\left[\frac{1}{G_C^k}\sum_{i\in\mathcal{G}_C^k}8\mathbb{E}_k\left[\left\|\Delta_i\left(x^{k+1},x^k\right)-\Delta\left(x^{k+1},x^k\right)\right\|^2 \mid [3]\right]\right].$$

Using a variance decomposition once again, we get

$$\widehat{T}_2 \leq \mathbb{E}\left[\frac{1}{G_C^k}\sum_{i\in\mathcal{G}_C^k}80(1+\omega)\mathbb{E}_k\left[\left\|\widehat{\Delta}_i\left(x^{k+1},x^k\right)-\Delta_i\left(x^{k+1},x^k\right)\right\|^2\mid[3]\right]\right]$$

$$+\mathbb{E}\left[\frac{1}{G_C^k}\sum_{i\in\mathcal{G}_C^k}80\omega\mathbb{E}_k\left[\left\|\Delta_i\left(x^{k+1},x^k\right)-\Delta\left(x^{k+1},x^k\right)\right\|^2\mid[3]\right]\right]$$

$$+\mathbb{E}\left[\frac{1}{G_C^k}\sum_{i\in\mathcal{G}_C^k}8\mathbb{E}_k\left[\left\|\Delta_i\left(x^{k+1},x^k\right)-\Delta\left(x^{k+1},x^k\right)\right\|^2\mid[3]\right]\right]$$

$$+\mathbb{E}\left[\frac{1}{G_C^k}\sum_{i\in\mathcal{G}_C^k}80\omega\mathbb{E}_k\left[\left\|\Delta\left(x^{k+1},x^k\right)\right\|^2\mid[3]\right]\right].$$

Using a similar argument to the one used in the previous lemma, we obtain

$$\widehat{T}_2 \leq \mathbb{E}\left[\frac{\mathcal{P}_{\mathcal{G}_C^k}}{(1-\delta)C}\sum_{i\in\mathcal{G}}80(1+\omega)\mathbb{E}_k\left[\left\|\widehat{\Delta}_i\left(x^{k+1},x^k\right)-\Delta_i\left(x^{k+1},x^k\right)\right\|^2\mid[3]\right]\right]$$

$$+\mathbb{E}\left[\frac{\mathcal{P}_{\mathcal{G}_C^k}}{(1-\delta)C}\sum_{i\in\mathcal{G}}80\omega\mathbb{E}_k\left[\left\|\Delta_i\left(x^{k+1},x^k\right)-\Delta\left(x^{k+1},x^k\right)\right\|^2\mid[3]\right]\right]$$

$$+\mathbb{E}\left[\frac{\mathcal{P}_{\mathcal{G}_C^k}}{(1-\delta)C}\sum_{i\in\mathcal{G}}8\mathbb{E}_k\left[\left\|\Delta_i\left(x^{k+1},x^k\right)-\Delta\left(x^{k+1},x^k\right)\right\|^2\mid[3]\right]\right]$$

$$+\mathbb{E}\left[\frac{\mathcal{P}_{\mathcal{G}_C^k}}{(1-\delta)C}\sum_{i\in\mathcal{G}}80\omega\mathbb{E}_k\left[\left\|\Delta\left(x^{k+1},x^k\right)\right\|^2\mid[3]\right]\right].$$

Using Assumptions [D.1, D.2, D.3](#):

$$\widehat{T}_2 \leq \mathbb{E}\left[\frac{80(1+\omega)G\mathcal{P}_{\mathcal{G}_C^k}\mathcal{L}_\pm^2}{(1-\delta)Cb}\|x^{k+1}-x^k\|^2\right]+\mathbb{E}\left[\frac{8(10\omega+1)G\mathcal{P}_{\mathcal{G}_C^k}L_\pm^2}{(1-\delta)C}\|x^{k+1}-x^k\|^2\right]$$

$$+\mathbb{E}\left[\frac{80G\mathcal{P}_{\mathcal{G}_C^k}\omega L^2}{(1-\delta)C}\|x^{k+1}-x^k\|^2\right].$$

Finally, we obtain

$$T_2 = \mathbb{E}\left[\mathbb{E}_k\left[\left\|\frac{1}{G_C^k}\sum_{i\in\mathcal{G}_C^k}\texttt{clip}_\lambda\left(\mathcal{Q}\left(\widehat{\Delta}_i\left(x^{k+1},x^k\right)\right)\right)-\texttt{ARAgg}_Q^{k+1}\right\|^2\mid[3]\right]\right]$$

$$\leq \frac{8G\mathcal{P}_{\mathcal{G}_C^k}}{(1-\delta)C}\left(10(1+\omega)\frac{\mathcal{L}_\pm^2}{b}+(10\omega+1)L_\pm^2+10\omega L^2\right)c\delta\mathbb{E}\left[\|x^{k+1}-x^k\|^2\right].$$

□

**Lemma D.12.** *Let Assumptions [2.3](#) and [D.1](#) hold. Also let us introduce the notation*

$$\texttt{ARAgg}_Q^{k+1} = \texttt{ARAgg}\left(\texttt{clip}_{\lambda_{k+1}}\left(\mathcal{Q}\left(\widehat{\Delta}_1(x^{k+1},x^k)\right)\right),\dots,\texttt{clip}_{\lambda_{k+1}}\left(\mathcal{Q}\left(\widehat{\Delta}_C(x^{k+1},x^k)\right)\right)\right).$$

*Assume that $\lambda_{k+1}=\alpha_{\lambda_{k+1}}\|x^{k+1}-x^k\|$. Then for all $k\geq 0$ the iterates produced by* Byz-VR-MARINA-PP *(Algorithm [1](#)) satisfy*

$$T_3 = \mathbb{E}\left[\mathbb{E}_k\left[\left\|\nabla f(x^{k+1})-\nabla f(x^k)-\texttt{ARAgg}_Q^{k+1}\right\|^2\mid[2]\right]\right]$$

$$\leq 2(L^2+F_{\mathcal{A}}^2\alpha_{\lambda_{k+1}}^2)\mathbb{E}\left[\|x^{k+1}-x^k\|^2\right]$$

*Proof.*

$$T_3 = \mathbb{E}\left[\mathbb{E}_k\left[\left\|\nabla f(x^{k+1}) - \nabla f(x^k) - \texttt{ARAgg}_Q^{k+1}\right\|^2 \mid [2]\right]\right]$$

$$\overset{(12)}{\leq} \mathbb{E}\left[\mathbb{E}_k\left[2\left\|\nabla f(x^{k+1}) - \nabla f(x^k)\right\|^2 + 2\left\|\texttt{ARAgg}_Q^{k+1}\right\|^2 \mid [2]\right]\right]$$

Using $L$-smoothness and Assumption 2.3 we have

$$T_3 \overset{(12)}{\leq} \mathbb{E}\left[\mathbb{E}_k\left[2L^2\left\|x^{k+1} - x^k\right\|^2 + 2F_{\mathcal{A}}^2\lambda_{k+1}^2 \mid [2]\right]\right]$$

$$\leq \mathbb{E}\left[\mathbb{E}_k\left[2L^2\left\|x^{k+1} - x^k\right\|^2 + 2F_{\mathcal{A}}^2\alpha_{\lambda_{k+1}}^2\|x^{k+1} - x^k\|^2 \mid [2]\right]\right]$$

$$\leq 2(L^2 + F_{\mathcal{A}}^2\alpha_{\lambda_{k+1}}^2)\mathbb{E}\left[\left\|x^{k+1} - x^k\right\|^2\right].$$

$\square$

**Lemma D.13.** *Let Assumptions 2.3, D.1, D.2, D.3, D.5 hold and Compression Operator satisfy Definition 2.2. Also let us introduce the notation*

$$\texttt{ARAgg}_Q^{k+1} = \texttt{ARAgg}\left(\texttt{clip}_{\lambda_{k+1}}\left(\mathcal{Q}\left(\widehat{\Delta}_1(x^{k+1}, x^k)\right)\right), \ldots, \texttt{clip}_{\lambda_{k+1}}\left(\mathcal{Q}\left(\widehat{\Delta}_C(x^{k+1}, x^k)\right)\right)\right).$$

*Then for all $k \geq 0$ the iterates produced by* Byz-VR-MARINA-PP *(Algorithm 1) satisfy*

$$\mathbb{E}\left[\left\|g^{k+1} - \nabla f\left(x^{k+1}\right)\right\|^2\right] \leq \left(1 - \frac{p}{4}\right)\mathbb{E}\left[\left\|g^k - \nabla f\left(x^k\right)\right\|^2\right]$$

$$+ \widehat{B}\mathbb{E}\left[\left\|\nabla f\left(x^k\right)\right\|^2\right] + \widehat{D}\zeta^2 + \frac{pA}{4}\|x^{k+1} - x^k\|^2,$$

*where*

$$A = \frac{4}{p}\left(\frac{80}{p}\frac{p_G\mathcal{P}_{\mathcal{G}_C^k}n}{C}\omega + 24\frac{G\mathcal{P}_{\mathcal{G}_{\widehat{C}}^k}c\delta}{(1-\delta)\widehat{C}}B + 6\widetilde{B} + \frac{4}{p}(1-p_G) + \frac{160}{p}p_G\frac{G\mathcal{P}_{\mathcal{G}_C^k}}{(1-\delta)C}c\delta\omega\right)L^2$$

$$+ \frac{4}{p}\left(\frac{8}{p}\frac{p_G\mathcal{P}_{\mathcal{G}_C^k}n}{C}(10\omega + 1) + \frac{16}{p}p_G\frac{G\mathcal{P}_{\mathcal{G}_C^k}}{(1-\delta)C}c\delta(10\omega + 1)\right)L_\pm^2$$

$$+ \frac{4}{p}\left(\frac{160}{p}p_G\frac{G\mathcal{P}_{\mathcal{G}_C^k}}{(1-\delta)C}(1+\omega)c\delta + \frac{80}{p}p_G\mathcal{P}_{\mathcal{G}_C^k}(1+\omega)\frac{n}{C}\right)\frac{\mathcal{L}_\pm^2}{b}$$

$$+ \frac{4}{p}\left(\frac{4}{p}(1-p_G)F_{\mathcal{A}}^2\alpha_{\lambda_{k+1}}^2\right),$$

$$\widehat{B} = 2\frac{\delta\mathcal{P}_{\mathcal{G}_{\widehat{C}}^k}}{1-\delta}B\left(\frac{12cG}{\widehat{C}} + p\right) + 6\widetilde{B}, \quad \widehat{D} = 2\frac{\delta\mathcal{P}_{\mathcal{G}_{\widehat{C}}^k}}{1-\delta}\left(\frac{6cG}{\widehat{C}} + p\right) + \widetilde{D},$$

*and where $\widetilde{B} := 0$ and $\widetilde{D} := 0$ when $\widehat{C} = n$, and $\widetilde{B} := \frac{\mathcal{P}_{\mathcal{G}_{\widehat{C}}^k}GB}{(1-\delta)\widehat{C}}$ and $\widetilde{D} := \frac{\mathcal{P}_{\mathcal{G}_{\widehat{C}}^k}G}{(1-\delta)\widehat{C}}$ when $\widehat{C} < n$, $p_G = \text{Prob}\left\{G_C^k \geq (1-\delta)C\right\}$, and $\mathcal{P}_{\mathcal{G}_C^k} = \text{Prob}\left\{i \in \mathcal{G}_C^k \mid G_C^k \geq (1-\delta)C\right\}$.*

*Proof.* Let us combine bounds for $A_1$ and $A_2$ together:

$$
A_0 = \mathbb{E}\left[\left\|g^{k+1} - \nabla f\left(x^{k+1}\right)\right\|^2\right]
$$

$$
\leq \left(1 + \frac{p}{2}\right)\mathbb{E}\left[\left\|\bar{g}^{k+1} - \nabla f\left(x^{k+1}\right)\right\|^2\right] + \left(1 + \frac{2}{p}\right)\mathbb{E}\left[\left\|g^{k+1} - \bar{g}^{k+1}\right\|^2\right]
$$

$$
\leq \left(1 + \frac{p}{2}\right)A_1 + \left(1 + \frac{2}{p}\right)A_2
$$

$$
\leq \left(1 + \frac{p}{2}\right)(1 - p)\left(1 + \frac{p}{4}\right)\mathbb{E}\left[\left\|g^k - \nabla f\left(x^k\right)\right\|^2\right]
$$

$$
+ \left(1 + \frac{p}{2}\right)(1 - p)p_G\left(1 + \frac{4}{p}\right)\frac{2 \cdot \mathcal{P}_{\mathcal{G}_C^k} n}{C}\left(10\omega L^2 + (10\omega + 1)L_\pm^2 + \frac{10(\omega + 1)\mathcal{L}_\pm^2}{b}\right)\mathbb{E}\left[\|x^{k+1} - x^k\|^2\right]
$$

$$
+ \left(1 + \frac{p}{2}\right)p\left(\frac{\delta \cdot \mathcal{P}_{\mathcal{G}_{\widehat{C}}^k}}{(1 - \delta)}\mathbb{E}\left[B\|\nabla f(x)\|^2 + \zeta^2\right]\right)
$$

$$
+ \left(1 + \frac{2}{p}\right)p\mathbb{E}\left[\mathbb{E}_k\left[\left\|\texttt{ARAgg}\left(\nabla f_1(x^{k+1}), \ldots, \nabla f_{\widehat{C}}(x^{k+1})\right) - \nabla f(x^{k+1})\right\|^2\right] \mid [1]\right]
$$

$$
+ \left(1 + \frac{2}{p}\right)(1 - p)p_G\mathbb{E}\left[\mathbb{E}_k\left[\left\|\frac{1}{G_C^k}\sum_{i \in \mathcal{G}_C^k}\texttt{clip}_\lambda\left(\mathcal{Q}\left(\widehat{\Delta}_i\left(x^{k+1}, x^k\right)\right)\right) - \texttt{ARAgg}_Q^{k+1}\right\|^2 \mid [3]\right]\right]
$$

$$
+ \left(1 + \frac{2}{p}\right)(1 - p)(1 - p_G)\mathbb{E}\left[\mathbb{E}_k\left[\left\|\nabla f(x^{k+1}) - \nabla f(x^k) - \texttt{ARAgg}_Q^{k+1}\right\|^2 \mid [2]\right]\right].
$$

Finally, we obtain the following bound:

$$
A_0 \overset{(12)}{\leq} \left(1 - \frac{p}{4}\right)\mathbb{E}\left[\left\|g^k - \nabla f\left(x^k\right)\right\|^2\right]
$$

$$
+ \frac{8}{p}\frac{\mathcal{P}_{\mathcal{G}_C^k} n}{C}p_G\left(10\omega L^2 + (10\omega + 1)L_\pm^2 + \frac{10(\omega + 1)\mathcal{L}_\pm^2}{b}\right)\mathbb{E}\left[\|x^{k+1} - x^k\|^2\right]
$$

$$
+ 2p\left(\frac{\delta \cdot \mathcal{P}_{\mathcal{G}_{\widehat{C}}^k}}{1 - \delta}\mathbb{E}\left[B\|\nabla f(x)\|^2 + \zeta^2\right]\right)
$$

$$
+ (p + 2)\mathbb{E}\left[\mathbb{E}_k\left[\left\|\texttt{ARAgg}\left(\nabla f_1(x^{k+1}), \ldots, \nabla f_n(x^{k+1})\right) - \nabla f(x^{k+1})\right\|^2\right] \mid [1]\right]
$$

$$
+ \frac{2}{p}p_G\mathbb{E}\left[\mathbb{E}_k\left[\left\|\frac{1}{G_C^k}\sum_{i \in \mathcal{G}_C^k}\texttt{clip}_\lambda\left(\mathcal{Q}\left(\widehat{\Delta}_i\left(x^{k+1}, x^k\right)\right)\right) - \texttt{ARAgg}_Q^{k+1}\right\|^2 \mid [3]\right]\right]
$$

$$
+ \frac{2}{p}(1 - p_G)\mathbb{E}\left[\mathbb{E}_k\left[\left\|\nabla f(x^{k+1}) - \nabla f(x^k) - \texttt{ARAgg}_Q^{k+1}\right\|^2 \mid [2]\right]\right]
$$

Now, we can apply Lemmas [D.10], [D.11], [D.12]:

$$
\begin{aligned}
A_0 &= \mathbb{E}\left[\left\|g^{k+1} - \nabla f\left(x^{k+1}\right)\right\|^2\right] \\
&\leq \left(1 - \frac{p}{4}\right)\mathbb{E}\left[\left\|g^k - \nabla f\left(x^k\right)\right\|^2\right] \\
&\quad + \frac{8}{p}\frac{\mathcal{P}_{\mathcal{G}_C^k}n}{C}p_G\left(10\omega L^2 + (10\omega+1)L_\pm^2 + \frac{10(\omega+1)\mathcal{L}_\pm^2}{b}\right)\mathbb{E}\left[\|x^{k+1} - x^k\|^2\right] \\
&\quad + (p+2)\left(\frac{8G\mathcal{P}_{\mathcal{G}_{\widehat{C}}^k}c\delta B}{(1-\delta)\widehat{C}} + 2\widetilde{B}\right)\mathbb{E}\left[\left\|\nabla f\left(x^k\right)\right\|^2 + L^2\left\|x^{k+1} - x^k\right\|^2\right] \\
&\quad + 4(p+2)\frac{G\mathcal{P}_{\mathcal{G}_{\widehat{C}}^k}c\delta}{(1-\delta)\widehat{C}}\zeta^2 + (p+2)\widetilde{\zeta}^2 \\
&\quad + \frac{2}{p}p_G\mathbb{E}\left[80(1+\omega)\frac{G\mathcal{P}_{\mathcal{G}_C^k}}{(1-\delta)C}\frac{\mathcal{L}_\pm^2}{b}c\delta\|x^{k+1} - x^k\|^2\right] \\
&\quad + \frac{2}{p}p_G\mathbb{E}\left[8(10\omega+1)\frac{G\mathcal{P}_{\mathcal{G}_C^k}}{(1-\delta)C}L_\pm^2c\delta\|x^{k+1} - x^k\|^2\right] \\
&\quad + \frac{2}{p}p_G\mathbb{E}\left[80\frac{G\mathcal{P}_{\mathcal{G}_C^k}}{(1-\delta)C}\omega L^2c\delta\|x^{k+1} - x^k\|^2\right] \\
&\quad + \frac{2}{p}(1-p_G)2(L^2 + F_\mathcal{A}^2\alpha_{\lambda_{k+1}}^2)\mathbb{E}\left[\left\|x^{k+1} - x^k\right\|^2\right] \\
&\quad + 2p\frac{\delta\mathcal{P}_{\mathcal{G}_{\widehat{C}}^k}}{1-\delta}\mathbb{E}\left[B\|\nabla f(x)\|^2 + \zeta^2\right].
\end{aligned}
$$

Finally, we have

$$
\begin{aligned}
\mathbb{E}\left[\left\|g^{k+1} - \nabla f\left(x^{k+1}\right)\right\|^2\right] &\leq \left(1 - \frac{p}{4}\right)\mathbb{E}\left[\left\|g^k - \nabla f\left(x^k\right)\right\|^2\right] \\
&\quad + \widehat{B}\mathbb{E}\left[\left\|\nabla f\left(x^k\right)\right\|^2\right] + \widehat{D}\zeta^2 + \frac{pA}{4}\mathbb{E}\left[\|x^{k+1} - x^k\|^2\right],
\end{aligned}
$$

where

$$
\begin{aligned}
A &= \frac{32p_G}{p^2}\frac{\mathcal{P}_{\mathcal{G}_C^k}n}{C}\left(10\omega L^2 + (10\omega+1)L_\pm^2 + \frac{10(\omega+1)\mathcal{L}_\pm^2}{b}\right) \\
&\quad + \frac{8}{p^2}\frac{G\mathcal{P}_{\mathcal{G}_C^k}}{(1-\delta)C}p_Gc\delta\left(80(1+\omega)\frac{\mathcal{L}_\pm^2}{b} + 8(10\omega+1)L_\pm^2 + 80\omega L^2\right) \\
&\quad + \frac{4}{p}\left(\frac{24G\mathcal{P}_{\mathcal{G}_{\widehat{C}}^k}c\delta B}{(1-\delta)\widehat{C}} + 6\widetilde{B}\right)L^2 + \frac{16(1-p_G)}{p^2}(L^2 + F_\mathcal{A}^2\alpha_{\lambda_{k+1}}^2),
\end{aligned}
$$

and

$$
\widehat{B} = 2\frac{\delta\mathcal{P}_{\mathcal{G}_{\widehat{C}}^k}}{1-\delta}B\left(\frac{12cG}{\widehat{C}} + p\right) + 6\widetilde{B}, \quad \widehat{D} = 2\frac{\delta\mathcal{P}_{\mathcal{G}_{\widehat{C}}^k}}{1-\delta}\left(\frac{6cG}{\widehat{C}} + p\right) + \widetilde{D},
$$

where $\widetilde{D} := 0$ when $\widehat{C} = n$, and $\widetilde{D} := \frac{\mathcal{P}_{\mathcal{G}_{\widehat{C}}^k}G}{(1-\delta)\widehat{C}}$ when $\widehat{C} < n$. Once we simplify the equation, we obtain

$$
\begin{aligned}
A &= \frac{4}{p}\left(80\frac{p_G\mathcal{P}_{\mathcal{G}_C^k}n}{p}\omega + 24\frac{G\mathcal{P}_{\mathcal{G}_{\widehat{C}}^k}c\delta}{(1-\delta)\widehat{C}}B + 6\widetilde{B} + \frac{4}{p}(1-p_G) + \frac{160}{p}p_G\frac{G\mathcal{P}_{\mathcal{G}_C^k}}{(1-\delta)C}c\delta\omega\right)L^2 \\
&\quad + \frac{4}{p}\left(\frac{8}{p}\frac{p_G\mathcal{P}_{\mathcal{G}_C^k}n}{C}(10\omega+1) + \frac{16}{p}p_G\frac{G\mathcal{P}_{\mathcal{G}_C^k}}{(1-\delta)C}c\delta(10\omega+1)\right)L_\pm^2 \\
&\quad + \frac{4}{p}\left(\frac{160}{p}p_G\frac{G\mathcal{P}_{\mathcal{G}_C^k}}{(1-\delta)C}(1+\omega)c\delta + \frac{80}{p}p_G\mathcal{P}_{\mathcal{G}_C^k}(1+\omega)\frac{n}{C}\right)\frac{\mathcal{L}_\pm^2}{b} \\
&\quad + \frac{4}{p}\left(\frac{4}{p}(1-p_G)F_\mathcal{A}^2\alpha_{\lambda_{k+1}}^2\right).
\end{aligned}
$$

$\square$

### D.3 Main Results

**Theorem D.14.** *Let Assumptions 2.3, D.1, D.2, D.3, D.5 hold. Setting $\lambda_{k+1} = 2\max_{i\in\mathcal{G}} L_i \left\| x^{k+1} - x^k \right\|$. Assume that*

$$0 < \gamma \leq \frac{1}{L + \sqrt{A}}, \quad 4\widehat{B} < p,$$

*where*

$$
\begin{aligned}
A =\ & \frac{4}{p}\left( 80\,\frac{p_G \mathcal{P}_{\mathcal{G}_C^k} n}{p}\omega + 24\frac{G\mathcal{P}_{\mathcal{G}_{\widehat{C}}^k} c\delta}{(1-\delta)\widehat{C}}B + 6\widetilde{B} + \frac{4}{p}(1-p_G) + \frac{160}{p}p_G\frac{G\mathcal{P}_{\mathcal{G}_C^k}}{(1-\delta)C}c\delta\omega \right) L^2 \\
& + \frac{4}{p}\left( \frac{8}{p}\frac{p_G \mathcal{P}_{\mathcal{G}_C^k} n}{C}(10\omega+1) + \frac{16}{p}p_G\frac{G\mathcal{P}_{\mathcal{G}_C^k}}{(1-\delta)C}c\delta(10\omega+1) \right) L_{\pm}^2 \\
& + \frac{4}{p}\left( \frac{160}{p}p_G\frac{G\mathcal{P}_{\mathcal{G}_C^k}}{(1-\delta)C}(1+\omega)c\delta + \frac{80}{p}p_G\mathcal{P}_{\mathcal{G}_C^k}(1+\omega)\frac{n}{C} \right)\frac{\mathcal{L}_{\pm}^2}{b} \\
& + \frac{4}{p}\left( \frac{4}{p}(1-p_G)F_{\mathcal{A}}^2\alpha_{\lambda_{k+1}}^2 \right),
\end{aligned}
$$

$$\widehat{B} = 2\frac{\delta\mathcal{P}_{\mathcal{G}_{\widehat{C}}^k}}{1-\delta}B\left( \frac{12cG}{\widehat{C}} + p \right) + 6\widetilde{B}, \quad \widehat{D} = 2\frac{\delta\mathcal{P}_{\mathcal{G}_{\widehat{C}}^k}}{1-\delta}\left( \frac{6cG}{\widehat{C}} + p \right) + \widetilde{D},$$

*and where $\widetilde{B} := 0$ and $\widetilde{D} := 0$ when $\widehat{C} = n$, and $\widetilde{B} := \frac{\mathcal{P}_{\mathcal{G}_{\widehat{C}}^k}GB}{(1-\delta)\widehat{C}}$ and $\widetilde{D} := \frac{\mathcal{P}_{\mathcal{G}_{\widehat{C}}^k}G}{(1-\delta)\widehat{C}}$ when $\widehat{C} < n$, and*

$$
\begin{aligned}
\mathcal{P}_{\mathcal{G}_C^k} &= \frac{C}{np_G} \cdot \sum_{(1-\delta)C \leq t \leq C} \left( \binom{G-1}{t-1}\binom{n-G}{C-t}\left( \binom{n}{C} \right)^{-1} \right), \\
p_G &= \mathrm{Prob}\left\{ G_C^k \geq (1-\delta)\,C \right\} \\
&= \sum_{\lceil (1-\delta)C \rceil \leq t \leq C} \left( \binom{G}{t}\binom{n-G}{C-t}\binom{n-1}{C-1}^{-1} \right).
\end{aligned}
$$

*Then for all $K \geq 0$ the iterates produced by* Byz-VR-MARINA *(Algorithm 1) satisfy*

$$\mathbb{E}\left[ \left\| \nabla f\left( \widehat{x}^K \right) \right\|^2 \right] \leq \frac{2\Phi^0}{\gamma\left( 1 - \frac{4\widehat{B}}{p} \right)(K+1)} + \frac{4\widehat{D}\zeta^2}{p - 4\widehat{B}},$$

*where $\widehat{x}^K$ is chosen uniformly at random from $x^0, x^1, \ldots, x^K$, and $\Phi^0 = f\left( x^0 \right) - f^* + \frac{2\gamma}{p}\left\| g^0 - \nabla f\left( x^0 \right) \right\|^2.$*

*Proof of Theorem D.14.* For all $k \geq 0$ we introduce $\Phi^k = f\left(x^k\right) - f^* + \frac{2\gamma}{p}\left\|g^k - \nabla f\left(x^k\right)\right\|^2$. Using the results of Lemmas D.13 and D.7, we derive

$$\mathbb{E}\left[\Phi^{k+1}\right] \overset{(D.7)}{\leq} \mathbb{E}\left[f\left(x^k\right) - f^* - \left(\frac{1}{2\gamma} - \frac{L}{2}\right)\left\|x^{k+1} - x^k\right\|^2 + \frac{\gamma}{2}\left\|g^k - \nabla f\left(x^k\right)\right\|^2\right]$$

$$- \frac{\gamma}{2}\mathbb{E}\left[\left\|\nabla f\left(x^k\right)\right\|^2\right] + \frac{2\gamma}{p}\mathbb{E}\left[\left\|g^{k+1} - \nabla f\left(x^{k+1}\right)\right\|^2\right]$$

$$\overset{(D.13)}{\leq} \mathbb{E}\left[f\left(x^k\right) - f^* - \left(\frac{1}{2\gamma} - \frac{L}{2}\right)\left\|x^{k+1} - x^k\right\|^2 + \frac{\gamma}{2}\left\|g^k - \nabla f\left(x^k\right)\right\|^2\right]$$

$$- \frac{\gamma}{2}\mathbb{E}\left[\left\|\nabla f\left(x^k\right)\right\|^2\right] + \frac{2\gamma}{p}\left(1 - \frac{p}{4}\right)\mathbb{E}\left[\left\|g^k - \nabla f\left(x^k\right)\right\|^2\right]$$

$$+ \frac{2\gamma}{p}\left(\widehat{B}\mathbb{E}\left[\left\|\nabla f\left(x^k\right)\right\|^2\right] + \widehat{D}\zeta^2 + \frac{pA}{4}\left\|x^{k+1} - x^k\right\|^2\right)$$

$$= \mathbb{E}\left[f\left(x^k\right) - f^*\right] + \frac{2\gamma}{p}\left(\left(1 - \frac{p}{4}\right) + \frac{p}{4}\right)\mathbb{E}\left[\left\|g^k - \nabla f\left(x^k\right)\right\|^2\right] + \frac{2\widehat{D}\zeta^2\gamma}{p}$$

$$+ \frac{1}{2\gamma}\left(1 - L\gamma - A\gamma^2\right)\mathbb{E}\left[\left\|x^{k+1} - x^k\right\|^2\right] - \frac{\gamma}{2}\left(1 - \frac{4\widehat{B}}{p}\right)\mathbb{E}\left[\left\|\nabla f\left(x^k\right)\right\|^2\right]$$

$$= \mathbb{E}\left[\Phi^k\right] + \frac{2\widehat{D}\zeta^2\gamma}{p} + \frac{1}{2\gamma}\left(1 - L\gamma - A\gamma^2\right)\mathbb{E}\left[\left\|x^{k+1} - x^k\right\|^2\right]$$

$$- \frac{\gamma}{2}\left(1 - \frac{4\widehat{B}}{p}\right)\mathbb{E}\left[\left\|\nabla f\left(x^k\right)\right\|^2\right].$$

Using choice of stepsize and second condition: $0 < \gamma \leq \frac{1}{L+\sqrt{A}}, 4\widehat{B} < p$ and lemma B.1 we have

$$\mathbb{E}\left[\Phi^{k+1}\right] \leq \mathbb{E}\left[\Phi^k\right] + \frac{2\widehat{D}\zeta^2\gamma}{p} - \frac{\gamma}{2}\left(1 - \frac{4\widehat{B}}{p}\right)\mathbb{E}\left[\left\|\nabla f\left(x^k\right)\right\|^2\right]$$

Next, we have $\frac{\gamma}{2}\left(1 - \frac{4\widehat{B}}{p}\right) > 0$ and $\Phi^{k+1} \geq 0$. Therefore, summing up the above inequality for $k = 0, 1, \ldots, K$ and rearranging the terms, we get

$$\frac{1}{K+1}\sum_{k=0}^{K}\mathbb{E}\left[\left\|\nabla f\left(x^k\right)\right\|^2\right] \leq \frac{2}{\gamma\left(1 - \frac{4\widehat{B}}{p}\right)(K+1)}\sum_{k=0}^{K}\left(\mathbb{E}\left[\Phi^k\right] - \mathbb{E}\left[\Phi^{k+1}\right]\right)$$

$$+ \frac{4\widehat{D}\zeta^2}{p - 4\widehat{B}}$$

$$= \frac{2\left(\mathbb{E}\left[\Phi^0\right] - \mathbb{E}\left[\Phi^{k+1}\right]\right)}{\gamma\left(1 - \frac{4\widehat{B}}{p}\right)(K+1)} + \frac{4\widehat{D}\zeta^2}{p - 4\widehat{B}}$$

$$\leq \frac{2\mathbb{E}\left[\Phi^0\right]}{\gamma\left(1 - \frac{4\widehat{B}}{p}\right)(K+1)} + \frac{4\widehat{D}\zeta^2}{p - 4\widehat{B}}.$$

$\square$

**Theorem D.15.** *Let Assumptions 2.3, D.1, D.2, D.3, D.5, 2.7 hold. Set* $\lambda_{k+1} = \max_{i\in\mathcal{G}} L_i\left\|x^{k+1} - x^k\right\|$. *Assume that*

$$0 < \gamma \leq \frac{1}{L + \sqrt{2A}}, \quad 8\widehat{B} < p$$

*where*

$$A = \frac{4}{p}\left(80\frac{p_G\mathcal{P}_{\mathcal{G}_C^k}n}{p}\omega + 24\frac{G\mathcal{P}_{\mathcal{G}_{\widehat{C}}^k}c\delta}{(1-\delta)\widehat{C}}B + 6\widetilde{B} + \frac{4}{p}(1-p_G) + \frac{160}{p}p_G\frac{G\mathcal{P}_{\mathcal{G}_C^k}}{(1-\delta)C}c\delta\omega\right)L^2$$

$$+ \frac{4}{p}\left(\frac{8}{p}\frac{p_G\mathcal{P}_{\mathcal{G}_C^k}n}{C}(10\omega+1) + \frac{16}{p}p_G\frac{G\mathcal{P}_{\mathcal{G}_C^k}}{(1-\delta)C}c\delta(10\omega+1)\right)L_{\pm}^2$$

$$+ \frac{4}{p}\left(\frac{160}{p}p_G\frac{G\mathcal{P}_{\mathcal{G}_C^k}}{(1-\delta)C}(1+\omega)c\delta + \frac{80}{p}p_G\mathcal{P}_{\mathcal{G}_C^k}(1+\omega)\frac{n}{C}\right)\frac{\mathcal{L}_{\pm}^2}{b}$$

$$+ \frac{4}{p}\left(\frac{4}{p}(1-p_G)F_{\mathcal{A}}^2\alpha_{\lambda_{k+1}}^2\right),$$

$$\widehat{B} = 2\frac{\delta\mathcal{P}_{\mathcal{G}_{\widehat{C}}^k}}{1-\delta}B\left(\frac{12cG}{\widehat{C}}+p\right) + 6\widetilde{B}, \quad \widehat{D} = 2\frac{\delta\mathcal{P}_{\mathcal{G}_{\widehat{C}}^k}}{1-\delta}\left(\frac{6cG}{\widehat{C}}+p\right) + \widetilde{D},$$

*and where $\widetilde{B} := 0$ and $\widetilde{D} := 0$ when $\widehat{C} = n$, and $\widetilde{B} := \frac{\mathcal{P}_{\mathcal{G}_{\widehat{C}}^k}GB}{(1-\delta)\widehat{C}}$ and $\widetilde{D} := \frac{\mathcal{P}_{\mathcal{G}_{\widehat{C}}^k}G}{(1-\delta)\widehat{C}}$ when $\widehat{C} < n$, and*

$$\mathcal{P}_{\mathcal{G}_C^k} = \frac{C}{np_G}\cdot\sum_{(1-\delta)C\leq t\leq C}\left(\begin{pmatrix}G-1\\t-1\end{pmatrix}\begin{pmatrix}n-G\\C-t\end{pmatrix}\left(\begin{pmatrix}n\\C\end{pmatrix}\right)^{-1}\right),$$

$$p_G = \text{Prob}\left\{G_C^k \geq (1-\delta)C\right\}$$

$$= \sum_{\lceil(1-\delta)C\rceil\leq t\leq C}\left(\begin{pmatrix}G\\t\end{pmatrix}\begin{pmatrix}n-G\\C-t\end{pmatrix}\begin{pmatrix}n-1\\C-1\end{pmatrix}^{-1}\right).$$

*Then for all $K \geq 0$ the iterates produced by* Byz-VR-MARINA *(Algorithm 1) satisfy*

$$\mathbb{E}\left[f\left(x^K\right) - f\left(x^*\right)\right] \leq (1-\rho)^K\Phi^0 + \frac{4\widehat{D}\gamma\zeta^2}{p\rho},$$

*where $\rho = \min\left[\gamma\mu\left(1 - \frac{8\widehat{B}}{p}\right), \frac{p}{8}\right]$ and $\Phi^0 = f\left(x^0\right) - f^* + \frac{4\gamma}{p}\left\|g^0 - \nabla f\left(x^0\right)\right\|^2$.*

*Proof.* For all $k \geq 0$ we introduce $\Phi^k = f\left(x^k\right) - f^* + \frac{4\gamma}{p}\left\|g^k - \nabla f\left(x^k\right)\right\|^2$. Using the results of Lemmas D.13 and D.7, we derive

$$\mathbb{E}\left[\Phi^{k+1}\right] \overset{(D.7)}{\leq} \mathbb{E}\left[f\left(x^k\right) - f^* - \left(\frac{1}{2\gamma} - \frac{L}{2}\right)\left\|x^{k+1} - x^k\right\|^2 + \frac{\gamma}{2}\left\|g^k - \nabla f\left(x^k\right)\right\|^2\right]$$

$$- \frac{\gamma}{2}\mathbb{E}\left[\left\|\nabla f\left(x^k\right)\right\|^2\right] + \frac{4\gamma}{p}\mathbb{E}\left[\left\|g^{k+1} - \nabla f\left(x^{k+1}\right)\right\|^2\right]$$

$$\overset{(D.13)}{\leq} \mathbb{E}\left[f\left(x^k\right) - f^* - \left(\frac{1}{2\gamma} - \frac{L}{2}\right)\left\|x^{k+1} - x^k\right\|^2 + \frac{\gamma}{2}\left\|g^k - \nabla f\left(x^k\right)\right\|^2\right]$$

$$- \frac{\gamma}{2}\mathbb{E}\left[\left\|\nabla f\left(x^k\right)\right\|^2\right] + \frac{4\gamma}{p}\left(1 - \frac{p}{4}\right)\mathbb{E}\left[\left\|g^k - \nabla f\left(x^k\right)\right\|^2\right]$$

$$+ \frac{4\gamma}{p}\left(\widehat{B}\mathbb{E}\left[\left\|\nabla f\left(x^k\right)\right\|^2\right] + \widehat{D}\zeta^2 + \frac{pA}{4}\left\|x^{k+1} - x^k\right\|^2\right)$$

$$= \mathbb{E}\left[f\left(x^k\right) - f^*\right] + \frac{4\gamma}{p}\left(\left(1 - \frac{p}{4}\right) + \frac{p}{8}\right)\mathbb{E}\left[\left\|g^k - \nabla f\left(x^k\right)\right\|^2\right] + \frac{4\widehat{D}\zeta^2\gamma}{p}$$

$$+ \frac{1}{2\gamma}\left(1 - L\gamma - 2A\gamma^2\right)\mathbb{E}\left[\left\|x^{k+1} - x^k\right\|^2\right] - \frac{\gamma}{2}\left(1 - \frac{8\widehat{B}}{p}\right)\mathbb{E}\left[\left\|\nabla f\left(x^k\right)\right\|^2\right].$$

Using Assumption 2.7 we obtain

$$\mathbb{E}\left[\Phi^{k+1}\right] \leq \mathbb{E}\left[f\left(x^k\right) - f^*\right] + \left(1 - \frac{p}{8}\right)\frac{4\gamma}{p}\mathbb{E}\left[\left\|g^k - \nabla f\left(x^k\right)\right\|^2\right] + \frac{4\widehat{D}\zeta^2\gamma}{p}$$
$$+ \frac{1}{2\gamma}\left(1 - L\gamma - 2A\gamma^2\right)\mathbb{E}\left[\left\|x^{k+1} - x^k\right\|^2\right]$$
$$- \gamma\mu\left(1 - \frac{8\widehat{B}}{p}\right)\mathbb{E}\left[f\left(x^k\right) - f^*\right].$$

Finally, we have

$$\mathbb{E}\left[\Phi^{k+1}\right] \leq \left(1 - \min\left[\gamma\mu\left(1 - \frac{\widehat{B}}{p}\right), \frac{p}{8}\right]\right)\mathbb{E}\left[\Phi^k\right] + \frac{4\widehat{D}\zeta^2\gamma}{p}.$$

Unrolling the recurrence with $\rho = \min\left[\gamma\mu\left(1 - \frac{8\widehat{B}}{p}\right), \frac{p}{8}\right]$, we obtain

$$\mathbb{E}\left[\Phi^k\right] \leq (1 - \rho)^K \mathbb{E}\left[\Phi^0\right] + \frac{4\widehat{D}\zeta^2\gamma}{p}\sum_{k=0}^{K-1}(1 - \rho)^k$$
$$\leq (1 - \rho)^K \mathbb{E}\left[\Phi^0\right] + \frac{4\widehat{D}\zeta^2\gamma}{p}\sum_{k=0}^{\infty}(1 - \rho)^k$$
$$= (1 - \rho)^K \mathbb{E}\left[\Phi^0\right] + \frac{4\widehat{D}\gamma\zeta^2}{p\rho}$$

Taking into account $\Phi^k \geq f\left(x^k\right) - f\left(x^*\right)$, we get the result. $\qquad\square$

# E Analysis for Bounded Compressors

## E.1 Technical Lemmas

**Lemma E.1.** *Let Assumptions D.1, D.2, D.3 and 2.4 hold and the Compression Operator satisfy Definition 2.2. We set $\lambda_{k+1} = D_Q \max_{i,j} L_{i,j}$. Let us define "ideal" estimator:*

$$
\overline{g}^{k+1} =
\begin{cases}
\frac{1}{G_{\widehat{C}}^k} \sum\limits_{i \in \mathcal{G}_{\widehat{C}}^k} \nabla f_i(x^{k+1}), & c_n = 1, & [1] \\[2mm]
g^k + \nabla f\left(x^{k+1}\right) - \nabla f\left(x^k\right), & c_n = 0 \text{ and } G_C^k < (1-\delta)C, & [2] \\[2mm]
g^k + \frac{1}{G_C^k} \sum\limits_{i \in \mathcal{G}_C^k} clip_\lambda\left(\mathcal{Q}\left(\widehat{\Delta}_i\left(x^{k+1}, x^k\right)\right)\right), & c_n = 0 \text{ and } G_C^k \geq (1-\delta)C. & [3]
\end{cases}
$$

*Then for all $k \geq 0$ the iterates produced by Byz-VR-MARINA-PP (Algorithm 1) satisfy*

$$
A_1 = \mathbb{E}\left[\left\|\overline{g}^{k+1} - \nabla f\left(x^{k+1}\right)\right\|^2\right]
$$

$$
\leq (1-p)\mathbb{E}\left[\left\|g^k - \nabla f(x^k)\right\|^2\right] + p\frac{\delta \mathcal{P}_{\mathcal{G}_{\widehat{C}}^k}}{(1-\delta)}\mathbb{E}\left[B\|\nabla f(x)\|^2 + \zeta^2\right]
$$

$$
+ (1-p)p_G\frac{\mathcal{P}_{\mathcal{G}_C^k}G}{C^2(1-\delta)^2}\left(\omega L^2 + (\omega+1)L_\pm^2 + \frac{(\omega+1)\mathcal{L}_\pm^2}{b}\right)\mathbb{E}\left[\|x^{k+1} - x^k\|^2\right].
$$

*where $p_G = \mathrm{Prob}\left\{G_C^k \geq (1-\delta)C\right\}$ and $\mathcal{P}_{\mathcal{G}_C^k} = \mathrm{Prob}\left\{i \in \mathcal{G}_C^k \mid G_C^k \geq (1-\delta)C\right\}$.*

*Proof.* Similarly to general analysis, we start from conditional expectations:

$$
A_1 = \mathbb{E}\left[\left\|\overline{g}^{k+1} - \nabla f\left(x^{k+1}\right)\right\|^2\right]
$$

$$
= \mathbb{E}\left[\mathbb{E}_k\left[\left\|\overline{g}^{k+1} - \nabla f\left(x^{k+1}\right)\right\|^2\right]\right]
$$

$$
= (1-p)\,p_G\mathbb{E}\left[\mathbb{E}_k\left[\left\|g^k + \frac{1}{G_C^k}\sum_{i \in \mathcal{G}_C^k} clip_\lambda\left(\mathcal{Q}\left(\widehat{\Delta}_i\left(x^{k+1}, x^k\right)\right)\right) - \nabla f\left(x^{k+1}\right)\right\|^2\right] \mid [3]\right]
$$

$$
+ (1-p)(1-p_G)\mathbb{E}\left[\mathbb{E}_k\left[\left\|g^k - \nabla f(x^k)\right\|^2\right] \mid [2]\right]
$$

$$
+ p\mathbb{E}\left[\left\|\frac{1}{G_{\widehat{C}}^k}\sum_{i \in \mathcal{G}_{\widehat{C}}^k}\nabla f_i(x^{k+1}) - \nabla f(x^{k+1})\right\|^2\right]. \tag{26}
$$

Using (12) and $\nabla f\left(x^k\right) - \nabla f\left(x^k\right) = 0$ we obtain

$$
B_1 = \mathbb{E}\left[\mathbb{E}_k\left[\left\|g^k + \frac{1}{G_C^k}\sum_{i \in \mathcal{G}_C^k} clip_\lambda\left(\mathcal{Q}\left(\widehat{\Delta}_i\left(x^{k+1}, x^k\right)\right)\right) - \nabla f\left(x^{k+1}\right)\right\|^2\right] \mid [3]\right]
$$

$$
= \mathbb{E}\left[\mathbb{E}_k\left[\left\|g^k + \frac{1}{G_C^k}\sum_{i \in \mathcal{G}_C^k} clip_\lambda\left(\mathcal{Q}\left(\widehat{\Delta}_i\left(x^{k+1}, x^k\right)\right)\right) - \nabla f\left(x^{k+1}\right) + \nabla f\left(x^k\right) - \nabla f\left(x^k\right)\right\|^2\right] \mid [3]\right]
$$

Using $\lambda_{k+1} = D_Q \max_{i,j} L_{i,j} \|x^{k+1} - x^k\|$ we can guarantee that clipping operator becomes identical since we have

$$\left\| \mathcal{Q}\left(\widehat{\Delta}_i\left(x^{k+1}, x^k\right)\right) \right\| \leq D_Q \left\| \widehat{\Delta}_i\left(x^{k+1}, x^k\right) \right\|$$

$$\leq D_Q \left\| \frac{1}{b} \sum_{j \in m} \nabla f_{i,j}(x^{k+1}) - \nabla f_{i,j}(x^k) \right\|$$

$$\leq D_Q \frac{1}{b} \sum_{j \in m} \left\| \nabla f_{i,j}(x^{k+1}) - \nabla f_{i,j}(x^k) \right\|$$

$$\leq D_Q \max_j L_{i,j} \left\| x^{k+1} - x^k \right\|$$

$$\leq D_Q \max_{i,j} L_{i,j} \left\| x^{k+1} - x^k \right\|.$$

Therefore, we can continue as follows

$$B_1 = \mathbb{E}\left[ \mathbb{E}_k \left[ \left\| g^k + \frac{1}{G_C^k} \sum_{i \in \mathcal{G}_C^k} \mathcal{Q}\left(\widehat{\Delta}_i\left(x^{k+1}, x^k\right)\right) - \nabla f\left(x^{k+1}\right) \right\|^2 \right] \mid [3] \right]$$

$$= \mathbb{E}\left[ \mathbb{E}_k \left[ \left\| g^k + \frac{1}{G_C^k} \sum_{i \in \mathcal{G}_C^k} \mathcal{Q}\left(\widehat{\Delta}_i\left(x^{k+1}, x^k\right)\right) - \nabla f\left(x^{k+1}\right) + \nabla f\left(x^k\right) - \nabla f\left(x^k\right) \right\|^2 \right] \mid [3] \right].$$

Moreover, we can avoid application of Young's inequality and use variance decomposition instead:

$$B_1 \leq \mathbb{E}\left[ \left\| g^k - \nabla f\left(x^k\right) \right\|^2 \right]$$

$$+ \mathbb{E}\left[ \mathbb{E}_k \left[ \left\| \frac{1}{G_C^k} \sum_{i \in \mathcal{G}_C^k} \mathcal{Q}\left(\widehat{\Delta}_i\left(x^{k+1}, x^k\right)\right) - \left(\nabla f(x^{k+1}) - \nabla f(x^k)\right) \right\|^2 \right] \mid [3] \right]$$

$$\leq \mathbb{E}\left[ \left\| g^k - \nabla f(x^k) \right\|^2 \right]$$

$$+ \mathbb{E}\left[ \mathbb{E}_k \left[ \left\| \frac{1}{G_C^k} \sum_{i \in \mathcal{G}_C^k} \mathcal{Q}\left(\widehat{\Delta}_i\left(x^{k+1}, x^k\right)\right) - \Delta\left(x^{k+1}, x^k\right) \right\|^2 \right] \mid [3] \right]. \tag{27}$$

Let us consider the last part of the inequality. Note that $G_C^k \geq (1 - \delta)C$ in this case and

$$B_1' = \mathbb{E}\left[ \mathbb{E}_k \left[ \left\| \frac{1}{G_C^k} \sum_{i \in \mathcal{G}_C^k} \mathcal{Q}\left(\widehat{\Delta}_i\left(x^{k+1}, x^k\right)\right) - \Delta\left(x^{k+1}, x^k\right) \right\|^2 \right] \mid [3] \right]$$

$$= \mathbb{E}\left[ \mathbb{E}_{S_k} \left[ \mathbb{E}_k \left[ \left\| \frac{1}{G_C^k} \sum_{i \in \mathcal{G}_C^k} \mathcal{Q}\left(\widehat{\Delta}_i\left(x^{k+1}, x^k\right)\right) - \Delta\left(x^{k+1}, x^k\right) \right\|^2 \right] \mid [3] \right] \right]$$

$$\leq \frac{1}{C^2(1-\delta)^2} \mathbb{E}\left[ \mathbb{E}_{S_k} \left[ \sum_{i \in \mathcal{G}_C^k} \mathbb{E}_k \left[ \left\| \mathcal{Q}\left(\widehat{\Delta}_i\left(x^{k+1}, x^k\right)\right) - \Delta\left(x^{k+1}, x^k\right) \right\|^2 \right] \mid [3] \right] \right]$$

$$\leq \frac{1}{C^2(1-\delta)^2} \mathbb{E}\left[ \sum_{i \in \mathcal{G}} \mathbb{E}_{S_k}\left[\mathcal{I}_{\mathcal{G}_C^k}\right] \mathbb{E}_k \left[ \left\| \mathcal{Q}\left(\widehat{\Delta}_i\left(x^{k+1}, x^k\right)\right) - \Delta\left(x^{k+1}, x^k\right) \right\|^2 \right] \mid [3] \right]$$

$$= \frac{1}{C^2(1-\delta)^2} \mathbb{E}\left[ \sum_{i \in \mathcal{G}} \mathcal{P}_{\mathcal{G}_C^k} \cdot \mathbb{E}_k \left[ \left\| \mathcal{Q}\left(\widehat{\Delta}_i\left(x^{k+1}, x^k\right)\right) - \Delta\left(x^{k+1}, x^k\right) \right\|^2 \right] \mid [3] \right], \tag{28}$$

where $\mathcal{I}_{\mathcal{G}_C^k}$ is an indicator function for the event $\left\{ i \in \mathcal{G}_C^k \mid G_C^k \geq (1 - \delta)C \right\}$ and $\mathcal{P}_{\mathcal{G}_C^k} = \text{Prob}\left\{ i \in \mathcal{G}_C^k \mid G_C^k \geq (1 - \delta)C \right\}$ is probability of such event. Note that $\mathbb{E}_{S_k}\left[\mathcal{I}_{\mathcal{G}_C^k}\right] = \mathcal{P}_{\mathcal{G}_C^k}$. In the

case of uniform sampling of clients, we have

$$\forall i \in \mathcal{G} \quad \mathcal{P}_{\mathcal{G}_C^k} = \text{Prob}\left\{i \in \mathcal{G}_C^k \mid G_C^k \geq (1-\delta)\,C\right\}$$

$$= \frac{C}{n}\frac{1}{p_G} \cdot \sum_{(1-\delta)C \leq t \leq C} \left(\binom{G-1}{t-1}\binom{n-G}{C-t}\left(\binom{n-1}{C-1}\right)^{-1}\right).$$

Now, we can continue with inequalities:

$$B_1' \leq \frac{\mathcal{P}_{\mathcal{G}_C^k}}{C^2(1-\delta)^2}\mathbb{E}\left[\sum_{i \in \mathcal{G}}\mathbb{E}_k\left[\left\|\mathcal{Q}\left(\widehat{\Delta}_i\left(x^{k+1},x^k\right)\right) - \Delta\left(x^{k+1},x^k\right)\right\|^2\right] \mid [3]\right]$$

$$\leq \frac{\mathcal{P}_{\mathcal{G}_C^k}}{C^2(1-\delta)^2}\mathbb{E}\left[\sum_{i \in \mathcal{G}}\mathbb{E}_k\left[\mathbb{E}_Q\left[\left\|\mathcal{Q}\left(\widehat{\Delta}_i\left(x^{k+1},x^k\right)\right) - \Delta\left(x^{k+1},x^k\right)\right\|^2\right]\right] \mid [3]\right]$$

$$\leq \frac{\mathcal{P}_{\mathcal{G}_C^k}}{C^2(1-\delta)^2}\mathbb{E}\left[\sum_{i \in \mathcal{G}}\mathbb{E}_k\left[\mathbb{E}_Q\left[\left\|\mathcal{Q}\left(\widehat{\Delta}_i\left(x^{k+1},x^k\right)\right) - \Delta_i\left(x^{k+1},x^k\right)\right\|^2\right]\right] \mid [3]\right]$$

$$+ \frac{\mathcal{P}_{\mathcal{G}_C^k}}{C^2(1-\delta)^2}\mathbb{E}\left[\sum_{i \in \mathcal{G}}\mathbb{E}_k\left[\left\|\Delta_i\left(x^{k+1},x^k\right) - \Delta\left(x^{k+1},x^k\right)\right\|^2\right] \mid [3]\right].$$

Using variance decomposition, we have

$$B_1' \leq \frac{\mathcal{P}_{\mathcal{G}_C^k}}{C^2(1-\delta)^2}\mathbb{E}\left[\sum_{i \in \mathcal{G}}\mathbb{E}_k\left[\mathbb{E}_Q\left[\left\|\mathcal{Q}\left(\widehat{\Delta}_i\left(x^{k+1},x^k\right)\right)\right\|^2\right]\right] - \sum_{i \in \mathcal{G}}\left\|\Delta_i\left(x^{k+1},x^k\right)\right\|^2 \mid [3]\right]$$

$$+ \frac{\mathcal{P}_{\mathcal{G}_C^k}}{C^2(1-\delta)^2}\mathbb{E}\left[\sum_{i \in \mathcal{G}}\mathbb{E}_k\left[\left\|\Delta_i\left(x^{k+1},x^k\right) - \Delta\left(x^{k+1},x^k\right)\right\|^2\right] \mid [3]\right].$$

Applying the definition of unbiased compressor, we get

$$B_1' \leq \frac{\mathcal{P}_{\mathcal{G}_C^k}}{C^2(1-\delta)^2}\mathbb{E}\left[\sum_{i \in \mathcal{G}}(1+\omega)\mathbb{E}_k\left\|\widehat{\Delta}_i\left(x^{k+1},x^k\right)\right\|^2 - \sum_{i \in \mathcal{G}}\left\|\Delta_i\left(x^{k+1},x^k\right)\right\|^2 \mid [3]\right]$$

$$+ \frac{\mathcal{P}_{\mathcal{G}_C^k}}{C^2(1-\delta)^2}\mathbb{E}\left[\sum_{i \in \mathcal{G}}\left\|\Delta_i\left(x^{k+1},x^k\right) - \Delta\left(x^{k+1},x^k\right)\right\|^2 \mid [3]\right]$$

$$\leq \frac{\mathcal{P}_{\mathcal{G}_C^k}}{C^2(1-\delta)^2}\mathbb{E}\left[\sum_{i \in \mathcal{G}}(1+\omega)\mathbb{E}_k\left\|\widehat{\Delta}_i\left(x^{k+1},x^k\right) - \Delta_i\left(x^{k+1},x^k\right)\right\|^2\right]$$

$$+ \frac{\mathcal{P}_{\mathcal{G}_C^k}}{C^2(1-\delta)^2}\mathbb{E}\left[\sum_{i \in \mathcal{G}}(1+\omega)\mathbb{E}_k\left\|\Delta_i\left(x^{k+1},x^k\right)\right\|^2 - \sum_{i \in \mathcal{G}}\mathbb{E}_k\left\|\Delta_i\left(x^{k+1},x^k\right)\right\|^2 \mid [3]\right]$$

$$+ \frac{\mathcal{P}_{\mathcal{G}_C^k}}{C^2(1-\delta)^2}\mathbb{E}\left[\sum_{i \in \mathcal{G}}\left\|\Delta_i\left(x^{k+1},x^k\right) - \Delta\left(x^{k+1},x^k\right)\right\|^2 \mid [3]\right].$$

Next, we rearrange terms and derive

$$
\begin{aligned}
B_1' &\leq \frac{\mathcal{P}_{\mathcal{G}_C^k}}{C^2(1-\delta)^2}(1+\omega)\mathbb{E}\left[\sum_{i\in\mathcal{G}}\mathbb{E}_k\left[\left\|\widehat{\Delta}_i\left(x^{k+1},x^k\right)-\Delta_i\left(x^{k+1},x^k\right)\right\|^2\right]\mid [3]\right] \\
&\quad + \frac{\mathcal{P}_{\mathcal{G}_C^k}}{C^2(1-\delta)^2}\omega\mathbb{E}\left[\sum_{i\in\mathcal{G}}\left\|\Delta_i\left(x^{k+1},x^k\right)\right\|^2\mid [3]\right] \\
&\quad + \frac{\mathcal{P}_{\mathcal{G}_C^k}}{C^2(1-\delta)^2}\mathbb{E}\left[\sum_{i\in\mathcal{G}}\left\|\Delta_i\left(x^{k+1},x^k\right)-\Delta\left(x^{k+1},x^k\right)\right\|^2\mid [3]\right] \\
&= \frac{\mathcal{P}_{\mathcal{G}_C^k}}{C^2(1-\delta)^2}(1+\omega)\mathbb{E}\left[\sum_{i\in\mathcal{G}}\mathbb{E}_k\left[\left\|\widehat{\Delta}_i\left(x^{k+1},x^k\right)-\Delta_i\left(x^{k+1},x^k\right)\right\|^2\right]\mid [3]\right] \\
&\quad + \frac{\mathcal{P}_{\mathcal{G}_C^k}}{C^2(1-\delta)^2}\omega\mathbb{E}\left[\sum_{i\in\mathcal{G}}\left\|\Delta_i\left(x^{k+1},x^k\right)-\Delta\left(x^{k+1},x^k\right)\right\|^2+\left\|\Delta\left(x^{k+1},x^k\right)\right\|^2\mid [3]\right] \\
&\quad + \frac{\mathcal{P}_{\mathcal{G}_C^k}}{C^2(1-\delta)^2}\mathbb{E}\left[\sum_{i\in\mathcal{G}}\left\|\Delta_i\left(x^{k+1},x^k\right)-\Delta\left(x^{k+1},x^k\right)\right\|^2\mid [3]\right].
\end{aligned}
$$

Rearranging terms leads to

$$
\begin{aligned}
B_1' &\leq \frac{\mathcal{P}_{\mathcal{G}_C^k}}{C^2(1-\delta)^2}(1+\omega)\mathbb{E}\left[\sum_{i\in\mathcal{G}}\mathbb{E}_k\left[\left\|\widehat{\Delta}_i\left(x^{k+1},x^k\right)-\Delta_i\left(x^{k+1},x^k\right)\right\|^2\right]\mid [3]\right] \\
&\quad + \frac{\mathcal{P}_{\mathcal{G}_C^k}}{C^2(1-\delta)^2}(\omega+1)\mathbb{E}\left[\sum_{i\in\mathcal{G}}\left\|\Delta_i\left(x^{k+1},x^k\right)-\Delta\left(x^{k+1},x^k\right)\right\|^2\mid [3]\right] \\
&\quad + \frac{\mathcal{P}_{\mathcal{G}_C^k}}{C^2(1-\delta)^2}\omega\mathbb{E}\left[\sum_{i\in\mathcal{G}}\left\|\Delta\left(x^{k+1},x^k\right)\right\|^2\mid [3]\right].
\end{aligned}
$$

Now we apply Assumptions D.1, D.2, D.3:

$$
\begin{aligned}
B_1' &\leq \frac{\mathcal{P}_{\mathcal{G}_C^k}}{C^2(1-\delta)^2}(1+\omega)\mathbb{E}\left[G\frac{\mathcal{L}_\pm^2}{b}\|x^{k+1}-x^k\|^2\right] \\
&\quad + \frac{\mathcal{P}_{\mathcal{G}_C^k}}{C^2(1-\delta)^2}(\omega+1)\mathbb{E}\left[GL_\pm^2\|x^{k+1}-x^k\|^2\right] + \frac{\mathcal{P}_{\mathcal{G}_C^k}}{C^2(1-\delta)^2}\omega\mathbb{E}\left[GL^2\left\|x^{k+1}-x^k\right\|^2\right].
\end{aligned}
$$

Finally, we have

$$
B_1' \leq \frac{\mathcal{P}_{\mathcal{G}_C^k}\cdot G}{C^2(1-\delta)^2}\left(\omega L^2+(\omega+1)L_\pm^2+\frac{(\omega+1)\mathcal{L}_\pm^2}{b}\right)\mathbb{E}\left[\|x^{k+1}-x^k\|^2\right].
$$

Let us plug the obtained results in (27):

$$
\begin{aligned}
B_1 &\leq \mathbb{E}\left[\left\|g^k-\nabla f(x^k)\right\|^2\right] \\
&\quad + \frac{\mathcal{P}_{\mathcal{G}_C^k}\cdot G}{C^2(1-\delta)^2}\left(\omega L^2+(\omega+1)L_\pm^2+\frac{(\omega+1)\mathcal{L}_\pm^2}{b}\right)\mathbb{E}\left[\|x^{k+1}-x^k\|^2\right].
\end{aligned}
$$

Also, we have

$$
\begin{aligned}
A_1 &= \mathbb{E}\left[\left\|\overline{g}^{k+1} - \nabla f(x^{k+1})\right\|^2\right] \\
&\overset{(26),(25)}{\leq} (1-p)p_G B_1 + (1-p)(1-p_G)\mathbb{E}\left[\left\|g^k - \nabla f(x^k)\right\|^2\right] \\
&\quad + p\frac{\delta \cdot \mathcal{P}_{\mathcal{G}_{\widehat{C}}^k}}{(1-\delta)}\mathbb{E}\left[B\|\nabla f(x)\|^2 + \zeta^2\right] \\
&\leq (1-p)p_G\mathbb{E}\left[\left\|g^k - \nabla f(x^k)\right\|^2\right] + p\frac{\delta \cdot \mathcal{P}_{\mathcal{G}_{\widehat{C}}^k}}{(1-\delta)}\mathbb{E}\left[B\|\nabla f(x)\|^2 + \zeta^2\right] \\
&\quad + (1-p)p_G \frac{\mathcal{P}_{\mathcal{G}_C^k} \cdot G}{C^2(1-\delta)^2}\left(\omega L^2 + (\omega+1)L_{\pm}^2 + \frac{(\omega+1)\mathcal{L}_{\pm}^2}{b}\right)\mathbb{E}\left[\|x^{k+1} - x^k\|^2\right] \\
&\quad + (1-p)(1-p_G)\mathbb{E}\left[\left\|g^k - \nabla f(x^k)\right\|^2\right].
\end{aligned}
$$

Rearranging the terms, we get

$$
\begin{aligned}
A_1 &\leq (1-p)\mathbb{E}\left[\left\|g^k - \nabla f(x^k)\right\|^2\right] + \frac{\delta\mathcal{P}_{\mathcal{G}_{\widehat{C}}^k}}{(1-\delta)}\mathbb{E}\left[B\|\nabla f(x)\|^2 + \zeta^2\right] \\
&\quad + (1-p)p_G\frac{\mathcal{P}_{\mathcal{G}_C^k}G}{C^2(1-\delta)^2}\left(\omega L^2 + (\omega+1)L_{\pm}^2 + \frac{(\omega+1)\mathcal{L}_{\pm}^2}{b}\right)\mathbb{E}\left[\|x^{k+1} - x^k\|^2\right].
\end{aligned}
$$

$\square$

**Lemma E.2.** *Let Assumptions D.1, D.2, D.3, D.4, 2.4 hold and the compression operator satisfy Definition 2.2. Also, let us introduce the notation*

$$
\mathtt{ARAgg}_Q^{k+1} = \mathtt{ARAgg}\left(\mathtt{clip}_{\lambda_{k+1}}\left(\mathcal{Q}\left(\widehat{\Delta}_1(x^{k+1}, x^k)\right)\right), \ldots, \mathtt{clip}_{\lambda_{k+1}}\left(\mathcal{Q}\left(\widehat{\Delta}_C(x^{k+1}, x^k)\right)\right)\right).
$$

*Then for all $k \geq 0$ the iterates produced by* Byz-VR-MARINA-PP *(Algorithm 1) satisfy*

$$
\begin{aligned}
T_2 &= \mathbb{E}\left[\mathbb{E}_k\left[\left\|\frac{1}{G_C^k}\sum_{i \in \mathcal{G}_C^k}\mathtt{clip}_{\lambda}\left(\mathcal{Q}\left(\widehat{\Delta}_i\left(x^{k+1}, x^k\right)\right)\right) - \mathtt{ARAgg}_Q^{k+1}\right\|^2 \mid [3]\right]\right] \\
&\leq 4\frac{G\mathcal{P}_{\mathcal{G}_C^k}}{C(1-\delta)}c\delta\left((1+\omega)\frac{\mathcal{L}_{\pm}^2}{b} + (\omega+1)L_{\pm}^2 + \omega L^2\right)\mathbb{E}\left[\|x^{k+1} - x^k\|^2\right],
\end{aligned}
$$

*where $\mathcal{P}_{\mathcal{G}_C^k} = \mathrm{Prob}\left\{i \in \mathcal{G}_C^k \mid G_C^k \geq (1-\delta)C\right\}$.*

*Proof.* By definition of the robust aggregation, we have

$$
\begin{aligned}
T_2 &= \mathbb{E}\left[\mathbb{E}_k\left[\left\|\frac{1}{G_C^k}\sum_{i \in \mathcal{G}_C^k}\mathtt{clip}_{\lambda}\left(\mathcal{Q}\left(\widehat{\Delta}_i\left(x^{k+1}, x^k\right)\right)\right) - \mathtt{ARAgg}_Q^{k+1}\right\|^2 \mid [3]\right]\right] \\
&\leq \mathbb{E}\left[\frac{c\delta}{D_2}\sum_{\substack{i,l \in \mathcal{G}_C^k \\ i \neq l}}\mathbb{E}_k\left[\left\|\mathtt{clip}_{\lambda}\left(\mathcal{Q}\left(\widehat{\Delta}_i\left(x^{k+1}, x^k\right)\right)\right) - \mathtt{clip}_{\lambda}\left(\mathcal{Q}\left(\widehat{\Delta}_l\left(x^{k+1}, x^k\right)\right)\right)\right\|^2 \mid [3]\right]\right],
\end{aligned}
$$

where $D_2 = G_C^k(G_C^k - 1)$.

Using $\lambda_{k+1} = D_Q \max_{i,j} L_{i,j} \|x^{k+1} - x^k\|$ we can guarantee that clipping operator becomes identical since we have $\forall i \in \mathcal{G}$

$$
\begin{aligned}
\left\| \mathcal{Q} \left( \widehat{\Delta}_i \left( x^{k+1}, x^k \right) \right) \right\| &\leq D_Q \left\| \widehat{\Delta}_i \left( x^{k+1}, x^k \right) \right\| \\
&\leq D_Q \left\| \frac{1}{b} \sum_{j \in m} \nabla f_{i,j}(x^{k+1}) - \nabla f_{i,j}(x^k) \right\| \\
&\leq D_Q \frac{1}{b} \sum_{j \in m} \left\| \nabla f_{i,j}(x^{k+1}) - \nabla f_{i,j}(x^k) \right\| \\
&\leq D_Q \max_j L_{i,j} \left\| x^{k+1} - x^k \right\|.
\end{aligned}
\tag{29}
$$

Let us consider pair-wise differences: $\forall i, l \in \mathcal{G}$

$$
\begin{aligned}
T_2'(i,l) &= \mathbb{E}_k \left[ \left\| \mathtt{clip}_\lambda \left( \mathcal{Q} \left( \widehat{\Delta}_i \left( x^{k+1}, x^k \right) \right) \right) - \mathtt{clip}_\lambda \left( \mathcal{Q} \left( \widehat{\Delta}_l \left( x^{k+1}, x^k \right) \right) \right) \right\|^2 \mid [3] \right] \\
&= \mathbb{E}_k \left[ \left\| \mathcal{Q} \left( \widehat{\Delta}_i \left( x^{k+1}, x^k \right) \right) - \mathcal{Q} \left( \widehat{\Delta}_l \left( x^{k+1}, x^k \right) \right) \right\|^2 \mid [3] \right] \\
&= \mathbb{E}_k \left[ \left\| \mathcal{Q} \left( \widehat{\Delta}_i \left( x^{k+1}, x^k \right) \right) - \Delta_i \left( x^{k+1}, x^k \right) + \Delta_l \left( x^{k+1}, x^k \right) - \mathcal{Q} \left( \widehat{\Delta}_l \left( x^{k+1}, x^k \right) \right) \right\|^2 \mid [3] \right] \\
&\quad + \mathbb{E}_k \left[ \left\| \Delta_i \left( x^{k+1}, x^k \right) - \Delta_l \left( x^{k+1}, x^k \right) \right\|^2 \mid [3] \right] \\
&\overset{(12)}{\leq} 2\mathbb{E}_k \left[ \left\| \mathcal{Q} \left( \widehat{\Delta}_i \left( x^{k+1}, x^k \right) \right) - \Delta_i \left( x^{k+1}, x^k \right) \right\|^2 \mid [3] \right] \\
&\quad + 2\mathbb{E}_k \left[ \left\| \Delta_l \left( x^{k+1}, x^k \right) - \mathcal{Q} \left( \widehat{\Delta}_l \left( x^{k+1}, x^k \right) \right) \right\|^2 \mid [3] \right] \\
&\quad + \mathbb{E}_k \left[ \left\| \Delta_l \left( x^{k+1}, x^k \right) - \Delta_i \left( x^{k+1}, x^k \right) \right\|^2 \mid [3] \right] ] \\
&\overset{(12)}{\leq} 2\mathbb{E}_k \left[ \left\| \mathcal{Q} \left( \widehat{\Delta}_i \left( x^{k+1}, x^k \right) \right) - \Delta_i \left( x^{k+1}, x^k \right) \right\|^2 \mid [3] \right] \\
&\quad + 2\mathbb{E}_k \left[ \left\| \Delta_l \left( x^{k+1}, x^k \right) - \mathcal{Q} \left( \widehat{\Delta}_l \left( x^{k+1}, x^k \right) \right) \right\|^2 \mid [3] \right] \\
&\quad + 2\mathbb{E}_k \left[ \left\| \Delta_l \left( x^{k+1}, x^k \right) - \Delta \left( x^{k+1}, x^k \right) \right\|^2 + \left\| \Delta_i \left( x^{k+1}, x^k \right) - \Delta \left( x^{k+1}, x^k \right) \right\|^2 \mid [3] \right].
\end{aligned}
$$

Now we can combine all the parts together:

$$
\widehat{T}_2 = \mathbb{E}\left[\frac{1}{G_C^k(G_C^k-1)} \sum_{\substack{i,l\in\mathcal{G}_C^k \\ i\neq l}} T_2'(i,l)\right]
$$

$$
\leq \mathbb{E}\left[\frac{1}{D_2} \sum_{\substack{i,l\in\mathcal{G}_C^k \\ i\neq l}} 2\mathbb{E}_k\left[\left\|\mathcal{Q}\left(\widehat{\Delta}_i\left(x^{k+1},x^k\right)\right) - \Delta_i\left(x^{k+1},x^k\right)\right\|^2 \mid [3]\right]\right]
$$

$$
+ \mathbb{E}\left[\frac{1}{D_2} \sum_{\substack{i,l\in\mathcal{G}_C^k \\ i\neq l}} 2\mathbb{E}_k\left[\left\|\Delta_l\left(x^{k+1},x^k\right) - \mathcal{Q}\left(\widehat{\Delta}_l\left(x^{k+1},x^k\right)\right)\right\|^2 \mid [3]\right]\right]
$$

$$
+ \mathbb{E}\left[\frac{1}{D_2} \sum_{\substack{i,l\in\mathcal{G}_C^k \\ i\neq l}} 2\mathbb{E}_k\left[\left\|\Delta_l\left(x^{k+1},x^k\right) - \Delta\left(x^{k+1},x^k\right)\right\|^2 \mid [3]\right]\right]
$$

$$
+ \mathbb{E}\left[\frac{1}{D_2} \sum_{\substack{i,l\in\mathcal{G}_C^k \\ i\neq l}} 2\mathbb{E}_k\left[\left\|\Delta_i\left(x^{k+1},x^k\right) - \Delta\left(x^{k+1},x^k\right)\right\|^2 \mid [3]\right]\right].
$$

Rearranging the terms, we get

$$
\widehat{T}_2 \leq \mathbb{E}\left[\frac{4}{G_C^k} \sum_{i\in\mathcal{G}_C^k} \mathbb{E}_k\left[\left\|\mathcal{Q}\left(\widehat{\Delta}_i\left(x^{k+1},x^k\right)\right) - \Delta_i\left(x^{k+1},x^k\right)\right\|^2 \mid [3]\right]\right]
$$

$$
+ \mathbb{E}\left[\frac{4}{G_C^k} \sum_{i\in\mathcal{G}_C^k} \mathbb{E}_k\left[\left\|\Delta_i\left(x^{k+1},x^k\right) - \Delta\left(x^{k+1},x^k\right)\right\|^2 \mid [3]\right]\right].
$$

Using variance decomposition, we get

$$
\widehat{T}_2 \leq \mathbb{E}\left[\frac{1}{G_C^k} \sum_{i\in\mathcal{G}_C^k} 4\mathbb{E}_k\left[\left\|\mathcal{Q}\left(\widehat{\Delta}_i\left(x^{k+1},x^k\right)\right)\right\|^2 \mid [3]\right]\right]
$$

$$
- \mathbb{E}\left[\frac{1}{G_C^k} \sum_{i\in\mathcal{G}_C^k} 4\mathbb{E}_k\left[\left\|\Delta_i\left(x^{k+1},x^k\right)\right\|^2 \mid [3]\right]\right]
$$

$$
+ \mathbb{E}\left[\frac{1}{G_C^k} \sum_{i\in\mathcal{G}_C^k} 4\mathbb{E}_k\left[\left\|\Delta_i\left(x^{k+1},x^k\right) - \Delta\left(x^{k+1},x^k\right)\right\|^2 \mid [3]\right]\right].
$$

Using the properties of unbiased compressors, we obtain

$$\widehat{T}_2 \leq \mathbb{E}\left[\frac{1}{G_C^k}\sum_{i\in\mathcal{G}_C^k}4(1+\omega)\mathbb{E}_k\left[\left\|\widehat{\Delta}_i\left(x^{k+1},x^k\right)\right\|^2 \mid [3]\right]\right]$$

$$-\mathbb{E}\left[\frac{1}{G_C^k}\sum_{i\in\mathcal{G}_C^k}4\mathbb{E}_k\left[\left\|\Delta_i\left(x^{k+1},x^k\right)\right\|^2 \mid [3]\right]\right]$$

$$+\mathbb{E}\left[\frac{1}{G_C^k}\sum_{i\in\mathcal{G}_C^k}4\mathbb{E}_k\left[\left\|\Delta_i\left(x^{k+1},x^k\right)-\Delta\left(x^{k+1},x^k\right)\right\|^2 \mid [3]\right]\right]$$

$$\leq \mathbb{E}\left[\frac{1}{G_C^k}\sum_{i\in\mathcal{G}_C^k}4(1+\omega)\mathbb{E}_k\left[\left\|\widehat{\Delta}_i\left(x^{k+1},x^k\right)-\Delta_i\left(x^{k+1},x^k\right)\right\|^2 \mid [3]\right]\right]$$

$$+\mathbb{E}\left[\frac{1}{G_C^k}\sum_{i\in\mathcal{G}_C^k}4(1+\omega)\mathbb{E}_k\left[\left\|\Delta_i\left(x^{k+1},x^k\right)\right\|^2 \mid [3]\right]\right]$$

$$-\mathbb{E}\left[\frac{1}{G_C^k}\sum_{i\in\mathcal{G}_C^k}4\mathbb{E}_k\left[\left\|\Delta_i\left(x^{k+1},x^k\right)\right\|^2 \mid [3]\right]\right]$$

$$+\mathbb{E}\left[\frac{1}{G_C^k}\sum_{i\in\mathcal{G}_C^k}4\mathbb{E}_k\left[\left\|\Delta_i\left(x^{k+1},x^k\right)-\Delta\left(x^{k+1},x^k\right)\right\|^2 \mid [3]\right]\right].$$

Let us simplify the inequality:

$$\widehat{T}_2 \leq \mathbb{E}\left[\frac{1}{G_C^k}\sum_{i\in\mathcal{G}_C^k}4(1+\omega)\mathbb{E}_k\left[\left\|\widehat{\Delta}_i\left(x^{k+1},x^k\right)-\Delta_i\left(x^{k+1},x^k\right)\right\|^2 \mid [3]\right]\right]$$

$$+\mathbb{E}\left[\frac{1}{G_C^k}\sum_{i\in\mathcal{G}_C^k}4\omega\mathbb{E}_k\left[\left\|\Delta_i\left(x^{k+1},x^k\right)\right\|^2 \mid [3]\right]\right]$$

$$+\mathbb{E}\left[\frac{1}{G_C^k}\sum_{i\in\mathcal{G}_C^k}4\mathbb{E}_k\left[\left\|\Delta_i\left(x^{k+1},x^k\right)-\Delta\left(x^{k+1},x^k\right)\right\|^2 \mid [3]\right]\right].$$

Using variance decomposition once again, we get

$$\widehat{T}_2 \leq \mathbb{E}\left[\frac{1}{G_C^k}\sum_{i\in\mathcal{G}_C^k}4(1+\omega)\mathbb{E}_k\left[\left\|\widehat{\Delta}_i\left(x^{k+1},x^k\right)-\Delta_i\left(x^{k+1},x^k\right)\right\|^2 \mid [3]\right]\right]$$

$$+\mathbb{E}\left[\frac{1}{G_C^k}\sum_{i\in\mathcal{G}_C^k}4\omega\mathbb{E}_k\left[\left\|\Delta_i\left(x^{k+1},x^k\right)-\Delta\left(x^{k+1},x^k\right)\right\|^2 \mid [3]\right]\right]$$

$$+\mathbb{E}\left[\frac{1}{G_C^k}\sum_{i\in\mathcal{G}_C^k}4\mathbb{E}_k\left[\left\|\Delta_i\left(x^{k+1},x^k\right)-\Delta\left(x^{k+1},x^k\right)\right\|^2 \mid [3]\right]\right]$$

$$+\mathbb{E}\left[\frac{1}{G_C^k}\sum_{i\in\mathcal{G}_C^k}4\omega\mathbb{E}_k\left[\left\|\Delta\left(x^{k+1},x^k\right)\right\|^2 \mid [3]\right]\right].$$

Then, we apply similar arguments to the ones used in deriving (28):

$$\widehat{T}_2 \leq \mathbb{E}\left[\frac{\mathcal{P}_{\mathcal{G}_C^k}}{C(1-\delta)}\sum_{i\in\mathcal{G}}4(1+\omega)\mathbb{E}_k\left[\left\|\widehat{\Delta}_i\left(x^{k+1},x^k\right)-\Delta_i\left(x^{k+1},x^k\right)\right\|^2 \mid [3]\right]\right]$$
$$+ \mathbb{E}\left[\frac{\mathcal{P}_{\mathcal{G}_C^k}}{C(1-\delta)}\sum_{i\in\mathcal{G}}4\omega\mathbb{E}_k\left[\left\|\Delta_i\left(x^{k+1},x^k\right)-\Delta\left(x^{k+1},x^k\right)\right\|^2 \mid [3]\right]\right]$$
$$+ \mathbb{E}\left[\frac{\mathcal{P}_{\mathcal{G}_C^k}}{C(1-\delta)}\sum_{i\in\mathcal{G}}4\mathbb{E}_k\left[\left\|\Delta_i\left(x^{k+1},x^k\right)-\Delta\left(x^{k+1},x^k\right)\right\|^2 \mid [3]\right]\right]$$
$$+ \mathbb{E}\left[\frac{\mathcal{P}_{\mathcal{G}_C^k}}{C(1-\delta)}\sum_{i\in\mathcal{G}}4\omega\mathbb{E}_k\left[\left\|\Delta\left(x^{k+1},x^k\right)\right\|^2 \mid [3]\right]\right].$$

Using Assumptions D.1, D.2, D.3:

$$\widehat{T}_2 \leq \mathbb{E}\left[4(1+\omega)\frac{G\mathcal{P}_{\mathcal{G}_C^k}}{C(1-\delta)}\frac{\mathcal{L}_\pm^2}{b}\|x^{k+1}-x^k\|^2\right] + \mathbb{E}\left[4(\omega+1)\frac{G\mathcal{P}_{\mathcal{G}_C^k}}{C(1-\delta)}\omega L_\pm^2\|x^{k+1}-x^k\|^2\right]$$
$$+ \mathbb{E}\left[4\frac{G\mathcal{P}_{\mathcal{G}_C^k}}{C(1-\delta)}\omega L^2\|x^{k+1}-x^k\|^2\right].$$

Finally, we obtain

$$T_2 = \mathbb{E}\left[\mathbb{E}_k\left[\left\|\frac{1}{G_C^k}\sum_{i\in\mathcal{G}_C^k}\texttt{clip}_\lambda\left(\mathcal{Q}\left(\widehat{\Delta}_i\left(x^{k+1},x^k\right)\right)\right)-\texttt{ARAgg}_Q^{k+1}\right\|^2 \mid [3]\right]\right]$$
$$\leq 4\frac{G\mathcal{P}_{\mathcal{G}_C^k}}{C(1-\delta)}c\delta\left((1+\omega)\frac{\mathcal{L}_\pm^2}{b}+(\omega+1)L_\pm^2+\omega L^2\right)\mathbb{E}\left[\|x^{k+1}-x^k\|^2\right].$$

$\square$

**Lemma E.3.** *Let Assumptions 2.3, D.1, D.2, D.3, D.4, D.5, 2.4 hold and the compression operator satisfy Definition 2.2. We set $\lambda_{k+1} = D_Q\max_{i,j}L_{i,j}\|x^{k+1}-x^k\|$. Also, let us introduce the notation*

$$\texttt{ARAgg}_Q^{k+1} = \texttt{ARAgg}\left(\texttt{clip}_{\lambda_{k+1}}\left(\mathcal{Q}\left(\widehat{\Delta}_1(x^{k+1},x^k)\right)\right),\ldots,\texttt{clip}_{\lambda_{k+1}}\left(\mathcal{Q}\left(\widehat{\Delta}_C(x^{k+1},x^k)\right)\right)\right).$$

*Then for all $k \geq 0$ the iterates produced by* Byz-VR-MARINA-PP *(Algorithm 1) satisfy*

$$\mathbb{E}\left[\left\|g^{k+1}-\nabla f\left(x^{k+1}\right)\right\|^2\right] \leq \left(1-\frac{p}{2}\right)\mathbb{E}\left[\left\|g^k-\nabla f\left(x^k\right)\right\|^2\right]$$
$$+ \widehat{B}\mathbb{E}\left[\left\|\nabla f\left(x^k\right)\right\|^2\right] + \widehat{D}\zeta^2 + \frac{pA}{4}\|x^{k+1}-x^k\|^2,$$

*with*

$$A = \frac{4}{p}\left(\frac{p_G\mathcal{P}_{\mathcal{G}_C^k}G}{C^2(1-\delta)^2}\omega + \frac{8G\mathcal{P}_{\mathcal{G}_{\widehat{C}}^k}c\delta}{(1-\delta)\widehat{C}}B + 6\widetilde{B} + \frac{4}{p}(1-p_G) + \frac{8}{p}p_G\frac{G\mathcal{P}_{\mathcal{G}_C^k}}{C(1-\delta)}c\delta\omega\right)L^2$$
$$+ \frac{4}{p}\left(\frac{p_G\mathcal{P}_{\mathcal{G}_C^k}G}{C^2(1-\delta)^2}(\omega+1) + \frac{8}{p}p_G\frac{G\mathcal{P}_{\mathcal{G}_C^k}}{C(1-\delta)}c\delta(\omega+1)\right)\left(L_\pm^2 + \frac{\mathcal{L}_\pm^2}{b}\right)$$
$$+ \frac{16}{p^2}(1-p_G)F_\mathcal{A}^2\left(D_Q\max_{i,j}L_{i,j}\right)^2$$

$$\widehat{B} = 2\frac{\delta\mathcal{P}_{\mathcal{G}_{\widehat{C}}^k}}{1-\delta}B\left(\frac{12cG}{\widehat{C}}+p\right)+6\widetilde{B}, \quad \widehat{D} = 2\frac{\delta\mathcal{P}_{\mathcal{G}_{\widehat{C}}^k}}{1-\delta}\left(\frac{6cG}{\widehat{C}}+p\right)+\widetilde{D},$$

*where $\widetilde{B} := 0$ and $\widetilde{D} := 0$ when $\widehat{C} = n$, and $\widetilde{B} := \frac{\mathcal{P}_{\mathcal{G}_{\widehat{C}}^k}GB}{(1-\delta)\widehat{C}}$ and $\widetilde{D} := \frac{\mathcal{P}_{\mathcal{G}_{\widehat{C}}^k}G}{(1-\delta)\widehat{C}}$ when $\widehat{C} < n$, $p_G = \mathrm{Prob}\left\{G_C^k \geq (1-\delta)C\right\}$ and $\mathcal{P}_{\mathcal{G}_C^k} = \mathrm{Prob}\left\{i \in \mathcal{G}_C^k \mid G_C^k \geq (1-\delta)C\right\}$.*

*Proof.* Let us combine bounds for $A_1$ and $A_2$ together:

$$A_0 = \mathbb{E}\left[\left\|g^{k+1} - \nabla f\left(x^{k+1}\right)\right\|^2\right]$$

$$\leq \left(1 + \frac{p}{2}\right)\mathbb{E}\left[\left\|\bar{g}^{k+1} - \nabla f\left(x^{k+1}\right)\right\|^2\right] + \left(1 + \frac{2}{p}\right)\mathbb{E}\left[\left\|g^{k+1} - \bar{g}^{k+1}\right\|^2\right]$$

$$\leq \left(1 + \frac{p}{2}\right)A_1 + \left(1 + \frac{2}{p}\right)A_2$$

$$\leq \left(1 + \frac{p}{2}\right)(1-p)\mathbb{E}\left[\left\|g^k - \nabla f(x^k)\right\|^2\right] + \left(1 + \frac{p}{2}\right)p\frac{\delta\mathcal{P}_{\mathcal{G}^k_{\widehat{C}}}}{(1-\delta)}\mathbb{E}\left[B\|\nabla f(x)\|^2 + \zeta^2\right]$$

$$+ \left(1 + \frac{p}{2}\right)(1-p)p_G\frac{\mathcal{P}_{\mathcal{G}^k_C}G}{C^2(1-\delta)^2}\left(\omega L^2 + (\omega+1)L^2_\pm + \frac{(\omega+1)\mathcal{L}^2_\pm}{b}\right)\mathbb{E}\left[\|x^{k+1} - x^k\|^2\right]$$

$$+ \left(1 + \frac{2}{p}\right)p\mathbb{E}\left[\mathbb{E}_k\left[\left\|\mathtt{ARAgg}\left(\nabla f_1(x^{k+1}), \ldots, \nabla f_n(x^{k+1})\right) - \nabla f(x^{k+1})\right\|^2\right] \mid [1]\right]$$

$$+ \left(1 + \frac{2}{p}\right)(1-p)p_G\mathbb{E}\left[\mathbb{E}_k\left[\left\|\frac{1}{G^k_C}\sum_{i\in\mathcal{G}^k_C}\mathtt{clip}_\lambda\left(\mathcal{Q}\left(\widehat{\Delta}_i\left(x^{k+1}, x^k\right)\right)\right) - \mathtt{ARAgg}^{k+1}_Q\right\|^2 \mid [3]\right]\right]$$

$$+ \left(1 + \frac{2}{p}\right)(1-p)(1-p_G)\mathbb{E}\left[\mathbb{E}_k\left[\left\|\nabla f(x^{k+1}) - \nabla f(x^k) - \mathtt{ARAgg}^{k+1}_Q\right\|^2 \mid [2]\right]\right].$$

Using Lemma E.2 and lemmas from General Analysis (Lemmas D.10 and D.12) we have

$$A_0 = \mathbb{E}\left[\left\|g^{k+1} - \nabla f\left(x^{k+1}\right)\right\|^2\right]$$

$$\leq \left(1 - \frac{p}{2}\right)\mathbb{E}\left[\left\|g^k - \nabla f\left(x^k\right)\right\|^2\right] + 2p\frac{\delta\mathcal{P}_{\mathcal{G}^k_{\widehat{C}}}}{(1-\delta)}\mathbb{E}\left[B\|\nabla f(x)\|^2 + \zeta^2\right]$$

$$+ \left(1 - \frac{p}{2}\right)p_G\frac{\mathcal{P}_{\mathcal{G}^k_C}G}{C^2(1-\delta)^2}\left(\omega L^2 + (\omega+1)L^2_\pm + \frac{(\omega+1)\mathcal{L}^2_\pm}{b}\right)\mathbb{E}\left[\|x^{k+1} - x^k\|^2\right]$$

$$+ (p+2)\left(\frac{8G\mathcal{P}_{\mathcal{G}^k_{\widehat{C}}}c\delta B}{(1-\delta)\widehat{C}} + 2\widetilde{B}\right)\mathbb{E}\left[\left\|\nabla f\left(x^k\right)\right\|^2 + L^2\left\|x^{k+1} - x^k\right\|^2\right]$$

$$+ 4(p+2)\frac{G\mathcal{P}_{\mathcal{G}^k_{\widehat{C}}}c\delta}{(1-\delta)\widehat{C}}\zeta^2 + (p+2)\widetilde{\zeta}^2$$

$$+ \frac{2}{p}p_G\mathbb{E}\left[4(1+\omega)\frac{G\mathcal{P}_{\mathcal{G}^k_C}}{C(1-\delta)}c\delta\frac{\mathcal{L}^2_\pm}{b}\|x^{k+1} - x^k\|^2\right]$$

$$+ \frac{2}{p}p_G\mathbb{E}\left[4(\omega+1)\frac{G\mathcal{P}_{\mathcal{G}^k_C}}{C(1-\delta)}c\delta L^2_\pm\|x^{k+1} - x^k\|^2\right]$$

$$+ \frac{2}{p}p_G\mathbb{E}\left[4\omega\frac{G\mathcal{P}_{\mathcal{G}^k_C}}{C(1-\delta)}c\delta L^2\|x^{k+1} - x^k\|^2\right] + \frac{2}{p}(1-p_G)2(L^2 + F^2_{\mathcal{A}}\alpha^2_{\lambda_{k+1}})\mathbb{E}\left[\left\|x^{k+1} - x^k\right\|^2\right].$$

Finally, we have

$$\mathbb{E}\left[\left\|g^{k+1} - \nabla f\left(x^{k+1}\right)\right\|^2\right] \leq \left(1 - \frac{p}{2}\right)\mathbb{E}\left[\left\|g^k - \nabla f\left(x^k\right)\right\|^2\right]$$

$$+ \widehat{B}\mathbb{E}\left[\left\|\nabla f\left(x^k\right)\right\|^2\right] + \widehat{D}\zeta^2 + \frac{pA}{4}\|x^{k+1} - x^k\|^2,$$

where

$$A = \frac{4}{p}\left(\frac{p_G\mathcal{P}_{\mathcal{G}^k_C}G}{C^2(1-\delta)^2}\omega + \frac{8G\mathcal{P}_{\mathcal{G}^k_{\widehat{C}}}c\delta}{(1-\delta)\widehat{C}}B + 6\widetilde{B} + \frac{4}{p}(1-p_G) + \frac{8}{p}p_G\frac{G\mathcal{P}_{\mathcal{G}^k_C}}{C(1-\delta)}c\delta\omega\right)L^2$$

$$+ \frac{4}{p}\left(\frac{p_G\mathcal{P}_{\mathcal{G}^k_C}G}{C^2(1-\delta)^2}(\omega+1) + \frac{8}{p}p_G\frac{G\mathcal{P}_{\mathcal{G}^k_C}}{C(1-\delta)}c\delta(\omega+1)\right)\left(L^2_\pm + \frac{\mathcal{L}^2_\pm}{b}\right)$$

$$+ \frac{16}{p^2}(1-p_G)F^2_{\mathcal{A}}\left(D_Q\max_{i,j}L_{i,j}\right)^2$$

and

$$\widehat{B} = 2\frac{\delta\mathcal{P}_{\mathcal{G}_{\widehat{C}}^k}}{1-\delta}B\left(\frac{12cG}{\widehat{C}} + p\right) + 6\widetilde{B}, \quad \widehat{D} = 2\frac{\delta\mathcal{P}_{\mathcal{G}_{\widehat{C}}^k}}{1-\delta}\left(\frac{6cG}{\widehat{C}} + p\right) + \widetilde{D}.$$

$\square$

## E.2  Main Results

**Theorem E.4.** *Let Assumptions* [2.3](#), [D.1](#), [D.2](#), [D.3](#), [D.4](#), [D.5](#), [2.4](#) *hold. Setting* $\lambda_{k+1} = \max_{i,j} L_{i,j}\left\|x^{k+1} - x^k\right\|$. *Assume that*

$$0 < \gamma \leq \frac{1}{L + \sqrt{A}}, \quad 4\widehat{B} < p,$$

*where*

$$A = \frac{4}{p}\left(\frac{p_G\mathcal{P}_{\mathcal{G}_C^k}G}{C^2(1-\delta)^2}\omega + \frac{8G\mathcal{P}_{\mathcal{G}_{\widehat{C}}^k}c\delta}{(1-\delta)\widehat{C}}B + 6\widetilde{B} + \frac{4}{p}(1-p_G) + \frac{8}{p}p_G\frac{G\mathcal{P}_{\mathcal{G}_C^k}}{C(1-\delta)}c\delta\omega\right)L^2$$

$$+ \frac{4}{p}\left(\frac{p_G\mathcal{P}_{\mathcal{G}_C^k}G}{C^2(1-\delta)^2}(\omega+1) + \frac{8}{p}p_G\frac{G\mathcal{P}_{\mathcal{G}_C^k}}{C(1-\delta)}c\delta(\omega+1)\right)\left(L_{\pm}^2 + \frac{\mathcal{L}_{\pm}^2}{b}\right)$$

$$+ \frac{16}{p^2}(1-p_G)F_{\mathcal{A}}^2\left(D_Q\max_{i,j}L_{i,j}\right)^2$$

$$\widehat{B} = 2\frac{\delta\mathcal{P}_{\mathcal{G}_{\widehat{C}}^k}}{1-\delta}B\left(\frac{12cG}{\widehat{C}} + p\right) + 6\widetilde{B}, \quad \widehat{D} = 2\frac{\delta\mathcal{P}_{\mathcal{G}_{\widehat{C}}^k}}{1-\delta}\left(\frac{6cG}{\widehat{C}} + p\right) + \widetilde{D},$$

*where* $\widetilde{B} := 0$ *and* $\widetilde{D} := 0$ *when* $\widehat{C} = n$, *and* $\widetilde{B} := \frac{\mathcal{P}_{\mathcal{G}_{\widehat{C}}^k}GB}{(1-\delta)\widehat{C}}$ *and* $\widetilde{D} := \frac{\mathcal{P}_{\mathcal{G}_{\widehat{C}}^k}G}{(1-\delta)\widehat{C}}$ *when* $\widehat{C} < n$, *and*

$$\mathcal{P}_{\mathcal{G}_C^k} = \frac{C}{np_G}\cdot\sum_{(1-\delta)C \leq t \leq C}\left(\binom{G-1}{t-1}\binom{n-G}{C-t}\left(\binom{n}{C}\right)^{-1}\right),$$

$$p_G = \text{Prob}\left\{G_C^k \geq (1-\delta)C\right\}$$

$$= \sum_{\lceil(1-\delta)C\rceil \leq t \leq C}\left(\binom{G}{t}\binom{n-G}{C-t}\binom{n}{C}^{-1}\right).$$

*Then for all* $K \geq 0$ *the iterates produced by* Byz-VR-MARINA *(Algorithm [1](#)) satisfy*

$$\mathbb{E}\left[\left\|\nabla f\left(\widehat{x}^K\right)\right\|^2\right] \leq \frac{2\Phi^0}{\gamma\left(1 - \frac{4\widehat{B}}{p}\right)(K+1)} + \frac{2\widehat{D}\zeta^2}{p - 4\widehat{B}},$$

*where* $\widehat{x}^K$ *is chosen uniformly at random from* $x^0, x^1, \ldots, x^K$, *and* $\Phi^0 = f\left(x^0\right) - f^* + \frac{\gamma}{p}\left\|g^0 - \nabla f\left(x^0\right)\right\|^2$.

*Proof.* The proof is analogous to the proof of Theorem [D.14](#). $\square$

**Theorem E.5.** *Let Assumptions* [2.3](#), [2.4](#), [D.1](#), [D.2](#), [D.3](#), [D.4](#), [D.5](#), [2.7](#) *hold. Setting* $\lambda_{k+1} = \max_{i,j} L_{i,j}\left\|x^{k+1} - x^k\right\|$. *Assume that*

$$0 < \gamma \leq \frac{1}{L + \sqrt{2A}}, \quad 8\widehat{B} < p,$$

*where*

$$A = \frac{4}{p}\left(\frac{p_G\mathcal{P}_{\mathcal{G}_C^k}G}{C^2(1-\delta)^2}\omega + \frac{8G\mathcal{P}_{\mathcal{G}_{\widehat{C}}^k}c\delta}{(1-\delta)\widehat{C}}B + 6\widetilde{B} + \frac{4}{p}(1-p_G) + \frac{8}{p}p_G\frac{G\mathcal{P}_{\mathcal{G}_C^k}}{C(1-\delta)}c\delta\omega\right)L^2$$

$$+ \frac{4}{p}\left(\frac{p_G\mathcal{P}_{\mathcal{G}_C^k}G}{C^2(1-\delta)^2}(\omega+1) + \frac{8}{p}p_G\frac{G\mathcal{P}_{\mathcal{G}_C^k}}{C(1-\delta)}c\delta(\omega+1)\right)\left(L_{\pm}^2 + \frac{\mathcal{L}_{\pm}^2}{b}\right)$$

$$+ \frac{16}{p^2}(1-p_G)F_{\mathcal{A}}^2\left(D_Q\max_{i,j}L_{i,j}\right)^2$$

$$\widehat{B} = 2\frac{\delta \mathcal{P}_{\mathcal{G}_{\widehat{C}}^k}}{1-\delta} B \left(\frac{12cG}{\widehat{C}} + p\right) + 6\widetilde{B}, \quad \widehat{D} = 2\frac{\delta \mathcal{P}_{\mathcal{G}_{\widehat{C}}^k}}{1-\delta} \left(\frac{6cG}{\widehat{C}} + p\right) + \widetilde{D},$$

*where $\widetilde{B} := 0$ and $\widetilde{D} := 0$ when $\widehat{C} = n$, and $\widetilde{B} := \frac{\mathcal{P}_{\mathcal{G}_{\widehat{C}}^k} GB}{(1-\delta)\widehat{C}}$ and $\widetilde{D} := \frac{\mathcal{P}_{\mathcal{G}_{\widehat{C}}^k} G}{(1-\delta)\widehat{C}}$ when $\widehat{C} < n$, and where $p_G = \mathrm{Prob}\left\{G_C^k \geq (1-\delta)C\right\}$ and $\mathcal{P}_{\mathcal{G}_C^k} = \mathrm{Prob}\left\{i \in \mathcal{G}_C^k \mid G_C^k \geq (1-\delta)C\right\}$. Then for all $K \geq 0$ the iterates produced by* Byz-VR-MARINA *(Algorithm 1) satisfy*

$$\mathbb{E}\left[f\left(x^K\right) - f\left(x^*\right)\right] \leq (1-\rho)^K \Phi^0 + \frac{2\widehat{D}\zeta^2}{p\rho},$$

*where $\rho = \min\left[\gamma\mu\left(1 - \frac{8\widehat{B}}{p}\right), \frac{p}{4}\right]$ and $\Phi^0 = f\left(x^0\right) - f^* + \frac{2\gamma}{p}\left\|g^0 - \nabla f\left(x^0\right)\right\|^2$.*

*Proof.* The proof is analogous to the proof of Theorem D.15. $\qquad\square$

### E.3  On the Technical Non-Triviality of the Analysis

As we explain in the main part of the paper, the main reason why we propose to use clipping is to handle the situations when Byzantine workers form a majority during some communication rounds since the existing approaches are vulnerable to such scenarios. However, the introduction of the clipping does not come for free: if the clipping level is too small, clipping can create a noticeable bias to the updates. Because of this issue, existing works such as (Zhang et al., 2020b; Gorbunov et al., 2020) use non-trivial policies for the choice of the clipping level, and the analysis in these works differs significantly from the existing analysis for the methods without clipping. The analysis of Byz-VR-MARINA is based on the unbiasedness of vectors $\mathcal{Q}(\hat{\Delta}_i(x^{k+1}, x^k))$, i.e., on the following identity: $\mathbb{E}[\mathcal{Q}(\hat{\Delta}_i(x^{k+1}, x^k)) \mid x^{k+1}, x^k] = \Delta_i(x^{k+1}, x^k) = \nabla f_i(x^{k+1}) - \nabla f_i(x^k)$. Since $\mathbb{E}[\mathtt{clip}_{\lambda_{k+1}}(\mathcal{Q}(\hat{\Delta}_i(x^{k+1}, x^k))) \mid x^{k+1}, x^k] \neq \nabla f_i(x^{k+1}) - \nabla f_i(x^k)$ in general, to analyze Byz-VR-MARINA-PP we also use a special choice of the clipping level: $\lambda_{k+1} = \alpha_{k+1}\|x^{k+1} - x^k\|$. To illustrate the main reasons for that, let us consider the case of uncompressed communication ($\mathcal{Q}(x) \equiv x$). In this setup, for large enough $\alpha_{k+1}$ we have $\mathtt{clip}_{\lambda_{k+1}} \hat{\Delta}_i(x^{k+1}, x^k) = \hat{\Delta}_i(x^{k+1}, x^k)$ for all $i \in \mathcal{G}$ (due to Assumption 2.6), which allows us using a similar proof to the one for Byz-VR-MARINA when good workers form a majority in a round. Moreover, when Byzantine workers form a majority, our choice of the clipping level allows us to bound the second moment of the shift from the Byzantine workers as $\sim \|x^{k+1} - x^k\|^2$ (see Lemmas D.9 and D.12), i.e., the second moment of the shift is of the same scale as the variance of $\{g_i\}_{i \in \mathcal{G}}$, which goes to zero. Next, to properly analyze these two situations, we overcame another technical challenge related to the estimation of the conditional expectations and probabilities of corresponding events (see Lemmas D.9 and D.10 and formulas for $p_G$ and $\mathcal{P}_{\mathcal{G}_C^k}$ at the beginning of Section 4). In particular, the derivation of formula (24) is quite non-standard for stochastic optimization literature: there are two sources of stochasticity – one comes from the sampling of clients, and the other one comes from the sampling of stochastic gradients and compression. This leads to the estimation of variance of the average of the random number of random vectors, which is novel on its own. In addition, when the compression operator is used, the analysis becomes even more involved since one cannot directly apply the main property of unbiased compression (Definition 2.2), and we use Lemma D.6 in the proof to address this issue. It is also worth mentioning that in contrast to Byz-VR-MARINA, our method does not require full participation even with a small probability $p$. Instead, it is sufficient for Byz-VR-MARINA-PP to sample a large enough cohort of $\widehat{C}$ clients with probability $p$ to ensure that Byzantine workers form a minority in such rounds.

# F  Byz-VR-MARINA-PP+: Simplified Version of Byz-VR-MARINA-PP

In this section, we present a simplified version of Byz-VR-MARINA-PP called Byz-VR-MARINA-PP+ (see Algorithm 3). The only difference between the two methods is in Line 10: Byz-VR-MARINA+ does not apply robust aggregation when $c_k = 0$ and just averages the clipped vectors received from the set of clients $S_k$. Nevertheless, when $c_k = 1$, i.e., a large cohort of clients is sampled, the method still uses robust aggregation.

---

**Algorithm 3** Byz-VR-MARINA-PP+: Simplified Byz-VR-MARINA-PP

---

1: **Input:** vectors $x^0, g^0 \in \mathbb{R}^d$, stepsize $\gamma$, mini-batch size $b$, probability $p \in (0,1]$, number of iterations $K$, $(\delta, c)$-ARAgg, clients' sample size $1 \leq C \leq \widehat{C} \leq n$, clipping coefficients $\{\alpha_k\}_{k \geq 1}$
2: **for** $k = 0, 1, \ldots, K-1$ **do**
3:      Get a sample from Bernoulli distribution with parameter $p$: $c_k \sim \text{Be}(p)$
4:      Sample the set of clients $S_k \subseteq [n]$, $|S_k| = C$ if $c_k = 0$; otherwise $|S_k| = \widehat{C}$
5:      Broadcast $g^k, c_k$ to all workers
6:      **for** $i \in \mathcal{G} \cap S_k$ in parallel **do**
7:          $x^{k+1} = x^k - \gamma g^k$ and $\lambda_{k+1} = \alpha_{k+1} \|x^{k+1} - x^k\|$
8:          Set $g_i^{k+1} = \begin{cases} \nabla f_i(x^{k+1}), & \text{if } c_k = 1, \\ g^k + \text{clip}_{\lambda_{k+1}}\left(\mathcal{Q}\left(\widehat{\Delta}_i(x^{k+1}, x^k)\right)\right), & \text{otherwise,} \end{cases}$

         where $\widehat{\Delta}_i(x^{k+1}, x^k)$ is a mini-batched estimator of $\nabla f_i(x^{k+1}) - \nabla f_i(x^k)$, $\mathcal{Q}(\cdot)$ for $i \in \mathcal{G} \cap S_k$ are computed independently
9:      **end for**
10:    $g^{k+1} = \begin{cases} \text{ARAgg}\left(\{g_i^{k+1}\}_{i \in S_k}\right), & \text{if } c_k = 1, \\ g^k + \frac{1}{C}\sum\limits_{i \in S_k} \text{clip}_{\lambda_{k+1}}\left(\mathcal{Q}\left(\widehat{\Delta}_i(x^{k+1}, x^k)\right)\right), & \text{otherwise} \end{cases}$
11: **end for**

---

The key idea behind this modification can be explained as follows. For simplicity, let us assume that $C$ is small and $\delta_{\text{real}}$ is also small. Then, for the communication rounds with $c_k = 0$, with a large probability, only good clients will be sampled. In this case, the method can use just an average of the received vectors and benefit from the lack of bias appearing due to the robust aggregation. Moreover, when $c_k = 0$ and at least one of the sampled clients is Byzantine, the method will tolerate due to the clipping. That is, when $C$ is small, the method can potentially benefit from the lack of robust aggregation when $c_k = 0$. However, for the rounds with $c_k = 1$, in the worst case, $\widehat{C} = n$, meaning that all Byzantines workers are guaranteed to be sampled. To tolerate such situations, we keep the robust aggregation in the method when $c_k = 1$.

## F.1  Analysis for Bounded Compressors

For simplicity, we analyze Byz-VR-MARINA-PP+ for bounded compressors only. The analysis is very similar to the one we provide for Byz-VR-MARINA-PP, but several steps are significantly simpler. In particular, the central part in the analysis of Byz-VR-MARINA-PP is in deriving a good recursive inequality for $\mathbb{E}[\|g^k - \nabla f(x^k)\|^2]$, which requires several quite technical steps. For Byz-VR-MARINA-PP+, one can obtain a similar inequality much easier as shown in the next lemma.

**Lemma F.1.** *Let Assumptions D.1, D.2, D.3, D.4, D.5, 2.4 hold and the compression operator satisfy Definition 2.2. Assume that $C \leq G$. We set $\lambda_{k+1} = D_Q \max_{i,j} L_{i,j}\|x^{k+1} - x^k\|$. Then for all $k \geq 0$ the iterates produced by Byz-VR-MARINA-PP+ (Algorithm 3) satisfy*

$$\mathbb{E}\left[\left\|g^{k+1} - \nabla f\left(x^{k+1}\right)\right\|^2\right] \leq \left(1 - \frac{p}{2}\right)\mathbb{E}\left[\left\|g^k - \nabla f\left(x^k\right)\right\|^2\right] \tag{30}$$

$$+ \widehat{B}\,\mathbb{E}\left[\left\|\nabla f\left(x^k\right)\right\|^2\right] + \widehat{D}\zeta^2 + \frac{pA}{4}\|x^{k+1} - x^k\|^2,$$

*with*

$$A = \frac{4}{p} \left( \frac{8pBG\mathcal{P}_{\mathcal{G}_{\widehat{C}}^k} c\delta}{(1-\delta)\widehat{C}} + 6p\widetilde{B} + \frac{(1-p)p_{\mathcal{G}}^k \omega}{C} + \frac{6(1-p)(1-p_{\mathcal{G}}^k)}{p} \right) L^2$$

$$+ \frac{4(1-p)p_{\mathcal{G}}^k}{p} \left( 1 + \frac{\omega}{C} \right) L_{\pm}^2 + \frac{4(1-p)p_{\mathcal{G}}^k(1+\omega)}{pC} \frac{\mathcal{L}_{\pm}^2}{b}$$

$$+ \frac{24(1-p)(1-p_{\mathcal{G}}^k)}{p^2} \left( D_Q \max_{i,j} L_{i,j} \right)^2,$$

$$\widehat{B} = \frac{8G\mathcal{P}_{\mathcal{G}_{\widehat{C}}^k} c\delta Bp}{(1-\delta)\widehat{C}} + 6p\widetilde{B}, \quad \widehat{D} = \frac{4G\mathcal{P}_{\mathcal{G}_{\widehat{C}}^k} c\delta p}{(1-\delta)\widehat{C}} + p\widetilde{D},$$

*where $\widetilde{B}$, $\widetilde{D}$, $p_{\mathcal{G}}^k$, $\mathcal{P}_{\mathcal{G}_{\widehat{C}}^k}$ are defined in Lemma D.13.*

*Proof.* From the update rule of $g^{k+1}$, we have

$$\mathbb{E}\left[ \left\| g^{k+1} - \nabla f(x^{k+1}) \right\|^2 \right] = p \underbrace{\mathbb{E}\left[ \left\| \mathtt{ARAgg}\left( \{g_i^{k+1}\}_{i \in S_k} \right) - \nabla f(x^{k+1}) \right\|^2 \mid c_k = 1 \right]}_{T_1} \qquad (31)$$

$$+ (1-p) \underbrace{\mathbb{E}\left[ \left\| g^k + \frac{1}{C} \sum_{i \in S_k} \mathtt{clip}_{\lambda_{k+1}}\left( \mathcal{Q}\left( \widehat{\Delta}_i(x^{k+1}, x^k) \right) \right) - \nabla f(x^{k+1}) \right\|^2 \mid c_k = 0 \right]}_{T_2}.$$

Next, we bound $T_1$ and $T_2$ separately. From Lemma D.10, we have

$$T_1 \leq \left( \frac{8G\mathcal{P}_{\mathcal{G}_{\widehat{C}}^k} c\delta B}{(1-\delta)\widehat{C}} + 2\widetilde{B} \right) \mathbb{E}\left[ \left\| \nabla f\left( x^k \right) \right\|^2 + L^2 \left\| x^{k+1} - x^k \right\|^2 \right] + \frac{4G\mathcal{P}_{\mathcal{G}_{\widehat{C}}^k} c\delta\zeta^2}{(1-\delta)\widehat{C}} + \widetilde{\zeta}^2.$$

As for $T_2$, we consider two possible situations: either $S_k \cap \mathcal{B} = \varnothing$ (no Byzantine workers are among sampled ones) or $S_k \cap \mathcal{B} \neq \varnothing$ (at least one Byzantine worker is sampled). Then, $T_2$ equals

$$T_2 = p_{\mathcal{G}}^k \underbrace{\mathbb{E}\left[ \left\| g^k + \frac{1}{C} \sum_{i \in S_k} \mathtt{clip}_{\lambda_{k+1}}\left( \mathcal{Q}\left( \widehat{\Delta}_i(x^{k+1}, x^k) \right) \right) - \nabla f(x^{k+1}) \right\|^2 \mid c_k = 0, S_k \cap \mathcal{B} = \varnothing \right]}_{\widehat{T}_2}$$

$$+ (1 - p_{\mathcal{G}}^k) \underbrace{\mathbb{E}\left[ \left\| g^k + \frac{1}{C} \sum_{i \in S_k} \mathtt{clip}_{\lambda_{k+1}}\left( \mathcal{Q}\left( \widehat{\Delta}_i(x^{k+1}, x^k) \right) \right) - \nabla f(x^{k+1}) \right\|^2 \mid c_k = 0, S_k \cap \mathcal{B} \neq \varnothing \right]}_{\widetilde{T}_2},$$

where

$$p_{\mathcal{G}}^k := \mathrm{Prob}\{S_k \cap \mathcal{B} = \varnothing \mid c_k = 0\} = \frac{\binom{G}{C}}{\binom{n}{C}} = \frac{(G-C+1)(G-C+2)\cdot\ldots\cdot(n-C)}{(G+1)(G+2)\cdot\ldots\cdot n}.$$

The choice of the clipping level $\lambda_{k+1} = D_Q \max_{i,j} L_{i,j} \| x^{k+1} - x^k \|$ and inequality (29) imply that $\mathtt{clip}_{\lambda_{k+1}}\left( \mathcal{Q}\left( \widehat{\Delta}_i(x^{k+1}, x^k) \right) \right)$ for all $i \in \mathcal{G}$. Therefore, for $\widehat{T}_2$, we have

$$\widehat{T}_2 = \mathbb{E}\left[ \left\| g^k + \frac{1}{C} \sum_{i \in S_k} \mathcal{Q}\left( \widehat{\Delta}_i(x^{k+1}, x^k) \right) - \nabla f(x^{k+1}) \right\|^2 \mid c_k = 0, S_k \cap \mathcal{B} = \varnothing \right]$$

$$= \mathbb{E}\left[ \| g^k - \nabla f(x^k) \|^2 \right]$$

$$+ \mathbb{E}\left[ \left\| \frac{1}{C} \sum_{i \in S_k} \mathcal{Q}\left( \widehat{\Delta}_i(x^{k+1}, x^k) \right) - (\nabla f(x^{k+1}) - \nabla f(x^k)) \right\|^2 \mid c_k = 0, S_k \cap \mathcal{B} = \varnothing \right],$$

where we use that $\mathbb{E}\left[\frac{1}{C}\sum_{i\in S_k}\mathcal{Q}\left(\widehat{\Delta}_i(x^{k+1},x^k)\right)\mid c_k=0, S_k\cap\mathcal{B}=\varnothing\right] = \nabla f(x^{k+1}) - \nabla f(x^k)$. Moreover, since $\mathbb{E}\left[\frac{1}{C}\sum_{i\in S_k}\mathcal{Q}\left(\widehat{\Delta}_i(x^{k+1},x^k)\right)\mid c_k=0, S_k\cap\mathcal{B}=\varnothing, S_k\right] = \frac{1}{C}\sum_{i\in S_k}(\nabla f_i(x^{k+1}) - \nabla f_i(x^k)) =: \frac{1}{C}\sum_{i\in S_k}\Delta_i(x^{k+1},x^k)$, we can decompose the last term in the upper-bound for $\widehat{T}_2$ as follows:

$$
\begin{aligned}
\widehat{T}_2 &= \mathbb{E}\left[\|g^k - \nabla f(x^k)\|^2\right] \\
&+ \mathbb{E}\left[\mathbb{E}\left[\left\|\frac{1}{C}\sum_{i\in S_k}\left(\mathcal{Q}\left(\widehat{\Delta}_i(x^{k+1},x^k)\right) - \Delta_i(x^{k+1},x^k)\right)\right\|^2 \mid S_k\right] \mid c_k=0, S_k\cap\mathcal{B}=\varnothing\right] \\
&+ \mathbb{E}\left[\left\|\frac{1}{C}\sum_{i\in S_k}\Delta_i(x^{k+1},x^k) - (\nabla f(x^{k+1}) - \nabla f(x^k))\right\|^2 \mid c_k=0, S_k\cap\mathcal{B}=\varnothing\right].
\end{aligned}
$$

Since the compression operator computations are independent on each client, we have

$$
\begin{aligned}
\widehat{T}_2 &= \mathbb{E}\left[\|g^k - \nabla f(x^k)\|^2\right] \\
&+ \frac{1}{C^2}\mathbb{E}\left[\mathbb{E}\left[\sum_{i\in S_k}\left\|\mathcal{Q}\left(\widehat{\Delta}_i(x^{k+1},x^k)\right) - \Delta_i(x^{k+1},x^k)\right\|^2 \mid S_k\right] \mid c_k=0, S_k\cap\mathcal{B}=\varnothing\right] \\
&+ \mathbb{E}\left[\left\|\frac{1}{C}\sum_{i\in S_k}\Delta_i(x^{k+1},x^k)\right\|^2 \mid c_k=0, S_k\cap\mathcal{B}=\varnothing\right] - \mathbb{E}\left[\|\nabla f(x^{k+1}) - \nabla f(x^k)\|^2\right] \\
&\leq \mathbb{E}\left[\|g^k - \nabla f(x^k)\|^2\right] \\
&+ \frac{1}{C^2}\mathbb{E}\left[\mathbb{E}\left[\sum_{i\in S_k}\left\|\mathcal{Q}\left(\widehat{\Delta}_i(x^{k+1},x^k)\right) - \widehat{\Delta}_i(x^{k+1},x^k)\right\|^2 \mid S_k\right] \mid c_k=0, S_k\cap\mathcal{B}=\varnothing\right] \\
&+ \frac{1}{C^2}\mathbb{E}\left[\mathbb{E}\left[\sum_{i\in S_k}\left\|\widehat{\Delta}_i(x^{k+1},x^k) - \Delta_i(x^{k+1},x^k)\right\|^2 \mid S_k\right] \mid c_k=0, S_k\cap\mathcal{B}=\varnothing\right] \\
&+ \mathbb{E}\left[\left\|\frac{1}{C}\sum_{i\in S_k}\Delta_i(x^{k+1},x^k)\right\|^2 \mid c_k=0, S_k\cap\mathcal{B}=\varnothing\right] - \mathbb{E}\left[\|\nabla f(x^{k+1}) - \nabla f(x^k)\|^2\right] \\
&\overset{\text{(Def. 2.2)}}{\leq} \mathbb{E}\left[\|g^k - \nabla f(x^k)\|^2\right] \\
&+ \frac{\omega}{C^2}\mathbb{E}\left[\sum_{i\in S_k}\left\|\widehat{\Delta}_i(x^{k+1},x^k)\right\|^2 \mid c_k=0, S_k\cap\mathcal{B}=\varnothing\right] \\
&+ \frac{1}{C^2}\mathbb{E}\left[\sum_{i\in S_k}\left\|\widehat{\Delta}_i(x^{k+1},x^k) - \Delta_i(x^{k+1},x^k)\right\|^2 \mid c_k=0, S_k\cap\mathcal{B}=\varnothing\right] \\
&+ \frac{1}{C}\mathbb{E}\left[\sum_{i\in S_k}\left\|\Delta_i(x^{k+1},x^k)\right\|^2 \mid c_k=0, S_k\cap\mathcal{B}=\varnothing\right] - \mathbb{E}\left[\|\nabla f(x^{k+1}) - \nabla f(x^k)\|^2\right] \\
&= \mathbb{E}\left[\|g^k - \nabla f(x^k)\|^2\right] + \frac{\omega}{CG}\sum_{i\in\mathcal{G}}\mathbb{E}\left[\left\|\widehat{\Delta}_i(x^{k+1},x^k)\right\|^2\right] \\
&+ \frac{1}{CG}\sum_{i\in\mathcal{G}}\mathbb{E}\left[\left\|\widehat{\Delta}_i(x^{k+1},x^k) - \Delta_i(x^{k+1},x^k)\right\|^2\right] \\
&+ \frac{1}{G}\sum_{i\in\mathcal{G}}\mathbb{E}\left[\left\|\Delta_i(x^{k+1},x^k)\right\|^2\right] - \mathbb{E}\left[\|\nabla f(x^{k+1}) - \nabla f(x^k)\|^2\right].
\end{aligned}
$$

Using $\mathbb{E}\left[\left\|\widehat{\Delta}_i(x^{k+1}, x^k)\right\|^2\right] = \mathbb{E}\left[\left\|\widehat{\Delta}_i(x^{k+1}, x^k) - \Delta_i(x^{k+1}, x^k)\right\|^2\right] + \mathbb{E}\left[\left\|\Delta_i(x^{k+1}, x^k)\right\|^2\right]$,
we continue the derivation as follows:

$$\widehat{T}_2 \leq \mathbb{E}\left[\|g^k - \nabla f(x^k)\|^2\right] + \frac{1+\omega}{CG} \sum_{i \in \mathcal{G}} \mathbb{E}\left[\left\|\widehat{\Delta}_i(x^{k+1}, x^k) - \Delta_i(x^{k+1}, x^k)\right\|^2\right]$$

$$+ \left(1 + \frac{\omega}{C}\right) \frac{1}{G} \sum_{i \in \mathcal{G}} \mathbb{E}\left[\|\Delta_i(x^{k+1}, x^k)\|^2\right] - \mathbb{E}\left[\|\nabla f(x^{k+1}) - \nabla f(x^k)\|^2\right]$$

$$\overset{(21)}{\leq} \mathbb{E}\left[\|g^k - \nabla f(x^k)\|^2\right] + \frac{(1+\omega)\mathcal{L}_{\pm}^2}{bC} \mathbb{E}\left[\|x^{k+1} - x^k\|^2\right]$$

$$+ \left(1 + \frac{\omega}{C}\right) \mathbb{E}\left[\frac{1}{G} \sum_{i \in \mathcal{G}} \|\Delta_i(x^{k+1}, x^k)\|^2 - \|\nabla f(x^{k+1}) - \nabla f(x^k)\|^2\right]$$

$$+ \frac{\omega}{C} \mathbb{E}\left[\|\nabla f(x^{k+1}) - \nabla f(x^k)\|^2\right]$$

$$\overset{(20),(18)}{\leq} \mathbb{E}\left[\|g^k - \nabla f(x^k)\|^2\right] + \left(\frac{\omega}{C}L^2 + \left(1 + \frac{\omega}{C}\right)L_{\pm}^2 + \frac{(1+\omega)\mathcal{L}_{\pm}^2}{bC}\right) \mathbb{E}\left[\|x^{k+1} - x^k\|^2\right].$$

Next, we estimate $\widetilde{T}_2$ using Young's inequality and the choice of the clipping level:

$$\widetilde{T}_2 \leq (1+\beta)\mathbb{E}\left[\|g^k - \nabla f(x^k)\|^2\right] + 2(1+\beta^{-1})\mathbb{E}\left[\|\nabla f(x^{k+1}) - \nabla f(x^k)\|^2\right]$$

$$+ 2(1+\beta^{-1})\mathbb{E}\left[\left\|\frac{1}{C} \sum_{i \in S_k} \texttt{clip}_{\lambda_{k+1}}\left(\mathcal{Q}\left(\widehat{\Delta}_i(x^{k+1}, x^k)\right)\right)\right\|^2 \mid c_k = 0, S_k \cap \mathcal{B} \neq \varnothing\right]$$

$$\overset{(18)}{\leq} (1+\beta)\mathbb{E}\left[\|g^k - \nabla f(x^k)\|^2\right] + 2(1+\beta^{-1})\left(L^2\mathbb{E}\left[\|x^{k+1} - x^k\|^2\right] + \mathbb{E}[\lambda_{k+1}^2]\right)$$

$$= (1+\beta)\mathbb{E}\left[\|g^k - \nabla f(x^k)\|^2\right] + 2(1+\beta^{-1})\left(L^2 + D_Q^2 \max_{i,j} L_{i,j}^2\right) \mathbb{E}\left[\|x^{k+1} - x^k\|^2\right],$$

where $\beta > 0$ will be specified later in the proof. Combining the derived upper bounds for $\widehat{T}_2$ and $\widetilde{T}_2$, we get

$$T_2 \leq \left(p_{\mathcal{G}}^k + (1-p_{\mathcal{G}}^k)(1+\beta)\right) \mathbb{E}\left[\|g^k - \nabla f(x^k)\|^2\right]$$

$$+ p_{\mathcal{G}}^k \left(\frac{\omega}{C}L^2 + \left(1 + \frac{\omega}{C}\right)L_{\pm}^2 + \frac{(1+\omega)\mathcal{L}_{\pm}^2}{bC}\right) \mathbb{E}\left[\|x^{k+1} - x^k\|^2\right]$$

$$+ 2(1-p_{\mathcal{G}}^k)(1+\beta^{-1})\left(L^2 + D_Q^2 \max_{i,j} L_{i,j}^2\right) \mathbb{E}\left[\|x^{k+1} - x^k\|^2\right].$$

Plugging the obtained bounds for $T_1$ and $T_2$ into (31), we obtain

$$\mathbb{E}\left[\|g^{k+1} - \nabla f(x^{k+1})\|^2\right] \leq (1-p)\left(p_{\mathcal{G}}^k + (1-p_{\mathcal{G}}^k)(1+\beta)\right) \mathbb{E}\left[\|g^k - \nabla f(x^k)\|^2\right]$$

$$+ p\left(\left(\frac{8G\mathcal{P}_{\mathcal{G}_{\widehat{C}}^k} c\delta B}{(1-\delta)\widehat{C}} + 2\widetilde{B}\right) \mathbb{E}\left[\|\nabla f(x^k)\|^2 + L^2\|x^{k+1} - x^k\|^2\right] + \frac{4G\mathcal{P}_{\mathcal{G}_{\widehat{C}}^k} c\delta\zeta^2}{(1-\delta)\widehat{C}} + \widetilde{\zeta}^2\right)$$

$$+ (1-p)p_{\mathcal{G}}^k \left(\frac{\omega}{C}L^2 + \left(1 + \frac{\omega}{C}\right)L_{\pm}^2 + \frac{(1+\omega)\mathcal{L}_{\pm}^2}{bC}\right) \mathbb{E}\left[\|x^{k+1} - x^k\|^2\right]$$

$$+ 2(1-p)(1-p_{\mathcal{G}}^k)(1+\beta^{-1})\left(L^2 + D_Q^2 \max_{i,j} L_{i,j}^2\right) \mathbb{E}\left[\|x^{k+1} - x^k\|^2\right].$$

Taking

$$\beta := \begin{cases} \frac{p}{2(1-p_{\mathcal{G}}^k)}, & \text{if } p_{\mathcal{G}}^k < 1, \\ 1, & \text{if } p_{\mathcal{G}}^k = 1, \end{cases}$$

we ensure that $p_{\mathcal{G}}^k + (1-p_{\mathcal{G}}^k)(1+\beta) \leq 1 + \frac{p}{2}$ and $(1-p_{\mathcal{G}}^k)(1+\beta^{-1}) \leq \frac{(1-p_{\mathcal{G}}^k)(p+2(1-p_{\mathcal{G}}^k))}{p} \leq \frac{3(1-p_{\mathcal{G}}^k)}{p}$. Using these inequalities and $(1-p)\left(1 - \frac{p}{2}\right) \leq 1 - \frac{p}{2}$, we simplify the upper bound for

$\mathbb{E}\left[\|g^{k+1} - \nabla f(x^{k+1})\|^2\right]$ as follows:

$$\mathbb{E}\left[\|g^{k+1} - \nabla f(x^{k+1})\|^2\right] \leq \left(1 - \frac{p}{2}\right)\mathbb{E}\left[\|g^k - \nabla f(x^k)\|^2\right]$$

$$+ p\left(\left(\frac{8G\mathcal{P}_{\mathcal{G}_{\widehat{C}}^k}c\delta B}{(1-\delta)\widehat{C}} + 2\widetilde{B}\right)\mathbb{E}\left[\left\|\nabla f\left(x^k\right)\right\|^2 + L^2\left\|x^{k+1} - x^k\right\|^2\right] + \frac{4G\mathcal{P}_{\mathcal{G}_{\widehat{C}}^k}c\delta\zeta^2}{(1-\delta)\widehat{C}} + \widetilde{\zeta}^2\right)$$

$$+ (1-p)p_{\mathcal{G}}^k\left(\frac{\omega}{C}L^2 + \left(1 + \frac{\omega}{C}\right)L_{\pm}^2 + \frac{(1+\omega)\mathcal{L}_{\pm}^2}{bC}\right)\mathbb{E}\left[\|x^{k+1} - x^k\|^2\right]$$

$$+ \frac{6(1-p)(1-p_{\mathcal{G}}^k)}{p}\left(L^2 + D_Q^2\max_{i,j}L_{i,j}^2\right)\mathbb{E}\left[\|x^{k+1} - x^k\|^2\right].$$

Rearranging the terms, we get (35). $\qquad\square$

Then, similarly to the analysis of Byz-VR-MARINA, we get the following result.

**Theorem F.2.** *Let Assumptions D.1, D.2, D.3, D.4, D.5, 2.4 hold. Set $\lambda_{k+1} = \max_{i,j}L_{i,j}\left\|x^{k+1} - x^k\right\|$. Assume that*

$$0 < \gamma \leq \frac{1}{L + \sqrt{A}}, \quad 4\widehat{B} < p,$$

*where*

$$A = \frac{4}{p}\left(\frac{8pBG\mathcal{P}_{\mathcal{G}_{\widehat{C}}^k}c\delta}{(1-\delta)\widehat{C}} + 6p\widetilde{B} + \frac{(1-p)p_{\mathcal{G}}^k\omega}{C} + \frac{6(1-p)(1-p_{\mathcal{G}}^k)}{p}\right)L^2$$

$$+ \frac{4(1-p)p_{\mathcal{G}}^k}{p}\left(1 + \frac{\omega}{C}\right)L_{\pm}^2 + \frac{4(1-p)p_{\mathcal{G}}^k(1+\omega)}{pC}\frac{\mathcal{L}_{\pm}^2}{b}$$

$$+ \frac{24(1-p)(1-p_{\mathcal{G}}^k)}{p^2}\left(D_Q\max_{i,j}L_{i,j}\right)^2,$$

$$\widehat{B} = \frac{8G\mathcal{P}_{\mathcal{G}_{\widehat{C}}^k}c\delta Bp}{(1-\delta)\widehat{C}} + 6p\widetilde{B}, \quad \widehat{D} = \frac{4G\mathcal{P}_{\mathcal{G}_{\widehat{C}}^k}c\delta p}{(1-\delta)\widehat{C}} + p\widetilde{D},$$

*and*

$$\mathcal{P}_{\mathcal{G}_C^k} = \frac{C}{np_G}\cdot\sum_{(1-\delta)C\leq t\leq C}\left(\binom{G-1}{t-1}\binom{n-G}{C-t}\left(\binom{n}{C}\right)^{-1}\right),$$

$$p_{\mathcal{G}}^k = \text{Prob}\{S_k \cap \mathcal{B} = \varnothing \mid c_k = 0\} = \frac{(G-C+1)(G-C+2)\cdot\ldots\cdot(n-C)}{(G+1)(G+2)\cdot\ldots\cdot n}.$$

*Then for all $K \geq 0$ the iterates produced by Byz-VR-MARINA+ (Algorithm 3) satisfy*

$$\mathbb{E}\left[\left\|\nabla f\left(\widehat{x}^K\right)\right\|^2\right] \leq \frac{2\Phi^0}{\gamma\left(1 - \frac{4\widehat{B}}{p}\right)(K+1)} + \frac{2\widehat{D}\zeta^2}{p - 4\widehat{B}},$$

*where $\widehat{x}^K$ is chosen uniformly at random from $x^0, x^1, \ldots, x^K$, and $\Phi^0 = f\left(x^0\right) - f^* + \frac{\gamma}{p}\left\|g^0 - \nabla f\left(x^0\right)\right\|^2$.*

*Proof.* The proof is analogous to the proof of Theorem D.14. $\qquad\square$

**Theorem F.3.** *Let Assumptions 2.4, D.1, D.2, D.3, D.4, D.5, 2.7 hold. Set $\lambda_{k+1} = \max_{i,j}L_{i,j}\left\|x^{k+1} - x^k\right\|$. Assume that*

$$0 < \gamma \leq \min\left\{\frac{1}{L + \sqrt{2A}}\right\}, \quad 8\widehat{B} < p,$$

*where*

$$A = \frac{4}{p} \left( \frac{8pBG\mathcal{P}_{\mathcal{G}_{\widehat{C}}^k} c\delta}{(1-\delta)\widehat{C}} + 6p\widetilde{B} + \frac{(1-p)p_{\mathcal{G}}^k \omega}{C} + \frac{6(1-p)(1-p_{\mathcal{G}}^k)}{p} \right) L^2$$

$$+ \frac{4(1-p)p_{\mathcal{G}}^k}{p} \left( 1 + \frac{\omega}{C} \right) L_{\pm}^2 + \frac{4(1-p)p_{\mathcal{G}}^k(1+\omega)}{pC} \frac{\mathcal{L}_{\pm}^2}{b}$$

$$+ \frac{24(1-p)(1-p_{\mathcal{G}}^k)}{p^2} \left( D_Q \max_{i,j} L_{i,j} \right)^2,$$

$$\widehat{B} = \frac{8G\mathcal{P}_{\mathcal{G}_{\widehat{C}}^k} c\delta Bp}{(1-\delta)\widehat{C}} + 6p\widetilde{B}, \quad \widehat{D} = \frac{4G\mathcal{P}_{\mathcal{G}_{\widehat{C}}^k} c\delta p}{(1-\delta)\widehat{C}} + p\widetilde{D},$$

*and*

$$\mathcal{P}_{\mathcal{G}_C^k} = \frac{C}{np_G} \cdot \sum_{(1-\delta)C \leq t \leq C} \left( \binom{G-1}{t-1} \binom{n-G}{C-t} \left( \binom{n}{C} \right)^{-1} \right),$$

$$p_{\mathcal{G}}^k = \text{Prob}\{S_k \cap \mathcal{B} = \varnothing \mid c_k = 0\} = \frac{(G-C+1)(G-C+2)\cdot\ldots\cdot(n-C)}{(G+1)(G+2)\cdot\ldots\cdot n}.$$

*Then for all $K \geq 0$ the iterates produced by* Byz-VR-MARINA+ *(Algorithm 3) satisfy*

$$\mathbb{E}\left[ f\left(x^K\right) - f\left(x^*\right) \right] \leq (1-\rho)^K \Phi^0 + \frac{2\widehat{D}\zeta^2}{p\rho},$$

*where $\rho = \min\left[ \gamma\mu \left( 1 - \frac{8\widehat{B}}{p} \right), \frac{p}{4} \right]$ and $\Phi^0 = f\left(x^0\right) - f^* + \frac{2\gamma}{p} \left\| g^0 - \nabla f\left(x^0\right) \right\|^2$.*

*Proof.* The proof is analogous to the proof of Theorem D.15. $\square$

## F.2 Discussion of the Results

**Improved neighborhood term and bound on $\delta$.** The key property of Byz-VR-MARINA+ is its better neighborhood terms, and maximal allowed fraction of Byzantine workers $\delta$ in comparison to Byz-VR-MARINA. To illustrate it, consider the non-PŁsetting (the discussion for the PŁ case is similar). For both algorithms, the neighborhood term in the convergence bounds equals $\mathcal{O}\left( \frac{\widehat{D}\zeta^2}{p-4\widehat{B}} \right)$, but corresponding constants $\widehat{B}$ and $\widehat{D}$ are different:

$$\widehat{B} = 2\frac{\delta\mathcal{P}_{\mathcal{G}_{\widehat{C}}^k}}{1-\delta} B \left( \frac{12cG}{\widehat{C}} + p \right) + 6\widetilde{B} \quad \text{and} \quad \widehat{D} = 2\frac{\delta\mathcal{P}_{\mathcal{G}_{\widehat{C}}^k}}{1-\delta} \left( \frac{6cG}{\widehat{C}} + p \right) + \widetilde{D} \quad \text{for} \quad \text{Byz-VR-MARINA},$$

$$\widehat{B} = \frac{8G\mathcal{P}_{\mathcal{G}_{\widehat{C}}^k} c\delta Bp}{(1-\delta)\widehat{C}} + 6p\widetilde{B} \quad \text{and} \quad \widehat{D} = \frac{4G\mathcal{P}_{\mathcal{G}_{\widehat{C}}^k} c\delta p}{(1-\delta)\widehat{C}} + p\widetilde{D} \quad \text{for} \quad \text{Byz-VR-MARINA+}.$$

For the simplicity of the comparison, consider the case of $\widehat{C} = n$. Then, $\mathcal{P}_{\mathcal{G}_{\widehat{C}}^k} = 1$ and

$$\widehat{B} = \Theta\left(c\delta B\right) \quad \text{and} \quad \widehat{D} = \Theta(c\delta) \quad \text{for} \quad \text{Byz-VR-MARINA},$$

$$\widehat{B} = \Theta\left(c\delta Bp\right) \quad \text{and} \quad \widehat{D} = \Theta(c\delta p) \quad \text{for} \quad \text{Byz-VR-MARINA+},$$

implying that the neighborhood term for Byz-VR-MARINA+ is $1/p$ times smaller than the neighborhood term for Byz-VR-MARINA. Moreover, the restriction $4\widehat{B} < p$ used in the analysis of both methods implies

$$c\delta = \mathcal{O}\left( \frac{1}{Bp} \right) \quad \text{for} \quad \text{Byz-VR-MARINA},$$

$$c\delta = \mathcal{O}\left( \frac{1}{B} \right) \quad \text{for} \quad \text{Byz-VR-MARINA+},$$

i.e., the result for Byz-VR-MARINA+ allows $(1/p)$-times more Byzantine workers when $B > 0$. We emphasize that the neighborhood term and the bound on $\delta$ in the results for Byz-VR-MARINA+ cannot be improved up to the numerical factors (Allouah et al., 2024b).

**Comparison of stepsizes when $\widehat{C} = n$ and $C = 1$.** For simplicity, to compare the stepsize restrictions for Byz-VR-MARINA and Byz-VR-MARINA+, we consider the case when $\widehat{C} = n$ and $C = 1$. Moreover, let us assume that $b = 1$ and let us ignore the differences between smoothness constants and replace them with their upper bound $\mathcal{L}$ from Assumption 2.6. Then, for both methods, the results in the non-PŁsetting (the discussion for the PŁ case is similar) with $B = 0$ hold for $0 < \gamma \leq 1/\mathcal{L}(1+\sqrt{A})$, where

$$A = \Theta\left(\frac{1}{p}\left(1 + \omega + \frac{(1+\omega)c\delta}{p}\right) + \frac{\delta_{\text{real}}(1 + F_{\mathcal{A}}^2 D_Q^2)}{p^2}\right) \quad \text{for} \quad \text{Byz-VR-MARINA},$$

$$A = \Theta\left(\frac{1+\omega}{p} + \frac{\delta_{\text{real}} D_Q^2}{p^2}\right) \quad \text{for} \quad \text{Byz-VR-MARINA+},$$

where we use $p_G = G/n = 1 - \delta_{\text{real}}$, $\mathcal{P}_{\mathcal{G}_C^k} = 1/G$, $p_{\mathcal{G}}^k = G/n = 1 - \delta_{\text{real}}$. That is, the result for Byz-VR-MARINA+ allows to use larger stepsizes than in Byz-VR-MARINA (though the methods are equivalent when $C = 1$). A similar comparison holds for small enough $C$ as well. Therefore, we recommend using Byz-VR-MARINA+ instead of Byz-VR-MARINA when $C$ is small. We also highlight that the result for Byz-VR-MARINA+ does not require Assumption 2.3.

# G  Analysis without Full-Batch Gradient Computations

In this section, we consider versions of Byz-VR-MARINA-PP and Byz-VR-MARINA-PP+ that do not use full-batch gradient computations at all – see Algorithms 4 and 5. These variants of Byz-VR-MARINA-PP and Byz-VR-MARINA-PP+ use $b'$-size mini-batched estimator $\widetilde{\nabla} f_i(x^{k+1})$ when $c_k = 1$ for every $i \in \mathcal{G} \cap S_k$ in line 8 and are identical to their original versions in all other steps/computations. This modification reduces the computation cost of iterations when $c_k = 1$, making the methods more practical.

---

**Algorithm 4** Byz-VR-MARINA-PP without full-batch gradient computations

1: **Input:** vectors $x^0, g^0 \in \mathbb{R}^d$, stepsize $\gamma$, mini-batch size $b$, mini-batch size $b'$, probability $p \in (0, 1]$, number of iterations $K$, $(\delta, c)$-ARAgg, clients' sample size $1 \leq C \leq \widehat{C} \leq n$, clipping coefficients $\{\alpha_k\}_{k \geq 1}$
2: **for** $k = 0, 1, \ldots, K - 1$ **do**
3:     Get a sample from Bernoulli distribution with parameter $p$: $c_k \sim \text{Be}(p)$
4:     Sample the set of clients $S_k \subseteq [n]$, $|S_k| = C$ if $c_k = 0$; otherwise $|S_k| = \widehat{C}$
5:     Broadcast $g^k, c_k$ to all workers
6:     **for** $i \in \mathcal{G} \cap S_k$ in parallel **do**
7:         $x^{k+1} = x^k - \gamma g^k$ and $\lambda_{k+1} = \alpha_{k+1} \|x^{k+1} - x^k\|$
8:         Set $g_i^{k+1} = \begin{cases} \widetilde{\nabla} f_i(x^{k+1}), & \text{if } c_k = 1, \\ g^k + \text{clip}_{\lambda_{k+1}} \left( \mathcal{Q} \left( \widehat{\Delta}_i(x^{k+1}, x^k) \right) \right), & \text{otherwise}, \end{cases}$

        where $\widetilde{\nabla} f_i(x^{k+1})$ is a $b'$-size mini-batched estimator of $\nabla f_i(x^{k+1})$, $\widehat{\Delta}_i(x^{k+1}, x^k)$ is a $b$-size mini-batched estimator of $\nabla f_i(x^{k+1}) - \nabla f_i(x^k)$, $\mathcal{Q}(\cdot)$ for $i \in \mathcal{G} \cap S_k$ are computed independently
9:     **end for**
10:    $g^{k+1} = \begin{cases} \text{ARAgg} \left( \{g_i^{k+1}\}_{i \in S_k} \right), & \text{if } c_k = 1, \\ g^k + \text{ARAgg} \left( \left\{ \text{clip}_{\lambda_{k+1}} \left( \mathcal{Q} \left( \widehat{\Delta}_i(x^{k+1}, x^k) \right) \right) \right\}_{i \in S_k} \right), & \text{otherwise} \end{cases}$
11: **end for**

---

**Algorithm 5** Byz-VR-MARINA-PP+: without full-batch gradient computations

1: **Input:** vectors $x^0, g^0 \in \mathbb{R}^d$, stepsize $\gamma$, mini-batch size $b$, mini-batch size $\widehat{b'}$, probability $p \in (0, 1]$, number of iterations $K$, $(\delta, c)$-ARAgg, clients' sample size $1 \leq C \leq \widehat{C} \leq n$, clipping coefficients $\{\alpha_k\}_{k \geq 1}$
2: **for** $k = 0, 1, \ldots, K - 1$ **do**
3:     Get a sample from Bernoulli distribution with parameter $p$: $c_k \sim \text{Be}(p)$
4:     Sample the set of clients $S_k \subseteq [n]$, $|S_k| = C$ if $c_k = 0$; otherwise $|S_k| = \widehat{C}$
5:     Broadcast $g^k, c_k$ to all workers
6:     **for** $i \in \mathcal{G} \cap S_k$ in parallel **do**
7:         $x^{k+1} = x^k - \gamma g^k$ and $\lambda_{k+1} = \alpha_{k+1} \|x^{k+1} - x^k\|$
8:         Set $g_i^{k+1} = \begin{cases} \widetilde{\nabla} f_i(x^{k+1}), & \text{if } c_k = 1, \\ g^k + \text{clip}_{\lambda_{k+1}} \left( \mathcal{Q} \left( \widehat{\Delta}_i(x^{k+1}, x^k) \right) \right), & \text{otherwise}, \end{cases}$

        where $\widetilde{\nabla} f_i(x^{k+1})$ is a $b'$-size mini-batched estimator of $\nabla f_i(x^{k+1})$, $\widehat{\Delta}_i(x^{k+1}, x^k)$ is a $b$-size mini-batched estimator of $\nabla f_i(x^{k+1}) - \nabla f_i(x^k)$, $\mathcal{Q}(\cdot)$ for $i \in \mathcal{G} \cap S_k$ are computed independently
9:     **end for**
10:    $g^{k+1} = \begin{cases} \text{ARAgg} \left( \{g_i^{k+1}\}_{i \in S_k} \right), & \text{if } c_k = 1, \\ g^k + \frac{1}{C} \sum\limits_{i \in S_k} \text{clip}_{\lambda_{k+1}} \left( \mathcal{Q} \left( \widehat{\Delta}_i(x^{k+1}, x^k) \right) \right), & \text{otherwise} \end{cases}$
11: **end for**

---

However, our analysis of Byz-VR-MARINA-PP/Byz-VR-MARINA-PP+ without full-batch gradient computations requires the following additional assumption.

**Assumption G.1.** We assume that there exist $\sigma \geq 0$ such that for all $x \in \mathbb{R}^d$ and $i \in [n]$

$$\mathbb{E}\left[\|\widetilde{\nabla} f_i(x) - \nabla f_i(x)\|^2\right] \leq \frac{\sigma^2}{b'}, \tag{32}$$

where $\widetilde{\nabla} f_i(x)$ is an unbiased $b'$-size mini-batched estimator of $\nabla f_i(x)$.

In particular, when $\frac{1}{m}\sum_{j=1}^m \|\nabla f_{i,j}(x) - \nabla f_i(x)\|^2 \leq \sigma$, which is a standard assumption for variance-reduced methods without full-batch gradient computations (Cutkosky and Orabona, 2019; Li et al., 2021; Gorbunov et al., 2021), estimator $\widetilde{\nabla} f_i(x) = \frac{1}{b'}\sum_{j=1}^{b'} \nabla f_{i,\xi_i^j}(x)$ with $\{\xi_i^j\}_{i\in[n],j\in[m]}$ being i.i.d. samples from the uniform distribution over $[m]$ satisfies (32). Assumption G.1 is also standard for general stochastic optimization (Nemirovski et al., 2009; Ghadimi and Lan, 2013).

## G.1 New Lemma

The main change in the analysis is related to Lemma D.10 since it is the only lemma that relies on the full-batch gradient computation. Nevertheless, it can be easily generalized to the case of Algorithms 4 and 5, as shown in the next result.

**Lemma G.2.** *Let Assumptions D.1, D.5, G.1 hold and Aggregation Operator (ARAgg) satisfy Definition 2.1. Then for all $k \geq 0$ the iterates produced by* Byz-VR-MARINA-PP/Byz-VR-MARINA-PP+ *(Algorithms 4 and 5) satisfy*

$$T_1 = \mathbb{E}\left[\mathbb{E}_k\left[\left\|\mathtt{ARAgg}\left(\{g_i^{k+1}\}_{i\in S_k}\right) - \nabla f(x^{k+1})\right\|^2\right] \mid [1]\right]$$

$$\leq \left(\frac{8G\mathcal{P}_{\mathcal{G}_{\widehat{C}}^k}c\delta B}{(1-\delta)\widehat{C}} + 2\widetilde{B}\right)\mathbb{E}\left[\left\|\nabla f\left(x^k\right)\right\|^2 + L^2\left\|x^{k+1} - x^k\right\|^2\right]$$

$$+ \frac{4G\mathcal{P}_{\mathcal{G}_{\widehat{C}}^k}c\delta B}{(1-\delta)\widehat{C}}\zeta^2 + \widetilde{\zeta}^2 + \left(\frac{\mathcal{P}_{\mathcal{G}_{\widehat{C}}^k}G}{(1-\delta)^2\widehat{C}^2} + 4c\delta\right)\frac{\sigma^2}{b'},$$

*where $\widetilde{B} := 0$ and $\widetilde{\zeta}^2 := 0$ when $\widehat{C} = n$, and $\widetilde{B} := \frac{\mathcal{P}_{\mathcal{G}_{\widehat{C}}^k}GB}{(1-\delta)\widehat{C}}$ and $\widetilde{\zeta}^2 := \frac{\mathcal{P}_{\mathcal{G}_{\widehat{C}}^k}G\zeta^2}{(1-\delta)\widehat{C}}$ when $\widehat{C} < n$.*

*Proof.* Using the definition of aggregation operator, we have

$$T_1 = \mathbb{E}\left[\mathbb{E}_k\left[\left\|\mathtt{ARAgg}\left(\{g_i^{k+1}\}_{i\in S_k}\right) - \nabla f(x^{k+1})\right\|^2\right] \mid [1]\right]$$

$$\overset{(12)}{\leq} \mathbb{E}\left[\mathbb{E}_k\left[\left\|\mathtt{ARAgg}\left(\{g_i^{k+1}\}_{i\in S_k}\right) - \frac{1}{G_{\widehat{C}}^k}\sum_{i\in\mathcal{G}_{\widehat{C}}^k}\widetilde{\nabla} f_i(x^{k+1})\right\|^2\right] \mid [1]\right]$$

$$+ \mathbb{E}\left[\mathbb{E}_k\left[\left\|\frac{1}{G_{\widehat{C}}^k}\sum_{i\in\mathcal{G}_{\widehat{C}}^k}\widetilde{\nabla} f_i(x^{k+1}) - \nabla f(x^{k+1})\right\|^2\right] \mid [1]\right]. \tag{33}$$

To proceed, we estimate the second term in the right-hand side of the above inequality first. From variance decomposition, we have

$$\mathbb{E}\left[\mathbb{E}_k\left[\left\|\frac{1}{G_{\widehat{C}}^k}\sum_{i\in\mathcal{G}_{\widehat{C}}^k}\widetilde{\nabla} f_i(x^{k+1}) - \nabla f(x^{k+1})\right\|^2\right] \mid [1]\right]$$

$$= \mathbb{E}\left[\mathbb{E}_k\left[\left\|\frac{1}{G_{\widehat{C}}^k}\sum_{i\in\mathcal{G}_{\widehat{C}}^k}(\widetilde{\nabla} f_i(x^{k+1}) - \nabla f_i(x^{k+1}))\right\|^2\right] \mid [1]\right]$$

$$+ \mathbb{E}\left[\mathbb{E}_k\left[\left\|\frac{1}{G_{\widehat{C}}^k}\sum_{i\in\mathcal{G}_{\widehat{C}}^k}\nabla f_i(x^{k+1}) - \nabla f(x^{k+1})\right\|^2\right] \mid [1]\right]. \tag{34}$$

The choice of $\widehat{C}$ implies that $G_{\widehat{C}}^k \geq (1-\delta)\widehat{C}$. Moreover, due to the independence of stochastic gradient computations on different workers, we have

$$\mathbb{E}\left[\mathbb{E}_k\left[\left\|\frac{1}{G_{\widehat{C}}^k}\sum_{i\in\mathcal{G}_{\widehat{C}}^k}(\widetilde{\nabla} f_i(x^{k+1}) - \nabla f_i(x^{k+1}))\right\|^2\right]\mid[1]\right]$$

$$\leq \frac{1}{(1-\delta)^2\widehat{C}^2}\mathbb{E}\left[\mathbb{E}_k\left[\left\|\sum_{i\in\mathcal{G}_{\widehat{C}}^k}(\widetilde{\nabla} f_i(x^{k+1}) - \nabla f_i(x^{k+1}))\right\|^2\right]\mid[1]\right]$$

$$= \frac{1}{(1-\delta)^2\widehat{C}^2}\mathbb{E}\left[\sum_{i\in\mathcal{G}_{\widehat{C}}^k}\mathbb{E}_k\left[\left\|\widetilde{\nabla} f_i(x^{k+1}) - \nabla f_i(x^{k+1})\right\|^2\right]\mid[1]\right]$$

$$= \frac{\mathcal{P}_{\mathcal{G}_{\widehat{C}}^k}}{(1-\delta)^2\widehat{C}^2}\sum_{i\in\mathcal{G}}\mathbb{E}\left[\left\|\widetilde{\nabla} f_i(x^{k+1}) - \nabla f_i(x^{k+1})\right\|^2\right] \overset{(32)}{\leq} \frac{\mathcal{P}_{\mathcal{G}_{\widehat{C}}^k}G\sigma^2}{(1-\delta)^2\widehat{C}^2 b'}.$$

Next, since $\frac{1}{G_{\widehat{C}}^k}\sum_{i\in\mathcal{G}_{\widehat{C}}^k}\nabla f_i(x^{k+1}) = \nabla f(x^{k+1})$ with probability 1 when $\widehat{C} = n$, we can estimate the last term in (34) as

$$\mathbb{E}\left[\mathbb{E}_k\left[\left\|\frac{1}{G_{\widehat{C}}^k}\sum_{i\in\mathcal{G}_{\widehat{C}}^k}\nabla f_i(x^{k+1}) - \nabla f(x^{k+1})\right\|^2\right]\mid[1]\right]$$

$$\leq \begin{cases} 0, & \text{if } \widehat{C} = n \\ \mathbb{E}\left[\frac{1}{G_{\widehat{C}}^k}\sum_{i\in\mathcal{G}_{\widehat{C}}^k}\mathbb{E}_k\left[\left\|\nabla f_i(x^{k+1}) - \nabla f(x^{k+1})\right\|^2\right]\mid[1]\right], & \text{if } \widehat{C} < n \end{cases}$$

$$\leq \begin{cases} 0, & \text{if } \widehat{C} = n \\ \frac{\mathcal{P}_{\mathcal{G}_{\widehat{C}}^k}}{(1-\delta)\widehat{C}}\sum_{i\in\mathcal{G}}\mathbb{E}\left[\left\|\nabla f_i(x^{k+1}) - \nabla f(x^{k+1})\right\|^2\right], & \text{if } \widehat{C} < n \end{cases}$$

$$\overset{\text{(As. D.5)}}{\leq} \begin{cases} 0, & \text{if } \widehat{C} = n \\ \frac{\mathcal{P}_{\mathcal{G}_{\widehat{C}}^k}G}{(1-\delta)\widehat{C}}\left(B\mathbb{E}\left[\|\nabla f(x^{k+1})\|^2\right] + \zeta^2\right), & \text{if } \widehat{C} < n \end{cases} = \widetilde{B}\mathbb{E}\left[\|\nabla f(x^{k+1})\|^2\right] + \widetilde{\zeta}^2,$$

where

$$\widetilde{B} := \begin{cases} 0, & \text{if } \widehat{C} = n, \\ \frac{\mathcal{P}_{\mathcal{G}_{\widehat{C}}^k}GB}{(1-\delta)\widehat{C}}, & \text{if } \widehat{C} < n, \end{cases} \quad \text{and} \quad \widetilde{\zeta}^2 := \begin{cases} 0, & \text{if } \widehat{C} = n, \\ \frac{\mathcal{P}_{\mathcal{G}_{\widehat{C}}^k}G\zeta^2}{(1-\delta)\widehat{C}}, & \text{if } \widehat{C} < n. \end{cases}$$

Plugging the derived bounds in (34), we get

$$\mathbb{E}\left[\mathbb{E}_k\left[\left\|\frac{1}{G_{\widehat{C}}^k}\sum_{i\in\mathcal{G}_{\widehat{C}}^k}\widetilde{\nabla} f_i(x^{k+1}) - \nabla f(x^{k+1})\right\|^2\right]\mid[1]\right] \leq \frac{\mathcal{P}_{\mathcal{G}_{\widehat{C}}^k}G\sigma^2}{(1-\delta)^2\widehat{C}^2 b'}$$

$$+ \widetilde{B}\mathbb{E}\left[\|\nabla f(x^{k+1})\|^2\right] + \widetilde{\zeta}^2.$$

Using the above bound in (33), we continue the estimation of $T_1$ as follows:

$$T_1 \overset{(\text{Def. 2.1})}{\leq} \mathbb{E}\left[\frac{c\delta}{G_{\widehat{C}}^k(G_{\widehat{C}}^k - 1)} \sum_{\substack{i,l \in \mathcal{G}_{\widehat{C}}^k \\ i \neq l}} \mathbb{E}_k\left[\left\|\widetilde{\nabla} f_i\left(x^{k+1}\right) - \widetilde{\nabla} f_l\left(x^{k+1}\right)\right\|^2 \mid [1]\right]\right]$$

$$+ \frac{\mathcal{P}_{\mathcal{G}_{\widehat{C}}^k} G \sigma^2}{(1-\delta)^2 \widehat{C}^2 b'} + \widetilde{B}\mathbb{E}\left[\|\nabla f(x^{k+1})\|^2\right] + \widetilde{\zeta}^2$$

$$= \mathbb{E}\left[\frac{c\delta}{G_{\widehat{C}}^k(G_{\widehat{C}}^k - 1)} \sum_{\substack{i,l \in \mathcal{G}_{\widehat{C}}^k \\ i \neq l}} \mathbb{E}_k\left[\left\|\nabla f_i\left(x^{k+1}\right) - \nabla f_l\left(x^{k+1}\right)\right\|^2 \mid [1]\right]\right]$$

$$+ \mathbb{E}\left[\frac{c\delta}{G_{\widehat{C}}^k(G_{\widehat{C}}^k - 1)} \sum_{\substack{i,l \in \mathcal{G}_{\widehat{C}}^k \\ i \neq l}} \mathbb{E}_k\left[\left\|\widetilde{\nabla} f_i\left(x^{k+1}\right) - \nabla f_i(x^{k+1}) - \widetilde{\nabla} f_l\left(x^{k+1}\right) + \nabla f_l\left(x^{k+1}\right)\right\|^2 \mid [1]\right]\right]$$

$$+ \frac{\mathcal{P}_{\mathcal{G}_{\widehat{C}}^k} G \sigma^2}{(1-\delta)^2 \widehat{C}^2 b'} + \widetilde{B}\mathbb{E}\left[\|\nabla f(x^{k+1})\|^2\right] + \widetilde{\zeta}^2,$$

where in the last inequality, we use the conditional independence of $\{\widetilde{\nabla} f_i(x^{k+1})\}_{i \in \mathcal{G}_{\widehat{C}}^k}$ for fixed $x^{k+1}$. Next, using Young's inequality, we derive

$$
T_1 \overset{(12)}{\leq} \mathbb{E}\left[ \frac{c\delta}{G_{\widehat{C}}^k(G_{\widehat{C}}^k - 1)} \sum_{\substack{i,l \in \mathcal{G}_{\widehat{C}}^k \\ i \neq l}} \mathbb{E}\left[ 2\left\| \nabla f_i\left(x^{k+1}\right) - \nabla f\left(x^{k+1}\right) \right\|^2 \mid [1] \right] \right]
$$

$$
+ \mathbb{E}\left[ \frac{c\delta}{G_{\widehat{C}}^k(G_{\widehat{C}}^k - 1)} \sum_{\substack{i,l \in \mathcal{G}_{\widehat{C}}^k \\ i \neq l}} \mathbb{E}\left[ 2\left\| \nabla f_l\left(x^{k+1}\right) - \nabla f\left(x^{k+1}\right) \right\|^2 \mid [1] \right] \right]
$$

$$
+ \mathbb{E}\left[ \frac{c\delta}{G_{\widehat{C}}^k(G_{\widehat{C}}^k - 1)} \sum_{\substack{i,l \in \mathcal{G}_{\widehat{C}}^k \\ i \neq l}} \mathbb{E}\left[ 2\left\| \widetilde{\nabla} f_i\left(x^{k+1}\right) - \nabla f_i\left(x^{k+1}\right) \right\|^2 \mid [1] \right] \right]
$$

$$
+ \mathbb{E}\left[ \frac{c\delta}{G_{\widehat{C}}^k(G_{\widehat{C}}^k - 1)} \sum_{\substack{i,l \in \mathcal{G}_{\widehat{C}}^k \\ i \neq l}} \mathbb{E}\left[ 2\left\| \widetilde{\nabla} f_l\left(x^{k+1}\right) - \nabla f_l\left(x^{k+1}\right) \right\|^2 \mid [1] \right] \right]
$$

$$
+ \frac{\mathcal{P}_{\mathcal{G}_{\widehat{C}}^k} G \sigma^2}{(1-\delta)^2 \widehat{C}^2 b'} + \widetilde{B} \mathbb{E}\left[ \|\nabla f(x^{k+1})\|^2 \right] + \widetilde{\zeta}^2
$$

$$
\overset{(32)}{\leq} \mathbb{E}\left[ \frac{c\delta}{G_{\widehat{C}}^k} \sum_{i \in \mathcal{G}_{\widehat{C}}^k} 4\mathbb{E}_k\left[ \left\| \nabla f_i\left(x^{k+1}\right) - \nabla f\left(x^{k+1}\right) \right\|^2 \mid [1] \right] \right] + \frac{4c\delta\sigma^2}{b'}
$$

$$
+ \frac{\mathcal{P}_{\mathcal{G}_{\widehat{C}}^k} G \sigma^2}{(1-\delta)^2 \widehat{C}^2 b'} + \widetilde{B} \mathbb{E}\left[ \|\nabla f(x^{k+1})\|^2 \right] + \widetilde{\zeta}^2
$$

$$
\leq \frac{\mathcal{P}_{\mathcal{G}_{\widehat{C}}^k} c\delta}{(1-\delta)\widehat{C}} \sum_{i \in \mathcal{G}} 4\mathbb{E}_k\left[ \left\| \nabla f_i\left(x^{k+1}\right) - \nabla f\left(x^{k+1}\right) \right\|^2 \right] + \frac{4c\delta\sigma^2}{b'}
$$

$$
+ \frac{\mathcal{P}_{\mathcal{G}_{\widehat{C}}^k} G \sigma^2}{(1-\delta)^2 \widehat{C}^2 b'} + \widetilde{B} \mathbb{E}\left[ \|\nabla f(x^{k+1})\|^2 \right] + \widetilde{\zeta}^2
$$

$$
\overset{(\text{As. D.5})}{\leq} \left( \frac{4G\mathcal{P}_{\mathcal{G}_{\widehat{C}}^k} c\delta B}{(1-\delta)\widehat{C}} + \widetilde{B} \right) \mathbb{E}\left[ \left\| \nabla f\left(x^{k+1}\right) \right\|^2 \right] + \frac{4G\mathcal{P}_{\mathcal{G}_{\widehat{C}}^k} c\delta B}{(1-\delta)\widehat{C}} \zeta^2 + \widetilde{\zeta}^2
$$

$$
+ \left( \frac{\mathcal{P}_{\mathcal{G}_{\widehat{C}}^k} G}{(1-\delta)^2 \widehat{C}^2} + 4c\delta \right) \frac{\sigma^2}{b'}
$$

$$
\overset{(12)}{\leq} \left( \frac{8G\mathcal{P}_{\mathcal{G}_{\widehat{C}}^k} c\delta B}{(1-\delta)\widehat{C}} + 2\widetilde{B} \right) \mathbb{E}\left[ \left\| \nabla f\left(x^k\right) \right\|^2 + \left\| \nabla f\left(x^{k+1}\right) - \nabla f\left(x^k\right) \right\|^2 \right]
$$

$$
+ \frac{4G\mathcal{P}_{\mathcal{G}_{\widehat{C}}^k} c\delta B}{(1-\delta)\widehat{C}} \zeta^2 + \widetilde{\zeta}^2 + \left( \frac{\mathcal{P}_{\mathcal{G}_{\widehat{C}}^k} G}{(1-\delta)^2 \widehat{C}^2} + 4c\delta \right) \frac{\sigma^2}{b'}
$$

$$
\leq \left( \frac{8G\mathcal{P}_{\mathcal{G}_{\widehat{C}}^k} c\delta B}{(1-\delta)\widehat{C}} + 2\widetilde{B} \right) \mathbb{E}\left[ \left\| \nabla f\left(x^k\right) \right\|^2 + L^2 \left\| x^{k+1} - x^k \right\|^2 \right]
$$

$$
+ \frac{4G\mathcal{P}_{\mathcal{G}_{\widehat{C}}^k} c\delta B}{(1-\delta)\widehat{C}} \zeta^2 + \widetilde{\zeta}^2 + \left( \frac{\mathcal{P}_{\mathcal{G}_{\widehat{C}}^k} G}{(1-\delta)^2 \widehat{C}^2} + 4c\delta \right) \frac{\sigma^2}{b'},
$$

which concludes the proof. $\square$

## G.2 Main Results for Byz-VR-MARINA without Full-Batch Gradient Computations

### G.2.1 General Results

All the lemmas derived in Appendix D.2 hold for Algorithm 4 as well except Lemma D.10, which can be replaced with Lemma G.2, and Lemma D.13 that has the following analog.

**Lemma G.3.** *Let Assumptions 2.3, D.1, D.2, D.3, D.5, G.1 hold and Compression Operator satisfy Definition 2.2. Also, let us introduce the notation*

$$\mathtt{ARAgg}_Q^{k+1} = \mathtt{ARAgg}\left(\mathtt{clip}_{\lambda_{k+1}}\left(\mathcal{Q}\left(\widehat{\Delta}_1(x^{k+1}, x^k)\right)\right), \ldots, \mathtt{clip}_{\lambda_{k+1}}\left(\mathcal{Q}\left(\widehat{\Delta}_C(x^{k+1}, x^k)\right)\right)\right).$$

*Then for all $k \geq 0$ the iterates produced by Byz-VR-MARINA-PP without full-batch gradient computations (Algorithm 4) satisfy*

$$\mathbb{E}\left[\left\|g^{k+1} - \nabla f\left(x^{k+1}\right)\right\|^2\right] \leq \left(1 - \frac{p}{4}\right)\mathbb{E}\left[\left\|g^k - \nabla f\left(x^k\right)\right\|^2\right] + \left(\frac{3\mathcal{P}_{\mathcal{G}_{\widehat{C}}^k}G}{(1-\delta)^2\widehat{C}^2} + 12c\delta\right)\frac{\sigma^2}{b'}$$

$$+ \widehat{B}\mathbb{E}\left[\left\|\nabla f\left(x^k\right)\right\|^2\right] + \widehat{D}\zeta^2 + \frac{pA}{4}\|x^{k+1} - x^k\|^2,$$

*where $A, \widehat{B}, \widehat{D}, p_G, \mathcal{P}_{\mathcal{G}_{\widehat{C}}^k}$ are defined in Lemma D.13.*

*Proof.* Up to the replacement of the bound from Lemma D.10 with the bound from Lemma G.2, the proof of the result is identical to the proof of Lemma D.13. $\square$

**Theorem G.4.** *Let Assumptions 2.3, D.1, D.2, D.3, D.5, G.1 hold. Set $\lambda_{k+1} = 2\max_{i\in\mathcal{G}} L_i \left\|x^{k+1} - x^k\right\|$. Assume that*

$$0 < \gamma \leq \frac{1}{L + \sqrt{A}}, \quad 4\widehat{B} < p,$$

*where $A$ and $\widehat{B}$ are defined in Theorem D.14. Then for all $K \geq 0$ the iterates produced by Byz-VR-MARINA without full-batch gradient computations (Algorithm 4) satisfy*

$$\mathbb{E}\left[\left\|\nabla f\left(\widehat{x}^K\right)\right\|^2\right] \leq \frac{2\Phi^0}{\gamma\left(1 - \frac{4\widehat{B}}{p}\right)(K+1)} + \frac{4\widehat{D}\zeta^2}{p - 4\widehat{B}} + \left(\frac{12\mathcal{P}_{\mathcal{G}_{\widehat{C}}^k}G}{(1-\delta)^2\widehat{C}^2} + 48c\delta\right)\frac{\sigma^2}{b'(p - 4\widehat{B})},$$

*where $\widehat{x}^K$ is chosen uniformly at random from $x^0, x^1, \ldots, x^K$, and $\Phi^0 = f\left(x^0\right) - f^* + \frac{2\gamma}{p}\left\|g^0 - \nabla f\left(x^0\right)\right\|^2$.*

*Proof.* The proof is identical to the proof of Theorem D.14 up to the replacement of Lemma D.13 with Lemma G.3. $\square$

**Theorem G.5.** *Let Assumptions 2.3, D.1, D.2, D.3, D.5, 2.7 hold. Set $\lambda_{k+1} = \max_{i\in\mathcal{G}} L_i \left\|x^{k+1} - x^k\right\|$. Assume that*

$$0 < \gamma \leq \min\left\{\frac{1}{L + \sqrt{2A}}\right\}, \quad 8\widehat{B} < p$$

*where $A$ and $\widehat{B}$ are defined in Theorem D.15. Then for all $K \geq 0$ the iterates produced by Byz-VR-MARINA without full-batch gradient computations (Algorithm 4) satisfy*

$$\mathbb{E}\left[f\left(x^K\right) - f\left(x^*\right)\right] \leq (1-\rho)^K\Phi^0 + \frac{4\widehat{D}\gamma\zeta^2}{p\rho} + \left(\frac{12\mathcal{P}_{\mathcal{G}_{\widehat{C}}^k}G}{(1-\delta)^2\widehat{C}^2} + 48c\delta\right)\frac{\gamma\sigma^2}{b'p\rho},$$

*where $\rho = \min\left[\gamma\mu\left(1 - \frac{8\widehat{B}}{p}\right), \frac{p}{8}\right]$ and $\Phi^0 = f\left(x^0\right) - f^* + \frac{4\gamma}{p}\left\|g^0 - \nabla f\left(x^0\right)\right\|^2$.*

*Proof.* The proof is identical to the proof of Theorem D.15 up to the replacement of Lemma D.13 with Lemma G.3. □

In contrast to their counterparts for Byz-VR-MARINA-PP with (periodical) full-batch gradient computations (Theorems D.14 and D.15), the above results have additional terms proportional to $\frac{\sigma^2}{b'}$ in the upper bounds. These terms cannot be reduced with the decrease of the stepsize but can be made smaller via the increase of $b'$. A similar phenomenon appears in the analysis of the methods with recursive variance reduction even in Byzantine-free case (Fang et al., 2018; Li et al., 2021; Gorbunov et al., 2021), and to address it, $b'$ is typically chosen to be large.

### G.2.2   Results for Bounded Compressors

Similarly to the previous section, we start with an adaptation of Lemma E.3 to the case without full-batch gradient computations.

**Lemma G.6.** *Let Assumptions 2.3, D.1, D.2, D.3, D.4, D.5, G.1, 2.4 hold and the compression operator satisfy Definition 2.2. We set $\lambda_{k+1} = D_Q \max_{i,j} L_{i,j} \|x^{k+1} - x^k\|$. Also, let us introduce the notation*

$$ARAgg_Q^{k+1} = ARAgg\left( clip_{\lambda_{k+1}}\left( \mathcal{Q}\left(\widehat{\Delta}_1(x^{k+1}, x^k)\right)\right), \ldots, clip_{\lambda_{k+1}}\left( \mathcal{Q}\left(\widehat{\Delta}_C(x^{k+1}, x^k)\right)\right)\right).$$

*Then for all $k \geq 0$ the iterates produced by* Byz-VR-MARINA-PP *without full-batch gradient computations (Algorithm 4) satisfy*

$$\mathbb{E}\left[\left\|g^{k+1} - \nabla f\left(x^{k+1}\right)\right\|^2\right] \leq \left(1 - \frac{p}{2}\right)\mathbb{E}\left[\left\|g^k - \nabla f\left(x^k\right)\right\|^2\right] + \left(\frac{3\mathcal{P}_{\mathcal{G}_{\widehat{C}}^k} G}{(1-\delta)^2 \widehat{C}^2} + 12c\delta\right)\frac{\sigma^2}{b'}$$

$$+ \widehat{B}\mathbb{E}\left[\left\|\nabla f\left(x^k\right)\right\|^2\right] + \widehat{D}\zeta^2 + \frac{pA}{4}\|x^{k+1} - x^k\|^2,$$

*where $A, \widehat{B}, \widehat{D}, p_G, \mathcal{P}_{\mathcal{G}_{\widehat{C}}^k}$ are defined in Lemma D.13.*

*Proof.* Up to the replacement of the bound from Lemma D.10 with the bound from Lemma G.2, the proof of the result is identical to the proof of Lemma E.3. □

**Theorem G.7.** *Let Assumptions 2.3, D.1, D.2, D.3, D.4, D.5, G.1, 2.4 hold. Setting $\lambda_{k+1} = \max_{i,j} L_{i,j} \left\|x^{k+1} - x^k\right\|$. Assume that*

$$0 < \gamma \leq \frac{1}{L + \sqrt{A}}, \quad 4\widehat{B} < p,$$

*where $A$ and $\widehat{B}$ are defined in Theorem E.4. Then for all $K \geq 0$ the iterates produced by* Byz-VR-MARINA *without full-batch computations (Algorithm 4) satisfy*

$$\mathbb{E}\left[\left\|\nabla f\left(\widehat{x}^K\right)\right\|^2\right] \leq \frac{2\Phi^0}{\gamma\left(1 - \frac{4\widehat{B}}{p}\right)(K+1)} + \frac{2\widehat{D}\zeta^2}{p - 4\widehat{B}} + \left(\frac{6\mathcal{P}_{\mathcal{G}_{\widehat{C}}^k} G}{(1-\delta)^2 \widehat{C}^2} + 24c\delta\right)\frac{\sigma^2}{b'(p - 4\widehat{B})},$$

*where $\widehat{x}^K$ is chosen uniformly at random from $x^0, x^1, \ldots, x^K$, and $\Phi^0 = f\left(x^0\right) - f^* + \frac{\gamma}{p}\left\|g^0 - \nabla f\left(x^0\right)\right\|^2$.*

*Proof.* The proof is identical to the proof of Theorem E.4 up to the replacement of Lemma E.3 with Lemma G.6. □

**Theorem G.8.** *Let Assumptions 2.3, D.1, D.2, D.3, D.4, D.5, G.1, 2.4, 2.7 hold. Setting $\lambda_{k+1} = \max_{i,j} L_{i,j} \left\|x^{k+1} - x^k\right\|$. Assume that*

$$0 < \gamma \leq \frac{1}{L + \sqrt{2A}}, \quad 8\widehat{B} < p,$$

*where $A$ and $\widehat{B}$ are defined in Theorem E.5. Then for all $K \geq 0$ the iterates produced by* Byz-VR-MARINA *without full-batch computations (Algorithm 4) satisfy*

$$\mathbb{E}\left[f\left(x^K\right) - f\left(x^*\right)\right] \leq (1-\rho)^K \Phi^0 + \frac{2\widehat{D}\zeta^2}{p\rho} + \left(\frac{6\mathcal{P}_{\mathcal{G}_{\widehat{C}}^k} G}{(1-\delta)^2\widehat{C}^2} + 24c\delta\right)\frac{\gamma\sigma^2}{b'p\rho},$$

*where $\rho = \min\left[\gamma\mu\left(1 - \frac{8\widehat{B}}{p}\right), \frac{p}{4}\right]$ and $\Phi^0 = f\left(x^0\right) - f^* + \frac{2\gamma}{p}\left\|g^0 - \nabla f\left(x^0\right)\right\|^2$.*

*Proof.* The proof is identical to the proof of Theorem E.5 up to the replacement of Lemma E.3 with Lemma G.6. □

### G.3 Main Results for Byz-VR-MARINA+ without Full-Batch Gradient Computations

#### G.3.1 Results for Bounded Compressors

Similarly to the analysis of Byz-VR-MARINA without full-batch gradient computations, we start with the adaptation of Lemma F.1 to the no-full-batch gradient computations case.

**Lemma G.9.** *Let Assumptions D.1, D.2, D.3, D.4, D.5, G.1, 2.4 hold and the compression operator satisfy Definition 2.2. Assume that $C \leq G$. We set $\lambda_{k+1} = D_Q \max_{i,j} L_{i,j}\|x^{k+1} - x^k\|$. Then for all $k \geq 0$ the iterates produced by* Byz-VR-MARINA-PP+ *without full-batch gradient computations (Algorithm 5) satisfy*

$$\mathbb{E}\left[\left\|g^{k+1} - \nabla f\left(x^{k+1}\right)\right\|^2\right] \leq \left(1 - \frac{p}{2}\right)\mathbb{E}\left[\left\|g^k - \nabla f\left(x^k\right)\right\|^2\right] + \left(\frac{\mathcal{P}_{\mathcal{G}_{\widehat{C}}^k} G}{(1-\delta)^2\widehat{C}^2} + 4c\delta\right)\frac{p\sigma^2}{b'}$$

$$+ \widehat{B}\mathbb{E}\left[\left\|\nabla f\left(x^k\right)\right\|^2\right] + \widehat{D}\zeta^2 + \frac{pA}{4}\|x^{k+1} - x^k\|^2, \tag{35}$$

*where $A, \widehat{B}, \widehat{D}, p_G, \mathcal{P}_{\mathcal{G}_{\widehat{C}}^k}$ are defined in Lemma F.1.*

*Proof.* Up to the replacement of the bound from Lemma D.10 with the bound from Lemma G.2, the proof of the result is identical to the proof of Lemma F.1. □

**Theorem G.10.** *Let Assumptions D.1, D.2, D.3, D.4, D.5, G.1, 2.4 hold. Set $\lambda_{k+1} = \max_{i,j} L_{i,j}\left\|x^{k+1} - x^k\right\|$. Assume that*

$$0 < \gamma \leq \frac{1}{L + \sqrt{A}}, \quad 4\widehat{B} < p,$$

*where $A$ and $\widehat{B}$ are defined in Theorem F.2. Then for all $K \geq 0$ the iterates produced by* Byz-VR-MARINA+ *without full-batch gradient computations (Algorithm 5) satisfy*

$$\mathbb{E}\left[\left\|\nabla f\left(\widehat{x}^K\right)\right\|^2\right] \leq \frac{2\Phi^0}{\gamma\left(1 - \frac{4\widehat{B}}{p}\right)(K+1)} + \frac{2\widehat{D}\zeta^2}{p - 4\widehat{B}} + \left(\frac{2\mathcal{P}_{\mathcal{G}_{\widehat{C}}^k} G}{(1-\delta)^2\widehat{C}^2} + 8c\delta\right)\frac{p\sigma^2}{b'(p - 4\widehat{B})},$$

*where $\widehat{x}^K$ is chosen uniformly at random from $x^0, x^1, \ldots, x^K$, and $\Phi^0 = f\left(x^0\right) - f^* + \frac{\gamma}{p}\left\|g^0 - \nabla f\left(x^0\right)\right\|^2$.*

*Proof.* The proof is analogous to the proof of Theorem D.14. □

**Theorem G.11.** *Let Assumptions 2.4, D.1, D.2, D.3, D.4, D.5, G.1 2.7 hold. Set $\lambda_{k+1} = \max_{i,j} L_{i,j}\left\|x^{k+1} - x^k\right\|$. Assume that*

$$0 < \gamma \leq \min\left\{\frac{1}{L + \sqrt{2A}}\right\}, \quad 8\widehat{B} < p,$$

*where $A$ and $\widehat{B}$ are defined in Theorem F.3. Then for all $K \geq 0$ the iterates produced by* Byz-VR-MARINA+ *without full-batch gradient computations (Algorithm 5) satisfy*

$$\mathbb{E}\left[f\left(x^K\right) - f\left(x^*\right)\right] \leq (1-\rho)^K \Phi^0 + \frac{2\widehat{D}\zeta^2}{p\rho} + \left(\frac{2\mathcal{P}_{\mathcal{G}_{\widehat{C}}^k} G}{(1-\delta)^2\widehat{C}^2} + 8c\delta\right)\frac{\gamma\sigma^2}{b'\rho},$$

*where $\rho = \min\left[\gamma\mu\left(1 - \frac{8\widehat{B}}{p}\right), \frac{p}{4}\right]$ and $\Phi^0 = f\left(x^0\right) - f^* + \frac{2\gamma}{p}\left\|g^0 - \nabla f\left(x^0\right)\right\|^2$.*

*Proof.* The proof is analogous to the proof of Theorem D.15. □

As in the case of Byz-VR-MARINA, the above upper bounds for Byz-VR-MARINA+ without full-batch gradient computations have additional terms proportional to $\frac{\sigma^2}{b'}$. In contrast to the results for Byz-VR-MARINA+ without full-batch gradient computations, these terms for Byz-VR-MARINA+ are $^1/_p$ times smaller.

# H   Experimental Details and Extra Experiments

## H.1   Experimental Details

For each experiment, we tune the step size using the following set of candidates $\{0.1, 0.01, 0.001\}$. The step size is fixed. We do not use learning rate warmup or decay. We use batches of size 32 for all methods. For partial participation, in each round, we sample $20\%$ of clients uniformly at random. For $\lambda_k = \lambda\|x^k - x^{k-1}\|$ used for clipping, we select $\lambda$ from $\{0.1, 1., 10.\}$. Each experiment is run with three varying random seeds, and we report the mean optimality gap with one standard error. The optimal value is obtained by running gradient descent (GD) on the complete dataset for 1000 epochs. Our implementation of attacks and robust aggregation schemes is based on the public implementation from (Gorbunov et al., 2023).

## H.2   Extra Experiments

Below we provide the missing neural network experiments from the main paper. We consider the MNIST dataset (LeCun and Cortes, 1998) and CIFAR10 (Krizhevsky et al., 2009) (as in (Karimireddy et al., 2021)) with 20 clients, 5 of which are malicious, and 4 clients are sampled in each step. For the attacks, we consider A Little is Enough (ALIE) (Baruch et al., 2019) and the aforementioned Shift-Back (SHB). For the aggregations, we consider coordinate median (CM) (Chen et al., 2017) and robust federated averaging (RFA) (Pillutla et al., 2022) with bucketing. For the MNIST dataset, we use a simple neural network with two convolution layers followed by two fully connected. For CIFAR 10, we use ResNet18 (He et al., 2016) architecture with layer norm. One can note that the results are consistent with the ones provided in the main paper, i.e., clipping performs on par or better than its variant without clipping, and no robust aggregator is able to withstand the shift-back attack without clipping. Our implementation is available at `https://github.com/SamuelHorvath/VR_Byzantine/tree/partial_participation`.

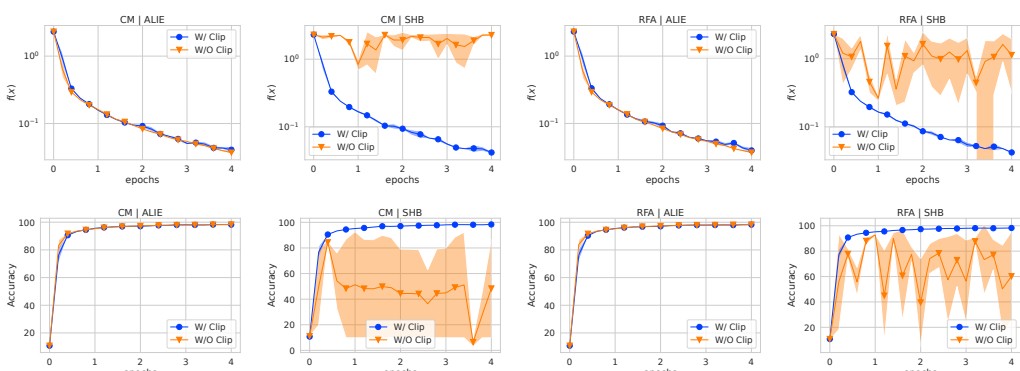

Figure 3: Training loss (top) and test accuracy (bottom) of 2 aggregation rules (CM, RFA) under 4 attacks (BF, LF, ALIE, SHB) on the MNIST dataset under heterogeneous data split with 20 clients, 5 of which are malicious, 4 clients sampled per round.

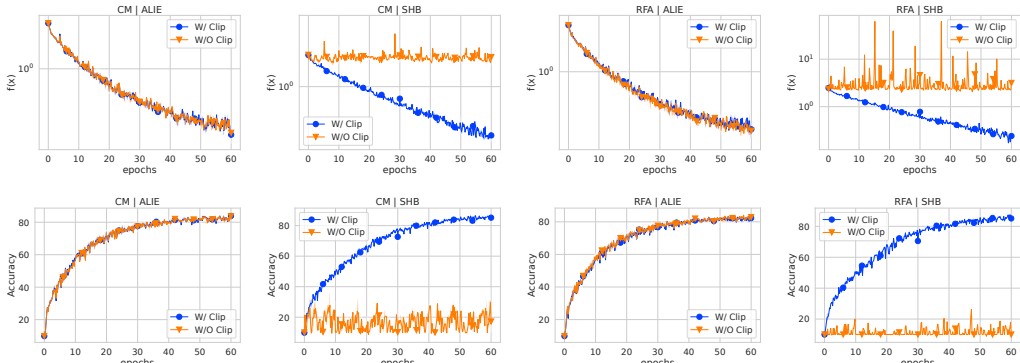

Figure 4: Training loss (top) and test accuracy (bottom) of 2 aggregation rules (CM, RFA) under 4 attacks (BF, LF, ALIE, SHB) on the CIFAR10 dataset under heterogeneous data split with 20 clients, 5 of which are malicious, 4 clients sampled per round.

