# OpenReview forum: "Byzantine Robustness and Partial Participation Can Be Achieved at Once: Just Clip Gradient Differences"
_NeurIPS.cc/2024/Conference — NeurIPS 2024 poster_

### Official Review · Reviewer_xeac · 2024-07-06

**Soundness:** 3
**Presentation:** 2
**Contribution:** 2
**Rating:** 6
**Confidence:** 4

**Summary:**

This paper presents a new distributed learning method called Byz-VR-MARINA-PP, which can achieve Byzantine robustness and partial participation at once. The authors theoretically analyze the convergence and Byzantine robustness of Byz-VR-MARINA-PP. Numerical results of Byz-VR-MARINA-PP are also provided in this paper.

**Strengths:**

The paper is generally well-written. The problem of obtaining Byzantine robustness and partial participation is important in practical applications.

**Weaknesses:**

1. The proposed method is a combination of Byz-VR-MARINA and clipping, and the convergence analysis is similar to that of Byz-VR-MARINA. In light of these, the novelty of the paper is limited.
2. The proposed method requires computing the full gradient with probability $p$ at each iteration, which is computationally expensive, especially when the number of training instances is large.
3. The numerical experiment in this paper is conducted on a9a and MNIST datasets. The scale of these two datasets and the corresponding learning models is quite small given the computation power of today's devices. It would be interesting to see whether the proposed method works on larger datasets and models.

**Questions:**

n/a

---

> ### Author Rebuttal · Authors · 2024-08-04
>
> We thank the reviewer for the feedback and time. Below we address the concerns and comments raised by the reviewer.
>
> >**The proposed method is a combination of Byz-VR-MARINA and clipping, and the convergence analysis is similar to that of Byz-VR-MARINA. In light of these, the novelty of the paper is limited.**
>
> We kindly ask the reviewer to check our general response.
>
> >**The proposed method requires computing the full gradient with probability $p$ at each iteration, which is computationally expensive, especially when the number of training instances is large.**
>
> In practice, one can replace the full gradient computation with just a larger batch computation, i.e., with probability $p$ good workers can compute a mini-batched stochastic gradient with batch-size $b’ > b$ similarly to Geom-SARAH [1].  The usage of periodic full gradient computation is a common issue of many existing variance-reduced methods. Moreover, as the reviewer acknowledged, the considered problem is challenging. Therefore, we believe that despite the mentioned limitation, our work makes an important contribution to the field, and achieving similar results without full gradient/large batch computations at all is an interesting direction for future research. We would like to highlight that we have resolved this issue, at least in practical implementation, where we propose and experimentally analyze a version of our method that works with only a mini-batch gradient oracle (see lines 305-315, 350-363 and Figure 2).
>
> [1] Horváth, Samuel, Lihua Lei, Peter Richtárik, and Michael I. Jordan. "Adaptivity of stochastic gradient methods for nonconvex optimization." SIAM Journal on Mathematics of Data Science 4, no. 2 (2022): 634-648.
>
> >**The numerical experiment in this paper is conducted on a9a and MNIST datasets. The scale of these two datasets and the corresponding learning models is quite small given the computation power of today's devices. It would be interesting to see whether the proposed method works on larger datasets and models.**
>
> Thank you for your comment. As requested, we have included an experiment with a larger model and dataset, namely a heterogeneous split of CIFAR10 with ResNet18 with GroupNorm. The setup for the MNIST dataset is described in the paper.
> Attached (see attached pdf in the main response), we provide a sample of these extra experiments, concretely [Shift Back + Coordinate-wise Mean] and [ALIE + Coordinate-wise Mean]. We note that the results are consistent with the ones provided in the paper, i.e., clipping performs on par or better than its variant without clipping, and no robust aggregator is able to withstand the shift-back attack without clipping. Finally, we are currently working on experiments on even larger datasets and models. If the reviewer has some concrete suggestions, we would like to hear them.
>
> However, we also want to highlight that our work is primarily theoretical, and experiments are mostly needed to illustrate and support our theoretical findings. Moreover, closely related works also consider models and datasets of similar sizes, e.g., , e.g., (Karimireddy et al., 2021) also test their method (Byzantine-Robust Momentum SGD) at the training of MLP on MNIST and (Gorbunov et al., 2023) also test their Byz-VR-MARINA on the logistic regression for a9a dataset. Since in our work, we propose versions of these methods with clipping and partial participation, it was natural for us to consider the same tasks in the experiments.

---

> > ### Comment · Reviewer_xeac · 2024-08-13
> >
> > I thank the authors for the explanation. However, there are some remaining concerns.
> >
> > 1. The authors propose to use a larger batch size $b'$ to avoid the heavy computation of full gradients. However, it is uncertain how this heuristic extension affects the theoretical results. A main contribution of this work is the convergence guarantee of achieving Byzantine robustness when allowing partial participation. Will this heuristic extension damage the theoretical results?
> >
> > 2. I appreciate the authors providing additional experimental results on the CIFAR dataset. However, the results are far from satisfactory. The test accuracy of the ResNet-18 model on the CIFAR-10 dataset can be up to 94% when trained with momentum SGD. In the additional experiment, the test accuracy is only about 50%. Meanwhile, periodically computing full gradients or stochastic gradients with a large batch size is computationally expensive. Therefore, I am not optimistic about the proposed methods scope of practical application.
> >
> > Due to the reasons above, my rating remains unchanged.

---

> > > ### Author Response · Authors · 2024-08-13
> > > **Extra clarification**
> > >
> > > We thank the reviewer for contacting us and would like to elaborate further on the concerns mentioned.
> > >
> > > >*The authors propose to use a larger batch size $b'$ to avoid the heavy computation of full gradients. However, it is uncertain how this heuristic extension affects the theoretical results. A main contribution of this work is the convergence guarantee of achieving Byzantine robustness when allowing partial participation. Will this heuristic extension damage the theoretical results?*
> > >
> > > Such an extension can also be analyzed if we additionally assume that stochastic gradients have uniformly bounded variance $\sigma^2$ (a classical assumption in this case for recursive variance reduction). Then, our analysis will remain almost unchanged: in Lemma D.8, $\zeta^2$ will be replaced by $\zeta^2 + \frac{\sigma^2}{\widehat{C}b'}$ (up to a constant factor). This is well-aligned with a similar term appearing in the analysis of VR-MARINA (see Theorem D.3 from [1]; in particular, for $\widehat{C} = n$ we also have $\frac{\sigma^2}{nb'}$ term). **That is, the modification of Byz-VR-MARINA-PP proposed in our response is guaranteed to converge** to some neighborhood that depends on the variance of stochastic gradients, the number of clients, and the batch size $b'$. **If the number of clients is sufficiently large, which is the case for many FL applications where client sampling is used, then this neighborhood term becomes negligible.**
> > >
> > > >*I appreciate the authors providing additional experimental results on the CIFAR dataset. However, the results are far from satisfactory. The test accuracy of the ResNet-18 model on the CIFAR-10 dataset can be up to 94% when trained with momentum SGD. In the additional experiment, the test accuracy is only about 50%.*
> > >
> > > This experiment's purpose was to show that our approach works for various models. We emphasize that without clipping, the method does not converge under SHB attack, not even to 50%. Recovering the best-known accuracy is not the goal of our experiments. Given the severe time limitations, we ran the methods only for $5$ epochs and did not tune the parameters extensively. Our current experiment corresponds to 200 communication rounds, and our obtained accuracy is consistent or better than other works that also use heterogeneous data split, e.g., see Figure 1 in [3], while we note that in our case, malicious workers are present. For camera-ready, we will provide experiments with 4000 communications rounds (similarly to [3]). Finally, please note that for the heterogeneous split in FL, we would generally expect much smaller final accuracy, e.g., 78% in [3].
> > >
> > > >*Meanwhile, periodically computing full gradients or stochastic gradients with a large batch size is computationally expensive. Therefore, I am not optimistic about the proposed method's scope of practical application.*
> > >
> > > For the experiments with neural networks, we used a heuristic extension of our method described in lines 305-315 of our paper. In particular, we used Byzantine-Robust SGD with momentum from [2] as the base method and applied our heuristic to it. This method does not require full/large batch computations at all.
> > >
> > > **We also note that our work is primarily theoretical, and we find it unfair to give our paper a “reject” score based on the criticism of the experiments.**
> > >
> > > ---
> > >
> > > References:
> > >
> > > [1] Gorbunov et al. "MARINA: Faster Non-Convex Distributed Learning with Compression", ICML 2021
> > >
> > > [2] Karimireddy et al. “Learning from history for Byzantine robust optimization”, ICML 2021
> > >
> > > [3] Reddy et al. “Adaptive Federated Optimization”, ICLR 2021

---

> > > > ### Comment · Reviewer_xeac · 2024-08-13
> > > >
> > > > I thank the authors for the timely and detailed explanation, which has addressed my concern about the effect of the extension on the theoretical results.
> > > >
> > > > Meanwhile, I would like to clarify that the score is not based only on the parts of the experiments. My major concern is about the full gradient (or the stochastic gradient with a large batch size). The full gradients are computationally expensive and empirically has little positive effect on the training of large-scale models in the experiments of existing works. Given the reasons above, it would be interesting to see the performance of Byz-VR-MARINA-PP on the model with a larger scale. Unfortunately, the additional experimental results do not address the concern.
> > > >
> > > > Since my concerns about the theory are almost addressed, the submission looks borderline to me now. I am willing to raise my rating accordingly. In summary, the theoretical results in this paper are generally solid, but the practical performance of the proposed method is not that satisfactory.

---

> > > > > ### Author Response · Authors · 2024-08-13
> > > > > **Additional clarification**
> > > > >
> > > > > We thank the reviewer for increasing the score and participating in the discussion with us. However, we want to provide additional clarifications that are important for making a final decision.
> > > > >
> > > > > >*My major concern is about the full gradient (or the stochastic gradient with a large batch size). The full gradients are computationally expensive and empirically has little positive effect on the training of large-scale models in the experiments of existing works.*
> > > > >
> > > > > As we explained in our previous response, the additional term in the convergence bound for the proposed modification without full batch computation is proportional to $\frac{\sigma^2}{nb'}$. For situations when $n$ is large enough, one can use small $b'$ (same as during other rounds), but the term $\frac{\sigma^2}{nb'}$ will remain small as well. Therefore, this limitation of our method is easily addressable.
> > > > >
> > > > > >*Meanwhile, I would like to clarify that the score is not based only on the parts of the experiments.*
> > > > >
> > > > > If our explanation above addresses the concern about full/large batch computation, then we believe we addressed all the concerns related to the theory and the method. Therefore, from the reviewer's responses, we conclude that the only factor preventing the reviewer from increasing the score further is experiments.
> > > > >
> > > > > >*Given the reasons above, it would be interesting to see the performance of Byz-VR-MARINA-PP on the model with a larger scale. Unfortunately, the additional experimental results do not address the concern.*
> > > > >
> > > > > >*In summary, the theoretical results in this paper are generally solid, but the practical performance of the proposed method is not that satisfactory.*
> > > > >
> > > > > The remaining concerns are related to the experiments, and we are committed to addressing them in the final version of our paper, as we explained in our previous responses.
> > > > >
> > > > > To sum up, **if the reviewer agrees with our responses above, then we kindly ask the reviewer to reconsider their score towards further improvement.** If this is not the case, we kindly ask the reviewer to let us know what are the remaining concerns and why they prevent the reviewer from increasing the score. **Finally, we also emphasize the difficulty of the resolved open question**: partial participation for Byzantine-robust learning was never considered in such generality as we do, e.g., there were no methods allowing Byzantines to form a majority during certain rounds of communication without extra assumptions (such as additional dataset on the server). **We believe it is another aspect justifying a higher score.**

---

> > > > > > ### Comment · Reviewer_xeac · 2024-08-13
> > > > > >
> > > > > > I thank the authors for the further explanation. I agree that the problem considered in this paper is challenging, as presented in the initial review, and I am inclined to further increase my rating. Meanwhile, I have two follow-up questions after reading the explanation, as listed below.
> > > > > >
> > > > > > 1. If I did not get it wrong, a similar theoretical guarantee can also be achieved without full gradients or large-batch stochastic gradients. So I am curious why the full gradient is introduced in the initial version of the proposed method. In other words, **could the authors briefly summarize the advantages and disadvantages of using full gradient periodically (instead of always using small-batch stochastic gradients)?**
> > > > > > Furthermore, to quantitatively study the effect of $b$ or $b'$, it may be helpful to fix the gradient computation number and rewrite the convergence rate in the form of $b$ or $b'$, like those in [Yang et al., 2024]. I understand that the problem setting considered in this paper is much complex and there is only a little time left. This is just a suggestion for future work, and please focus on my question above in bold.
> > > > > >
> > > > > > 2. If full gradients are not necessary to obtain the theoretical result, can Byzantine robustness and partial participation be achieved at once if we just use the proposed clipping strategy with some simpler training methods, such as momentum SGD with robust aggregators? Similarly, given the limited time, a brief discussion is okay.
> > > > > >
> > > > > > Yi-Rui Yang, Chang-Wei Shi, and Wu-Jun Li. "On the Effect of Batch Size in Byzantine-Robust Distributed Learning." ICLR 2024.

---

> > > > > > > ### Author Response · Authors · 2024-08-14
> > > > > > > **Response to Reviewer xeac**
> > > > > > >
> > > > > > > We thank the reviewer for actively engaging in the discussion with us and for their willingness to further increase the score. We address the follow-up questions below.
> > > > > > >
> > > > > > > >*If I did not get it wrong, a similar theoretical guarantee can also be achieved without full gradients or large-batch stochastic gradients. So I am curious why the full gradient is introduced in the initial version of the proposed method.*
> > > > > > >
> > > > > > > Since the proposed modification has an additional term $\frac{\sigma^2}{nb'}$ in the convergence bound and the original version with a full batch computation (with small probability $p$) does not have such terms, we focused on the original version. That is, our paper is mainly theory-focused (though motivated by existing practical challenges), and that is why we considered only the version with better guarantees in the original submission. Nevertheless, the extra term in the convergence bound for the non-full batch version can be negligible in certain situations (as explained in previous responses) and we promise to include the proposed modification in the final version of our work.
> > > > > > >
> > > > > > > >*In other words, could the authors briefly summarize the advantages and disadvantages of using full gradient periodically (instead of always using small-batch stochastic gradients)?*
> > > > > > >
> > > > > > > - The main advantage of using full gradient periodically is described above -- it leads to better convergence bounds than for the version that always uses small-batch.
> > > > > > > - On the other hand, full-batch computation might be computationally expensive (though this computation happens only with a small probability in our method), implying that the method that always uses small-batch has cheaper iterations and is better suited to the case when devices have low computation resources. This can be seen as the main advantage of always using small-batch (and main disadvantage of the original version respectively).
> > > > > > > - In terms of the other aspects, the methods are similar.
> > > > > > >
> > > > > > > >*Furthermore, to quantitatively study the effect of $b$ or $b'$, it may be helpful to fix the gradient computation number and rewrite the convergence rate in the form of $b$ or $b'$, like those in [Yang et al., 2024]. I understand that the problem setting considered in this paper is much complex and there is only a little time left. This is just a suggestion for future work, and please focus on my question above in bold.*
> > > > > > >
> > > > > > > We thank the reviewer for the great suggestion! Indeed, we can explicitly write how the convergence bounds depend on $b$ and $b'$ and then optimize them under the constraint that the computation budget is limited. This can be deduced from our current analysis relatively easily, and we promise to do it in the final version.
> > > > > > >
> > > > > > > >*If full gradients are not necessary to obtain the theoretical result, can Byzantine robustness and partial participation be achieved at once if we just use the proposed clipping strategy with some simpler training methods, such as momentum SGD with robust aggregators? Similarly, given the limited time, a brief discussion is okay.*
> > > > > > >
> > > > > > > This is a great question! In fact, we are currently working on this problem, but the theoretical analysis of Clipped Client Momentum SGD with partial participation becomes quite challenging since the key recurrences (e.g., the ones used in Lemma 9 from [1]) do not hold in this case. Nevertheless, this is a very important direction for future work, and we hope to resolve the mentioned challenges in the near future.
> > > > > > >
> > > > > > > ---
> > > > > > >
> > > > > > > References
> > > > > > >
> > > > > > > [1] Karimireddy et al. “Learning from history for Byzantine robust optimization”, ICML 2021

---

> > > > > > > > ### Comment · Reviewer_xeac · 2024-08-14
> > > > > > > >
> > > > > > > > I thank the authors for the clarification. As I promised, I increased my rating.

---

### Official Review · Reviewer_mV5C · 2024-07-12

**Soundness:** 3
**Presentation:** 4
**Contribution:** 2
**Rating:** 4
**Confidence:** 4

**Summary:**

This paper addresses an important problem: how to achieve Byzantine robustness when the clients partially participate in distributed learning and the Byzantine clients form a majority of sampled clients in some rounds. To solve this problem, the authors propose using the gradient clipping technique to control potential disturbances caused by Byzantine clients. The proposed method, which combines Byz-VR-MARINA and gradient clipping methods, has provable convergence for general smooth non-convex functions and PL functions.

**Strengths:**

1. The paper is well-written and easy to follow.

2. The investigated problem, achieving Byzantine robustness with clients' partial participation, is important and not well understood in the field of  Byzantine-robust distributed learning.

**Weaknesses:**

1. The novelty is limited since the proposed method is a simple combination of the existing method Byz-VR-MARINA and gradient clipping. The idea of using gradient clipping to bound the potential harm caused by Byzantine clients in partial participation is straightforward and not surprising.

2. In Line 225, the authors claim that ''In contrast, Byz-VR-MARINA-PP tolerates any attacks even when all sampled clients are Byzantine workers since the update remains bounded due to the clipping". Why is this the case? Can Byz-VR-MARINA-PP still converge when Byzantine clients constitute a majority of the sampled clients in all rounds? This appears counter-intuitive. I believe there should be specific requirements on the client sample sizes $C$ and $\hat{C}$ concerning the ratio of Byzantine clients $\delta_{\text{real}}$ or the upper bound ratio $\delta$, but I did not find such conditions in Theorem 4.1 and Theorem 4.2. Have I overlooked something? If I have misunderstood any aspects of the paper, please correct me.

3. Since the gradient clipping method is able to control the potential harm caused by Byzantine clients, is it necessary to use the robust aggregator $\text{ARAgg}(\cdot)$? Can the mean aggregator be used as a substitute for the robust aggregator $\text{ARAgg}(\cdot)$?

4. The recent work [1] also considers  partial participation within Byzantine-robust distributed learning. The authors should discuss it in the related works part.

	[1] Allouah, Y., Farhadkhani, S., Guerraoui, R., Gupta, N., Pinot, R., Rizk, G., \& Voitovych, S. (2024). Tackling Byzantine Clients in Federated Learning. arXiv preprint arXiv:2402.12780.

5. Why does partial participation show faster convergence than full participation, as depicted in the middle of Figure 1? Could the authors provide insights or theoretical explanations for this phenomenon?

6. I am curious about how the client's sample size affects the proposed method in practice. Could the authors vary the client's sample size in experiments?

7. In Definition 2.1, '($\delta, c$)-Robust Aggregator' should be '($\delta_{\max}, c$)-Robust Aggregator', as $\delta_{\max}$ represents the breakdown point of the robust aggregator, not $\delta$; please refer to Karimireddy et al. (2021).

**Questions:**

My detailed questions are listed in the above section; please refer to it.

**Limitations:**

The authors have adequately addressed the limitations of their work.

---

> ### Author Rebuttal · Authors · 2024-08-04
>
> We thank the reviewer for the feedback and time. Below, we address the concerns and comments raised by the reviewer.
>
> >**The novelty is limited since the proposed method is a simple combination of the existing method Byz-VR-MARINA and gradient clipping. The idea of using gradient clipping to bound the potential harm caused by Byzantine clients in partial participation is straightforward and not surprising.**
>
> We kindly ask the reviewer to check our general response.
>
> >**In Line 225, the authors claim that ''In contrast, Byz-VR-MARINA-PP tolerates any attacks even when all sampled clients are Byzantine workers since the update remains bounded due to the clipping". Why is this the case? Can Byz-VR-MARINA-PP still converge when Byzantine clients constitute a majority of the sampled clients in all rounds? This appears counter-intuitive. I believe there should be specific requirements on the client sample sizes $C$ and $\widehat{C}$ concerning the ratio of Byzantine clients $\delta_{\text{real}}$ or the upper bound ratio $\delta$, but I did not find such conditions in Theorem 4.1 and Theorem 4.2. Have I overlooked something? If I have misunderstood any aspects of the paper, please correct me.**
>
> As we explain in Footnote 3 on page 6, for our results, we need $\widehat{C} \geq \max\{1, \delta_{\text{real}}n/\delta\}$. This condition ensures that with probability $p$, honest workers are guaranteed to be in the majority during the communication round, which allows the method to “adjust” the update direction $g^{k+1}$. Regarding line 225: we meant that even if for *some communication rounds* all sampled clients are Byzantines, our method provably tolerates this.
>
> >**Since the gradient clipping method is able to control the potential harm caused by Byzantine clients, is it necessary to use the robust aggregator $\text{ARAgg}(\cdot)$? Can the mean aggregator be used as a substitute for the robust aggregator $\text{ARAgg}(\cdot)$?**
>
> This is an excellent question. We have an analysis showing that robust aggregation is needed only with probability $p$ (when a large number of workers participate in the communication round) for the case $C = 1$. The proof can be generalized to the case of any $C > 1$ as well, but this will decrease the probability of steps when the method actually does the progress, and as a result convergence rate will be slower.
>
> >**The recent work [1] also considers partial participation within Byzantine-robust distributed learning. The authors should discuss it in the related works part.
> [1] Allouah, Y., Farhadkhani, S., Guerraoui, R., Gupta, N., Pinot, R., Rizk, G., & Voitovych, S. (2024). Tackling Byzantine Clients in Federated Learning. arXiv preprint arXiv:2402.12780.**
>
> Thank you for the reference; we will add it to the revised version. However, similarly to (Data & Diggavi, 2021), the method from the mentioned paper requires the number of sampled clients to be such that honest clients always form a majority (otherwise, there is a certain probability of divergence, and this probability is growing over time).
>
> >**Why does partial participation show faster convergence than full participation, as depicted in the middle of Figure 1? Could the authors provide insights or theoretical explanations for this phenomenon?**
>
> When the honest clients have similar data, it is natural that partial participation saves a lot of computation, and this is exactly the setup of the experiment presented in Figure 1 (data was homogeneously split between workers). More precisely, the more honest workers participate, the less noisy the aggregated vector is (since they compute stochastic gradients independently; this can be seen from our bounds as well – the larger $C$ and $\widehat{C}$ are, the fewer *communication rounds* the method requires). However, with the increase of participating clients, the total number of stochastic gradient calculations grows, i.e., the overall batch growth. From the standard practice of usage SGD, we know that for many ML problems the best (in terms of computation complexity) batch size is rarely a full batch, and very often, one can achieve reasonable results with a relatively small batch size. This is exactly what our Figure 1 shows.
>
> >**I am curious about how the client's sample size affects the proposed method in practice. Could the authors vary the client's sample size in experiments?**
>
> Thank you for the suggestion. We have added this case for the MNIST dataset, where we sample 1, 4, and 11 clients. We have attached (see attached pdf in the main response) several samples from these experiments on the heterogeneous split with the Shift Back attack and Coordinate-wise Mean as an aggregator. We first compare how clipping behaves across different sample sizes. We note that the smaller the sample size, the faster the convergence in terms of total computations; however, for sample size 1, the method does not converge to the same precision solution due to high noise. For the version without clipping ), we observe convergence only for sample size 11, as this is the only sample size where it is guaranteed that within each round, malicious clients cannot form a majority. Finally, we compare clipping vs. no clipping across these sample sizes. We note that the results are consistent with those provided in the paper, i.e., clipping performs on par or better than its variant without clipping, and no robust aggregator is able to withstand the shift-back attack without clipping unless malicious clients cannot form a majority.
>
> >**Definition 2.1**
>
> We prefer to call it $(\delta,c)$-robust aggregator to emphasize that it requires to know parameter $\delta$. We will add this remark to the final version.

---

> ### Comment · Reviewer_mV5C · 2024-08-11
> **Response to authors**
>
> I thank the authors for their careful responses to each of my comments. The authors have addressed some of my concerns. I will respond to the authors' responses which I think should be further clarified.
>
> A1. Although the authors highlight several technical challenges in their responses, I'm still not convinced by the novelty of the paper. Since the proposed method is a simple combination of two existing methods, I believe the analysis may not be very challenging.
>
> A2.  I believe the condition $\hat{C} \geq \max{1, \delta_{\text{real}} n / \delta}$ is critical and should not be placed in the footnote. In the current presentation of Theorem 4.1 and Theorem 4.2, there seems no guarantee that the Byzantine clients form a majority only in some communication rounds rather than in all rounds.
>
> A3.  If I understand the authors' response correctly, they confirm the need for a robust aggregator to handle Byzantine attacks, even when the gradient clipping method is used. However, I still don’t fully understand why the robust aggregator is necessary. Could the authors please clarify this further?
>
> A5.  I agree that for many machine learning problems, the optimal batch size is not a full batch but rather a smaller one. This is especially true in neural network training, where stochastic noise can help achieve better solutions with smaller generalization errors. Since the objective function in the experiments is strongly convex, does this conclusion still apply? Could the authors provide more insights into this phenomenon?
>
> Given the comments mentioned above, I am keeping my rating unchanged.

---

> > ### Author Response · Authors · 2024-08-11
> > **Response to reviewer**
> >
> > Thank you for your comment!
> >
> > Let us address each issue one by one.
> >
> > A1. We are having difficulty understanding the reviewer's perspective. In our response, we have thoroughly outlined the challenges faced during the analysis and provided **detailed explanations** as to why this work is not merely a "simple combination" of existing techniques. The **reviewer acknowledges** that we have done this. Despite this, the reviewer has not offered any **specific reasoning** to support the belief that the analysis is not particularly challenging. If the analysis were indeed as straightforward as suggested, it raises the question of why this is the **first result** to achieve Byzantine robustness in the context of partial participation without relying on strong assumptions, such as additional data on the server, etc.
> >
> > We believe that a scientific discussion should be based on **well-reasoned** arguments and evidence, rather than subjective feelings or impressions. We would greatly appreciate a more detailed explanation or critique so that we can engage in a meaningful and **constructive** dialogue.
> >
> > A2. Please note that the condition on $\hat{C}$ is only required once every several communication rounds, and this happens with a small probability $p$. We will certainly add a footnote to clarify this point, as suggested.
> >
> > Additionally, it is important to understand that we do not need a situation where the good clients form a majority in every round with absolute certainty. What is necessary is that the probability of good clients forming a majority is greater than zero. If this probability were not greater than zero, it would indicate that the Byzantine clients form a majority not just in a subset of clients (cohort) but across the entire client population. In such a case, it would be impossible to develop any effective method, as the system would be fundamentally compromised.
> >
> > We addressed this crucial point at the beginning of the paper (lines 104-105), but we will ensure that it is emphasized appropriately in the camera-ready version.
> >
> > A3.Let us provide a more detailed clarification on this point. Robust aggregators are designed to function effectively as long as the Byzantine clients do not form a majority, meaning that the proportion of Byzantine clients $\delta$ is within the acceptable range $\delta \leq \delta_{\max} < 0.5$. For a more precise definition, please refer to Definition 2.1 in the paper.
> >
> > However, when the Byzantine clients do form a majority, robust aggregators are no longer effective, as they fail to maintain their robustness under such conditions. To mitigate this issue and handle scenarios where the Byzantine clients might dominate, we incorporate a clipping technique. This approach helps to manage the influence of outliers and reduces the potential impact of malicious clients on the overall aggregation process.
> >
> > We hope this expanded explanation clarifies how our method addresses situations where the number of Byzantine clients might form a majority.
> >
> > A4. Please note that our method employs variance reduction techniques for both data and client sampling. Because the method is designed to reduce variance, increasing the cohort size or batch size does not further reduce variance (as this is already achieved through the variance reduction technique). Consequently, increasing the cohort size or batch size does not lead to substantial gains in convergence. In other words, the benefits of using a larger cohort or batch are limited when variance reduction is already in place.
> >
> > In this context, the situation is similar to what is observed with other variance-reduced methods. Therefore, utilizing a smaller cohort (i.e., a subset of clients) is more advantageous for minimizing communication load. By doing so, we can maintain efficiency and effectiveness while managing the overall communication overhead, which is a key consideration in practical implementations.
> >
> > We hope that we have addressed all the raised issues. If all concerns have been resolved, we would appreciate it if the score could be increased. If there are still any outstanding issues, we remain open to providing further clarification.

---

> > > ### Comment · Reviewer_mV5C · 2024-08-14
> > >
> > > Thank you to the authors for their detailed responses. After carefully reading their replies and theoretical analysis, I am inclined to agree that the analysis may be more challenging than I initially thought, and I will reconsider the paper's contribution. However, some of my other concerns remain unresolved, and I would like to discuss these further.
> > >
> > > A2. The authors claim that "If this probability were not greater than zero, it would indicate that the Byzantine clients form a majority not just in a subset of clients but across the entire client participation." I disagree with this statement for the following reasons. Since the number of iterations $K$ is finite, it is possible, though with a small probability, that Byzantine clients could form a majority in all communication rounds. For example, consider the case where $K = 10$ and $n = 3$, with $B = 1$ Byzantine client. If we only sample $C = 1$ client in each iteration, the Byzantine client would form a majority in all communication rounds with a probability of $(\frac{1}{3})^{10}$. In this scenario, it would be impossible to derive any meaningful results from the proposed method.
> > >
> > > Therefore, I still believe the condition $\hat{C} \geq \max{1, \delta_{\text{real}} n / \delta}$ is critical, as it ensures that Byzantine clients form a majority only in some communication rounds, rather than in all rounds. This condition should be included in Theorem 4.1 or Theorem 4.2, rather than relegated to the footnote.
> > >
> > > A3. According to Karimireddy et al. (2021) and [1], the gradient clipping method can defend against Byzantine attacks when Byzantine clients do not form a majority. Therefore, I think the use of the robust aggregator in the proposed method is unnecessary. I believe that gradient clipping alone is sufficient to handle Byzantine attacks whether the Byzantine clients form a majority or not. Related to this, I think the method from Karimireddy et al. (2021) could be directly applied to the partial participation scenario.
> > >
> > > Beyond the concerns mentioned above, after revisiting the paper, I have a new concern:
> > >
> > > B1. Although the authors prove the convergence of the proposed method in Theorem 4.1 and Theorem 4.2, the learning error seems large. The learning error in Theorems 4.1 and 4.2 is $O(\frac{\hat{D} \xi^2}{p})$ in the non-convex case, which is far from the optimal learning error $O(\delta \xi^2)$ proved by [2] when the probability $p$ is small. I suggest the authors clarify this point further in the paper.
> > >
> > > [1] He, L., Karimireddy, S. P., \& Jaggi, M. (2022). Byzantine-robust decentralized learning via clipped gossip. arXiv preprint arXiv:2202.01545.
> > >
> > > [2] Karimireddy, S. P., He, L., \& Jaggi, M. (2022). Byzantine-Robust Learning on Heterogeneous Datasets via Bucketing. In International Conference on Learning Representations.

---

> > > > ### Author Response · Authors · 2024-08-14
> > > > **Response to Reviewer mV5C**
> > > >
> > > > We thank the reviewer for actively engaging in the discussion with us and for their willingness to reconsider the score. We address the remaining concerns and new questions below.
> > > >
> > > > >*The authors claim that "If this probability were not greater than zero, it would indicate that the Byzantine clients form a majority not just in a subset of clients but across the entire client participation." I disagree with this statement for the following reasons. Since the number of iterations $K$ is finite, it is possible, though with a small probability, that Byzantine clients could form a majority in all communication rounds. For example, consider the case where $K=10$ and $n=3$, with $1$ Byzantine client. If we only sample $1$ client in each iteration, the Byzantine client would form a majority in all communication rounds with a probability of $\left(\frac{1}{10}\right)^{10}$. In this scenario, it would be impossible to derive any meaningful results from the proposed method.*
> > > >
> > > > We believe there is a slight misunderstanding: in the response, we meant that the probability that good clients form a majority **during a particular round** has to be greater than zero. If this is not the case, then it means that a number of good clients is smaller than $C/2$ (otherwise, the probability is strictly larger than zero), which is smaller than $n/2$. However, when the number of good clients is smaller than half of the overall number of clients, it is provably impossible to solve the problem (this is a well-known result, e.g., see [1]).
> > > >
> > > > Next, the provided example fits our setting, and our Theorems 4.1 and 4.2 are valid as well because they provide **in-expectation guarantees** for the squared norm of the gradient/function suboptimality. In-expectation convergence does not imply convergence with probability $1$ (and we never claimed that our method converges with probability $1$).
> > > >
> > > > >*Therefore, I still believe the condition $\widehat{C} \geq \max\lbrace 1, \delta_{\text{real}}n / \delta\rbrace$ is critical...*
> > > >
> > > > We agree that this condition is important for our theoretical results and we will add it to the formulations of Theorems 4.1 and 4.2. However, we would like to emphasize that this condition is needed only for $\widehat{C}$, i.e., for the number of clients that are sampled with a small probability $p$. In contrast, we have no restrictions on $C$, e.g., our theory works even for $C = 1$.
> > > >
> > > > >*According to Karimireddy et al. (2021) and [1]... I think the method from Karimireddy et al. (2021) could be directly applied to the partial participation scenario.*
> > > >
> > > > We want to clarify that Karimireddy et al. (2021) do not use **gradient** clipping, they apply CenteredClipping aggregation, which is an iterative procedure ensuring $(\delta,c)$-robustness of the aggregation. Although CenteredClipping uses a clipping operator inside, it is applied differently from what we do and for a different purpose. As explained in [2], CenteredClipping is a fixed-point iteration for solving a special equation (corresponding to the minimization of Huber-type loss function, see formula (1) in [2]). In contrast, we propose to clip gradient differences without any iterative procedure.
> > > >
> > > > Regarding the direct application of the method from Karimireddy et al. (2021) to the problem of partial participation, we believe that it is quite challenging to show theoretically. In fact, we are currently working on this problem, but the theoretical analysis of Clipped Client Momentum SGD with partial participation becomes quite challenging since the key recurrences (e.g., the ones used in Lemma 9 from [1]) do not hold in this case. Nevertheless, this is a very important direction for future work, and we hope to resolve the mentioned challenges in the near future.
> > > >
> > > > >*...the learning error seems large...*
> > > >
> > > > We agree that the mentioned term is worse than the corresponding term in [3] and we will clarify it in the final version explicitly. However,
> > > >
> > > > - In [3], the authors do not consider a partial participation setup, which is more challenging. Moreover, our method converges as $1/K$, while the method from [3] has $1/\sqrt{K}$ convergence.
> > > >
> > > > - Our term is comparable to the one for Byz-VR-MARINA.
> > > >
> > > > - When the problem is homogeneous ($\zeta = 0$), this term vanishes. We note that a homogeneous setup is also important.
> > > >
> > > > - We have a simplified proof for the case of $C = 1$, where we achieve $O(\widehat{D}\zeta^2)$ error term instead. The key idea is that one can use robust aggregation only with probability $p$ in our method while clipping robustifies the remaining rounds. We will include it in the final version.
> > > >
> > > > **If the reviewer agrees with our clarifications, then we kindly ask to increase the score.**
> > > >
> > > > ---
> > > >
> > > > References
> > > >
> > > > [1] Karimireddy et al. “Learning from history for Byzantine robust optimization”, ICML 2021
> > > >
> > > > [2] Gorbunov et al. "Secure Distributed Training at Scale", ICML 2022
> > > >
> > > > [3] Karimireddy et al. "Byzantine-Robust Learning on Heterogeneous Datasets via Bucketing", ICLR 2022

---

### Official Review · Reviewer_oCpn · 2024-07-14

**Soundness:** 3
**Presentation:** 4
**Contribution:** 3
**Rating:** 5
**Confidence:** 4

**Summary:**

The paper studied the federated learning problem with Byzatine clients and partial participation. The paper proposed a new algorithm called Byzantine-tolerant Variance-Reduced MARINA with Partial Participation or Byz-VR-MARINA-PP and proved its convergence upper bound when the aggregator is a $(\delta, c)$-Robust Aggregator. The key idea of Byz-VR-MARINA-PP is to use clipping. In addition, the paper also proposed a heuristic algorithm in the general case. The performance of the proposed algorithm is verified via experiments as well.

**Strengths:**

1. The paper proposed a new federated learning coping with Byzatine clients and partial participation. The main focus is to consider the partial participation. The algorithm is called Byz-VR-MARINA-PP and the paper proved its convergence upper bound when the aggregator is a $(\delta, c)$-Robust Aggregator.

2. Using the idea of Byz-VR-MARINA-PP, the paper proposed a general algorithm.

3. The paper is very well written. The explanation is very clear.

**Weaknesses:**

1. The proposed algorithm and its analysis focused on the case when the number of local update is 1.

2. There is no any analysis of the heuristic algorithm.

3. The experiments only use two simple datasets, LIBSVM and MNIST.

**Questions:**

1. Can the results be extended when the number of local update is larger than 1?

2. Is it possible to obtain any analysis of the heuristic algorithm under some assumptions?

3. Could you please add more experiments using harder and more popular datasets in Federated Learning?

**Limitations:**

The proposed algorithm and its analysis focused on the case when the number of local update is 1.

---

> ### Author Rebuttal · Authors · 2024-08-04
>
> Thank you for your positive feedback and time.
>
> >**The proposed algorithm and its analysis focused on the case when the number of local update is 1.**
>
> >**Can the results be extended when the number of local update is larger than 1?**
>
> Thank you for raising this point! We understand that in a Federated Learning setting, several local steps are important. However, initially, we need to understand how to deal with partial participation for the provably Byzantine-robust training in case of 1 local update since partial participation is an interesting and complicated question on its own under the presence of Byzantine clients. Moreover, we have page limits for the paper, so we cannot cover all possible scenarios. Taking this into consideration, we leave the analysis of multiple local steps for future work since we believe a proper consideration of the effect of multiple local steps deserves a separate paper.
>
> >**There is no any analysis of the heuristic algorithm.**
>
> We kindly disagree that this is a weakness of our paper. We provided an extensive analysis of the proposed method Byz-VR-MARINA. The heuristic framework is an additional idea that we provide to generalize the proposed algorithm. Note that before this paper, there was no other algorithm with partial participation and with provable Byzantine robust guarantees without additional assumptions on the number of participating clients. Moreover, we have a page limit for the paper, so we cannot cover all settings and analyze all methods.
>
> >**The experiments only use two simple datasets, LIBSVM and MNIST.**
>
> >**Could you please add more experiments using harder and more popular datasets in Federated Learning?**
>
> Thank you for your comment. As requested, we have included an experiment with a larger model and dataset, namely a heterogeneous split of CIFAR10 with ResNet18 with GroupNorm. The setup for the MNIST dataset is described in the paper.
> Attached (see attached pdf in the main response), we provide a sample of these extra experiments, concretely [Shift Back + Coordinate-wise Mean] and [ALIE + Coordinate-wise Mean]. We note that the results are consistent with the ones provided in the paper, i.e., clipping performs on par or better than its variant without clipping, and no robust aggregator is able to withstand the shift-back attack without clipping. Finally, we are currently working on experiments on even larger datasets and models. If the reviewer has some concrete suggestions, we would like to hear them.
>
> > **Is it possible to obtain any analysis of the heuristic algorithm under some assumptions?**
>
> If we assume a similar condition to the smoothness of communicated vectors $g^k_i$ for good workers, we can apply a similar analysis and obtain similar guarantees for the general method. We can add such an analysis in the appendix. However, we are not aware of any examples except for Byz-VR-MARINA, for which this smoothness property holds. We tried to analyze our heuristic extension of Byzantine Robust Momentum SGD (Karimireddy et al., 2021, 2022) but faced some technical difficulties that we have not overcome yet.

---

### Official Review · Reviewer_LGpc · 2024-07-15

**Soundness:** 2
**Presentation:** 3
**Contribution:** 2
**Rating:** 3
**Confidence:** 5

**Summary:**

This paper proposes Byzantine robust approaches in the case of partial participation

**Strengths:**

The paper is well written, but the claims of novelty are problematic, cf bellow.

**Weaknesses:**

The paper starts with a bold claim, "literally, *all* existing methods with provable Byzantine robustness require the full participation of clients." Such a claim overlooks tens or at least a dozen papers on Asynchronous Byzantine machine learning that have been published in the past decade. All of these asynchronous solutions allow various forms of partial participation. Such a claim has no place in a properly researched paper on the topic.

Update : While the analysis made in the paper is non-trivial, it relies on several confusions about what asynchrony means in distributed systems, in particular, any asynchronous systems allows Byzantine nodes not only to form a majority during some communication rounds, but to constitute 100% of the nodes during that round. For instance, in an asynchronous system, one update from one node can be enough to move on to the next iteration.

**Questions:**

Did you compare to the literature on Asynchronous Byzantine ML?

**Limitations:**

Not applicable.

---

> ### Author Rebuttal · Authors · 2024-08-02
>
> >**The paper starts with a bold claim, "literally, all existing methods with provable Byzantine robustness require the full participation of clients." Such a claim overlooks tens, if not hundreds, of papers on Asynchronous Byzantine machine learning that have been published in the past decade. All of these asynchronous solutions allow various forms of partial participation. Such a claim has no place in a properly researched paper on the topic.**
>
> The reviewer provides a very strong and vague claim that our work overlooks “tens, if not hundreds,” of papers that address the problem of partial participation in Byzantine-robust learning **without providing even a single reference** supporting this claim. **The review completely ignores the essence of our paper – new results and algorithms.** This is not a scientific approach to writing the review.
>
> - First of all, we never claimed that there exist no approaches considering Byzantine robustness in the context of partial participation. In contrast, we cite (Data & Diggavi, 2021), who also study this problem, and discuss the relation of our results to their ones in detail.
>
> - Next, in the phrase "literally, all existing methods with provable Byzantine robustness require the full participation of clients", a central part for us is **provable Byzantine robustness**. The existing work (Data & Diggavi, 2021) requires that at each round, the majority of participating clients are honest, which in terms of the theoretical analysis, is very similar to the case of full participation.
>
> - Moreover, we found **just five works** [1-5], not even ten and certainly not even close to hundreds, as the reviewer claims. However, none of these approaches are **provably** robust against Byzantine attacks when there are no additional assumptions (on some extra data or on the frequency filters). Indeed, in [1], the authors propose to use Lipschitz filter and frequency filters in order to filter out Byzantine workers. However, Theorem 4 from [1] establishes convergence only to some neighborhood that depends on the variance of the stochastic gradients and does not depend on the stepsize and number of Byzantine clients. This result is shown for homogeneous data regime, when the convergence to any optimization error can be achieved. Next, in [2, 4], the authors use additional validation data on the server to decide whether to accept the update from workers. This assumption is restrictive for many FL applications when the data on clients is private and is not available on the server. In [3], the authors propose so-called Buffered ASGD (and its momentum version) where the key idea is to split workers into the buffers and wait until each buffer gets at least one gradient update. In the case when the number of buffers is sufficiently large (should be at least $2B$, where $B$ is the number of Byzantine workers), the authors show that BASGD converges. However, this means that to make the step BASGD requires to collect sufficiently large number of gradients such that the good buffers form majority, which is closer to full participation than to the partial participation. We also found work [5], where the authors do not provide theoretical convergence analysis. We will add the discussion of work [1-5] to our paper: **this will be just a minor addition that will not change the main message of our work at all.**
>
> - Finally, the asynchronous communication protocol does not fit the setting of our paper, where we consider clients sampling. For example, one cannot model a synchronous communication with sampling of more than 1 client each round through the asynchronous one. **Therefore, the “criticism” provided by the reviewer is completely unrelated and it does not justify such a low score.**
>
> ---
>
> References:
>
> [1] Damaskinos et al. Asynchronous Byzantine Machine Learning (the case of SGD). ICML 2018
>
> [2] Xie et al. Zeno++: Robust Fully Asynchronous SGD. ICML 2020
>
> [3] Yang & Li. Buffered Asynchronous SGD for Byzantine Learning. JMLR 2023
>
> [4] Fang et al. AFLGuard: Byzantine-robust Asynchronous Federated Learning. ACSAC 2022
>
> [5] Zhang et al. Anti-Byzantine Attacks Enabled Vehicle Selection for Asynchronous Federated Learning in Vehicular Edge Computing. arXiv:2404.08444

---

> > ### Comment · Reviewer_LGpc · 2024-08-13
> >
> > Thanks to the authors for taking the time to reply.
> >
> > For the claim "literally, all existing methods etc." to be correct, there needs to be no single other work with provable Byzantine robustness allowing partial participation. If there exists just one, then the statement is an exaggeration, and the type that demotivates further reading, and I apologies if my review felt harsh in that sense.
> >
> > Regarding existing works, you could add several other contributions on asynchrony in Byzantine robust ML such as (at least) :
> >
> > "Dynamic Byzantine-Robust Learning: Adapting to Switching Byzantine Workers" (ICML 2024)
> > "Robust collaborative learning with linear gradient overhead" (ICML 2023)
> > "Democratizing Machine Learning: Resilient Distributed Learning with Heterogeneous Participants" (IEEE SRDS 2022)
> > "Collaborative learning in the jungle (decentralized, byzantine, heterogeneous, asynchronous and nonconvex learning)" (NeurIPS 2021)
> > "GARFIELD: System Support for Byzantine Machine Learning" (IEEE DSN 2021)
> > "Fault-Tolerance in Distributed Optimization: The Case of Redundancy" (ACM PODC 2020).
> >
> > So yes probably not a hundred but at least a dozen, but again, one is enough not to make the bold and non-nuanced statement the paper have.
> >
> > Most importantly, there seems to be a confusion in the paper and the rest of this discussion about what asynchrony means : unbounded communication delays, please refer to, e.g. Section 5 of the last reference (Fault-Tolerance in Distributed Optimization: The Case of Redundancy) about partial asynchrony.
> >
> > Another aspect of this confusion appears in statements such as "allowing Byzantines to form a majority during certain rounds of communication" (cf comments on openreview). But in an asynchronous setting, adversaries can not only form a majority during a communication round, but the round could also consists *only* of adversaries (i.e. the Byzantine nodes constitute 100% of the updates in such a round). See e.g. "Distributed Algorithms" Lynch 1996.
> >
> > I recognise that my score was biased by the repeated unpleasant experience of reading statements such as "there exists no other method" in modern ML papers, and will update it. I hope you now understand the serious problem in having such a statement, but also address the issues about what partial participation and asynchrony means in a distributed setting. For now, this confusion prevents me from assessing the real contribution of this paper.

---

> > > ### Author Response · Authors · 2024-08-14
> > > **Further clarifications**
> > >
> > > We thank the reviewer for contacting us. Below, we address further comments provided by the reviewer.
> > >
> > > >*For the claim "literally, all existing methods etc." to be correct, there needs to be no single other work with provable Byzantine robustness allowing partial participation. If there exists just one, then the statement is an exaggeration, and the type that demotivates further reading, and I apologies if my review felt harsh in that sense.*
> > >
> > > As we explained in our rebuttal, there is no work that addresses the same problem as we do in the same or comparable generality (without extra assumptions on some extra data or on the frequency filters). We will adjust the writing and add the discussion of extra related work -- it can be done easily and does not change the scientific essence of the paper. **Therefore, the mentioned writing issue is minor -- this should not be the reason for rejection (see NeurIPS 2024 Reviewers Guidelines https://neurips.cc/Conferences/2024/ReviewerGuidelines).**
> > >
> > > >*Regarding existing works, you could add several other contributions on asynchrony in Byzantine robust ML such as (at least):*
> > >
> > > We thank the reviewer for providing additional references.
> > >
> > > - [1] does not consider partial participation.
> > >
> > > - [2] requires all-to-all communication, i.e., it is not applicable to partial participation.
> > >
> > > - [3] requires that the number of sampled clients is such that robust aggregation is possible, i.e., it is necessary for their theoretical results to have majority of honest workers at each communication round, while our work does not have such a requirement.
> > >
> > > - [4] considers the setup when the number of participating clients has to be at least $C \cdot B$, where $C \geq 2$ and $B$ is the overall number of Byzantine workers, i.e., the majority of participating clients need to be honest.
> > >
> > > - [5] proposes a library for Byzantine-Robust Machine Learning and does not provide theoretical results (in particular, it does not provide a theory for partial participation).
> > >
> > > - [6] does not consider partial participation.
> > >
> > > Overall, we would like to highlight that all of the mentioned works focus **on a different problem setup**, and the methods from the mentioned papers **are not guaranteed to converge in the setup we consider**. **Therefore, the mentioned works do not undermine the contribution, novelty, and significance of our paper.**
> > >
> > > >*Most importantly, there seems to be a confusion in the paper and the rest of this discussion about what asynchrony means : unbounded communication delays, please refer to, e.g. Section 5 of the last reference (Fault-Tolerance in Distributed Optimization: The Case of Redundancy) about partial asynchrony.*
> > >
> > > We understand this aspect, but as we already explained, asynchronous settings are quite different from synchronous settings with partial participation. Without additional assumptions on the extra data on the server or frequency filters, in the asynchronous regime, it is impossible to guarantee anything: even $1$ fast Byzantine clients can sequentially send multiple small updates that can shift the model arbitrarily far before good workers send their updates.
> > >
> > > >*Another aspect of this confusion appears in statements such as "allowing Byzantines to form a majority during certain rounds of communication" (cf comments on openreview). But in an asynchronous setting, adversaries can not only form a majority during a communication round, but the round could also consists only of adversaries (i.e. the Byzantine nodes constitute 100% of the updates in such a round). See e.g. "Distributed Algorithms" Lynch 1996.*
> > >
> > > As we explained above, without additional assumptions, asynchronous algorithms may fail in the presence of Byzantine nodes and **they cannot be directly applied to the problem we solve**. Moreover, we are not aware of any other paper (with asynchronous or synchronous communications) that shows **theoretical convergence guarantees in the setup we consider**.
> > >
> > > **If the reviewer agrees with us, we kindly ask the reviewer to further improve the score.**
> > >
> > > ---
> > >
> > > References:
> > >
> > > [1] "Dynamic Byzantine-Robust Learning: Adapting to Switching Byzantine Workers" (ICML 2024)
> > >
> > > [2] "Robust collaborative learning with linear gradient overhead" (ICML 2023)
> > >
> > > [3] "Democratizing Machine Learning: Resilient Distributed Learning with Heterogeneous Participants" (IEEE SRDS 2022)
> > >
> > > [4] "Collaborative learning in the jungle (decentralized, byzantine, heterogeneous, asynchronous and nonconvex learning)" (NeurIPS 2021)
> > >
> > > [5] "GARFIELD: System Support for Byzantine Machine Learning" (IEEE DSN 2021)
> > >
> > > [6] "Fault-Tolerance in Distributed Optimization: The Case of Redundancy" (ACM PODC 2020).

---

### Official Review · Reviewer_RwBg · 2024-07-23

**Soundness:** 4
**Presentation:** 4
**Contribution:** 2
**Rating:** 6
**Confidence:** 3

**Summary:**

This work considers the problem of Byzantine robustness in the framework of federated learning. The main contribution of this work is proposing and analyzing a novel federated algorithm, Byz-VR-MARINA-PP that utilizes gradient clipping. This algorithm is an extension of prior work, Byz-VR-MARINA, but importantly provides for the first time Byzantine robustness guarantees even in partial participation settings (even when Byzantine nodes could form a majority in some of the training rounds). The main idea of this method revolves around limiting the effects of Byzantine clients per round which is a consequence of the gradient clipping. As a result, Byz-VR-MARINA-PP is resilient to shift-back attacks. The authors further strengthen the proposed algorithm by incorporating communication compression. Theoretical results in terms of convergence guarantees are provided. The authors compare their results to SOTA works form the FL with Byzantine-workers literature and showcase the cost of obtaining Byzantine robustness in this more challenging regime. Additionally, numerical results showcase the merits of the proposed method.

**Strengths:**

- The paper addresses a very interesting and relevant problem in the area of Federated Learning.
- The presentation of this work is very good and despite its theoretical depth it is easy to follow.
- The theoretical results derived are comparable to the ones previously known for the full participation regime (despite being somewhat weaker) and the authors are comparing and contrasting their results with prior literature in a fair and clear manner.
- Despite most of the tools used in this work are already known there has being a significant effort to combine them in an efficient manner and the derivation of the theoretical results is non-trivial.

**Weaknesses:**

- As I mentioned before most of the tools used in this work are not novel and as a result the novelty of this work is somewhat limited.
- Although the main focus of this paper is theoretical it has to pointed out that the experimental results are derived in somewhat limited and less challenging settings i.e. a9a LIBSVM and MNIST datasets with 15 good and 5 Byzantine workers.

**Questions:**

- In the main body of the paper the authors chose to use the more restrictive Assumption 2.5 whereas similar analysis has been performed in the Appendix with the less restrictive Assumption D.5.  Could the authors please elaborate on their decision and the differences between the two results?
- On line 304 the authors mention that in some cases partial participation is beneficial to their algorithm. Could they please provide some intuition behind this observation?
- Although the theoretical results are convincing I believe that including more experimental results in broader regimes (of with a few more clients) would provide stronger evidence supporting the superiority of your method. This is not necessary but merely a suggestion.
- In the Introduction/Contributions both "Byz-VR-MARINA-PP" and "By-VR-MARINA-PP" are met. Is this a typo?

**Limitations:**

The authors adequately addressed the limitations of their work.

---

> ### Author Rebuttal · Authors · 2024-08-04
>
> Thank you for your detailed feedback and time. We appreciate your positive evaluation of our work.
>
> >**As I mentioned before most of the tools used in this work are not novel and as a result the novelty of this work is somewhat limited.**
>
> We kindly ask the reviewer to check our general response.
>
> >**Although the main focus of this paper is theoretical it has to pointed out that the experimental results are derived in somewhat limited and less challenging settings i.e. a9a LIBSVM and MNIST datasets with 15 good and 5 Byzantine workers.**
> >**Although the theoretical results are convincing I believe that including more experimental results in broader regimes (of with a few more clients) would provide stronger evidence supporting the superiority of your method. This is not necessary but merely a suggestion.**
>
> Thank you for your comment! As requested, we have included an experiment with a larger model and dataset, namely a heterogeneous split of CIFAR10 with ResNet18 with GroupNorm. The setup for the MNIST dataset is described in the paper.
> Attached (see attached pdf in the main response), we provide a sample of these extra experiments, concretely [Shift Back + Coordinate-wise Mean] and [ALIE + Coordinate-wise Mean]. We note that the results are consistent with the ones provided in the paper, i.e., clipping performs on par or better than its variant without clipping, and no robust aggregator is able to withstand the shift-back attack without clipping. Finally, we are currently working on the experiments on even larger datasets and models, also with a larger number of clients.
>
> >**In the main body of the paper the authors chose to use the more restrictive Assumption 2.5 whereas similar analysis has been performed in the Appendix with the less restrictive Assumption D.5. Could the authors please elaborate on their decision and the differences between the two results?**
>
> Assumption D.5 is a direct generalization of Assumption 2.5 in the case of $B=0$. In the main part of the paper, we decided to consider a simplified version of the assumption to make it easier to read. The main message and the core ideas are still valid in the case of a simplified version of the assumption, but it allows us to present the result in a more compact and cleaner form. Also, we believe that obtaining convergence results for more complicated settings is important and we provide analysis for the more general case supplementary materials section. In contrast to Assumption 2.5, in Assumption D.5, instead of uniformly bounded heterogeneity, we have a bound that depends on the norm of the full gradient and constant, which makes this bound much more general.
>
> >**On line 304 the authors mention that in some cases partial participation is beneficial to their algorithm. Could they please provide some intuition behind this observation?**
>
> The key reason for this phenomenon is that regardless of the real number of participating clients, the usage of $(\delta,c)$-robust aggregator affects the final rate (i.e., stepsize) through the terms depending on parameter of the aggregator $\delta$ (e.g., see formula (7)). Therefore, in the situation described in lines 296-301, the rate depends on two terms: one is decreasing in $C$ (roughly speaking, it corresponds to the decrease of the variance of the stochastic gradient $\sim C$ times since the batchsize is increased $\geq C/2$ times for each round in the worst case) and the second one is independent of $C$ and depends only on $\delta$. Therefore, when the second term dominates the first one, it is optimal to decrease $C$ as long as the second term remains the main one and as long as $C \geq \max\{1, \delta_{\text{real}}n / \delta\}$. Such a strategy allows to save on overall computations and at the same time keeps the number of communication rounds the same.
>
> Moreover, from the practical perspective, when the honest clients have similar data, it is natural that partial participation saves a lot of computation and this is exactly the setup of the experiment presented in Figure 1 (data was homogeneously split between workers). More precisely, the more honest workers participate, the less noisy the aggregated vector is (since the compute stochastic gradients independently; this can be seen from our bounds as well – the larger $C$ and $\widehat{C}$ are, the fewer *communication rounds* the method requires). However, with the increase of participating clients, the total number of stochastic gradient calculations grows, i.e., the overall batch growth. From the standard practice of usage SGD, we know that for many ML problems the best (in terms of computation complexity) batch size is rarely a full batch and very often one can achieve reasonable results with relatively small batch size. This is exactly what our Figure 1 shows.
>
> >**In the Introduction/Contributions both "Byz-VR-MARINA-PP" and "By-VR-MARINA-PP" are met. Is this a typo?**
>
> Thank you for noting this! It is indeed a typo, and "Byz-VR-MARINA-PP" is the correct option. We will fix this typo and also check for other typos in the text.

---

> > ### Comment · Reviewer_RwBg · 2024-08-11
> > **Post Rebuttal**
> >
> > I appreciate the authors efforts to address my question and include more experiments.
> >
> > After carefully reading the comments from the other reviewers and the responses of the authors I find that all my concerns are addressed. I am inclined to keep my score and I look forward to the reviewers discussion.

---

> > > ### Author Response · Authors · 2024-08-13
> > > **Thank you**
> > >
> > > Dear Reviewer,
> > >
> > > Thank you for checking our responses and other reviews. We are glad to hear that we addressed all your concerns. We would also appreciate it if you could champion our paper in discussion with other reviewers: as the reviewer acknowledged and as we explain in the paper, **the problem we addressed is very important and non-trivial, and we managed to circumvent multiple technical challenges to achieve our results (as explained in the general rebuttal message)**.
> > >
> > > Thank you once again for your feedback and time. If the paper gets accepted, we promise to incorporate all the requested changes and add extra experiments attached to the general rebuttal message, in particular, to the camera-ready version.
> > >
> > > Best regards,
> > >
> > > Authors

---

### Author Rebuttal · Authors · 2024-08-04

We thank the reviewers for their feedback and time. Since several reviewers had concerns about the novelty, which we kindly but firmly disagree with, we prepared a general message addressing this.

As we mention in the introduction (page 2, left column, lines 49-54) and explain in Section 3 (paragraph “New ingredients: client sampling and clipping”), all existing methods (and Byz-VR-MARINA in particular) cannot be naively combined with client sampling/partial participation with an arbitrary small number of sampled clients: this can lead to communication rounds when Byzantine workers form a majority, which allows them to shift the updates arbitrary far from the solution. To handle such situations, we propose to use gradient clipping to the vectors that are communicated by the clients. As we explain in the same paragraph (page 6, lines 226-230), gradient clipping makes the updates bounded. Therefore, even if Byzantine workers accidentally form a majority during some rounds, the norm of the shift they can produce is bounded by a clipping level $\lambda_{k+1}$.

However, the introduction of the clipping does not come for free: if the clipping level is too small, clipping can create a noticeable bias to the updates. Because of this issue, existing works such as (Gorbunov et al., 2020; Zhang et al., 2020) use non-trivial policies for the choice of the clipping level, and the analysis in these works differs significantly from the existing analysis for the methods without clipping. The analysis of Byz-VR-MARINA is based on the unbiasedness of vectors $\mathcal{Q}(\hat \Delta_i(x^{k+1}, x^k))$, i.e., on the following identity: $\mathbb{E}[\mathcal{Q}(\hat \Delta_i(x^{k+1}, x^k)) \mid x^{k+1}, x^k] = \Delta_i(x^{k+1}, x^k) = \nabla f_i(x^{k+1}) - \nabla f_i(x^k)$. Since $\mathbb{E}[\text{clip}\_{\lambda\_{k+1}}(\mathcal{Q}(\hat \Delta_i(x^{k+1}, x^k))) \mid x^{k+1}, x^k] \neq \nabla f\_i(x^{k+1}) - \nabla f\_i(x^k)$ in general, to analyze Byz-VR-MARINA-PP we also use a special choice of the clipping level: $\lambda_{k+1} = \alpha_{k+1} \|\|x^{k+1} - x^k\|\|$. To illustrate the main reasons for that, let us consider the case of uncompressed communication ($\mathcal{Q}(x) \equiv x$). In this setup, for large enough $\alpha_{k+1}$ we have $\text{clip}\_{\lambda\_{k+1}}\hat \Delta\_i(x^{k+1}, x^k) = \hat \Delta\_i(x^{k+1}, x^k)$ for all $i\in \mathcal{G}$ (due to Assumption 2.6), which allows us using a similar proof to the one for Byz-VR-MARINA when good workers form a majority in a round. Moreover, when Byzantine workers form a majority, our choice of the clipping level allows us to bound the second moment of the shift from the Byzantine workers as $\sim \|\| x^{k+1} - x^k \|\|^2$ (see Lemmas D.9 and D.12), i.e., the second moment of the shift is of the same scale as the variance of $\lbrace g_i \rbrace_{i\in \mathcal{G}}$, which goes to zero (see page 5, lines 205-209). Next, to properly analyze these two situations, we overcame another technical challenge related to the estimation of the conditional expectations and probabilities of corresponding events (see Lemmas D.9 - D.10 and formulas for $p_G$ and $\mathcal{P}_{\mathcal{G}_C^k}$ at the beginning of Section 4). In particular, the derivation of formula (22) is quite non-standard for stochastic optimization literature: there are two sources of stochasticity – one comes from the sampling of clients and the other one comes from the sampling of stochastic gradients and compression. This leads to the estimation of variance of the average of the random number of random vectors, which is novel on its own. In addition, when the compression operator is used, the analysis becomes even more involved since one cannot directly apply the main property of unbiased compression (Definition 2.2), and we use Lemma D.6 in the proof to address this issue. It is also worth mentioning that in contrast to Byz-VR-MARINA, our method does not require full participation even with a small probability $p$. Instead, it is sufficient for Byz-VR-MARINA-PP to sample a large enough cohort of $\hat{C}$ clients with probability $p$ to ensure that Byzantine workers form a minority in such rounds. **Taking into account all of these multiple technical challenges that we circumvented, we believe that our choice of the clipping level is not obvious beforehand, and our analysis significantly differs from the analysis of Byz-VR-MARINA.**

Finally, we also want to emphasize that the idea of using gradient clipping to handle the Byzantine workers in the case of partial participation is novel on its own. Karimireddy et al. (2021) used clipping to construct robust aggregation, but it was never used in the way we apply it. We believe our work is an important step towards building more efficient Byzantine-robust methods supporting partial participation.

If our paper gets accepted, we will expand these clarifications in the main text (the accepted papers have one extra page).

---

### Decision · Program_Chairs · 2024-09-25

**Decision:**

Accept (poster)

**Comment:**

This paper considers federated learning with Byzantine clients and partial participation. The problem is of practical importance, but lack of investigation. The reviewer raise several concerns, such as lack of new elements in the algorithm design (the proposed algorithm is the combination of several existing tools), lack of numerical experiments (the current numerical experiments are relatively simple), etc. However, the strengths of this paper outweigh its weaknesses.

- Reviewer LGpc commented on the connection between partial participation and asynchronous participation. These two settings are tightly related but different. The investigated problem is originated from the recent federated learning applications, and has its own importance.

- Reviewer mV5C pointed that the main weakness of this paper is the combination of existing tools. Considering the importance and timeliness of the investigated problem, the proposed method would potentially motivate further research efforts. In addition, the analysis made in the paper is non-trivial.

Therefore, I recommend to accept.